# Gasdermin D in macrophages drives orchitis by regulating inflammation and antigen presentation processes

Chunmei Ma [1,3], Jiajia Huang [2,3], Yuying Jiang [1,3], Lu Liu [1], Na Wang [1], Shaoqiong Huang [1], Honghui Li [1], Xiangyu Zhang [1], Shuang Wen [1], Bingwei Wang [2✉] & Shuo Yang [1✉]

## Abstract

Inflammation in the testes induced by infection and autoimmunity contributes significantly to male infertility, a public health issue. Current therapies using antibiotics and broad-spectrum anti-inflammatory drugs are ineffective against non-bacterial orchitis and induce side effects. This highlights the need to explore the pathogenesis of orchitis and develop alternative therapeutic strategies. In this study, we demonstrated that Gasdermin D (GSDMD) was activated in the testes during uropathogenic *Escherichia coli* (UPEC)-induced acute orchitis, and that GSDMD in macrophages induced inflammation and affected spermatogenesis during acute and chronic orchitis. In testicular macrophages, GSDMD promoted inflammation and antigen presentation, thereby enhancing the T-cell response after orchitis. Furthermore, the pharmacological inhibition of GSDMD alleviated the symptoms of UPEC-induced acute orchitis. Collectively, these findings provide the first demonstration of GSDMD's role in driving orchitis and suggest that GSDMD may be a potential therapeutic target for treating orchitis.

**Keywords** GSDMD; Pyroptosis; Testicular Macrophage; Antigen Presentation; Orchitis
**Subject Categories** Immunology; Urogenital System

## Introduction

Infertility affects ~15% couples of childbearing age worldwide, and approximately half of the cases are attributed to male infertility. Infection, autoimmunity, and inflammation of the reproductive tract have been reported as etiological factors affecting male fertility (Fijak et al, 2018). Approximately 6−15% cases of male infertility is related to infection and inflammation of the urogenital tract (Fijak et al, 2018). Uropathogenic *Escherichia coli* (UPEC) has been identified as the predominant etiological pathogen of genitourinary infections and can be isolated from 50 to 95% patients with genitourinary infections (Wiles et al, 2008). Bacterial orchitis is treated with antibiotics (Trojian et al, 2009). However, antibiotic treatment cannot restore fertility completely because of epididymal and testicular inflammatory impairment in most cases (Salonia et al, 2020; Schuppe et al, 2008). In addition, broad-spectrum anti-inflammatory drugs, such as aspirin and ibuprofen, are used for the symptomatic treatment of bacterial and autoimmune orchitis; however, these drugs target non-specifically and may disrupt endocrine and sperm development (Albert et al, 2013). Thus, further elucidation of the immunopathological mechanisms underlying orchitis is essential for developing novel therapeutic strategies.

The testis has a unique immunological environment characterized by exceptional immune privilege for immunogenic germ cells. Multiple mechanisms and factors, including physical structure, local immunosuppressive milieu, and systemic immune tolerance, coordinate testicular immune privilege (Zhao et al, 2014). The testes consist of seminiferous tubules and interstitial spaces. Seminiferous tubules are the sites of spermatogenesis, and the interstitial spaces located between the seminiferous tubules harbor Leydig cells, fibroblasts, and a large population of immune cells (Guo et al, 2018). Under physiological conditions, testicular macrophages account for the largest proportion of immune cells in the testicular interstitial spaces (Zhao et al, 2014), and they have been reported to maintain immune privilege and regulate the development and steroidogenesis of Leydig cells (Bhushan and Meinhardt, 2017; Hutson, 2006). Accumulating evidence suggests that testicular macrophages play an essential role in the development of orchitis caused by infection and sterile inflammation. During orchitis, inflammatory testicular macrophages, mainly from circulating monocytes, promote tissue damage and adversely affect spermatogenesis (Wang et al, 2021). In addition, peritubular testicular macrophages have been reported to express high levels of MHC (Mossadegh-Keller et al, 2017), suggesting that peritubular testicular macrophages may have the ability to present antigens to T cells.

Testicular macrophages utilize pattern recognition receptors, including Toll-like receptors (TLRs), retinoic acid-inducible gene I (RIG-I)-like receptors, and nucleotide-binding oligomerization

[1]Department of Immunology, State Key Laboratory of Reproductive Medicine, Jiangsu Key Lab of Cancer Biomarkers, Prevention and Treatment, Collaborative Innovation Center for Personalized Cancer Medicine,National Vaccine Innovation Platform, The Affiliated Wuxi People's Hospital of Nanjing Medical University, Wuxi People's Hospital, Wuxi Medical Center, Nanjing Medical University, 211166 Nanjing, China. [2]Department of Pharmacology, Nanjing University of Chinese Medicine, 138 Xianlin Avenue, 210023 Nanjing, China. [3]These authors contributed equally: Chunmei Ma, Jiajia Huang, Yuying Jiang. ✉E-mail: bingweiwang@njucm.edu.cn; shuoyang01@njmu.edu.cn

domain protein-like receptors (NLRs), to elicit local immune response against pathogen invasion and tissue damage (Zhao et al, 2014). Among these receptors, NLRs, together with apoptosis-associated speck-like protein (ASC) and proinflammatory caspases (caspase-1 and caspase-11), can be assembled into oligomeric complexes, termed inflammasomes, leading to caspase autoactivation. Subsequently, activated proinflammatory caspases manipulate the cleavage of pro-interleukin (IL)-1β and pro-IL-18 precursors into their mature forms, and also trigger cell pyroptosis (Rathinam et al, 2012). In addition to NLRs, absent in melanoma 2 (AIM2) have been shown to assemble into inflammasomes (Hornung et al, 2009). IL-1β is well known to contribute to the damage of testicular tissue (Guazzone et al, 2009). Several independent studies have shown that the NLRP3 inflammasome is involved in the pathogenesis of orchitis (Cao et al; 2019; Li et al; 2019b; Su et al, 2020). However, clinical progress in the treatment of orchitis by targeting NLRP3 is lacking. Therefore, further elucidation of other relevant molecular mechanisms underlying orchitis is required to develop novel therapeutic strategies. Gasdermin D (GSDMD), a pore-forming protein, is a key downstream effector of cell pyroptosis by inflammasomes (Shi et al, 2015). Upon inflammasome activation, inflammatory caspases cleave GSDMD from their central linker domains. Cleavage of GSDMD releases its N-terminal fragments, followed by oligomerization and formation of membrane pores, which triggers lytic cell death or secretion of IL-1β and IL-18 (Liu et al, 2016). Accumulating evidence has revealed that GSDMD plays an important role in bacterial infections and autoinflammatory diseases such as experimental autoimmune encephalomyelitis and colitis (Li et al, 2019a; Ma et al; 2020). Interestingly, a recent study has shown that GSDMD is activated in the testes during cadmium-induced testicular damage (Zhou et al, 2022). However, the pathophysiological function of GSDMD in orchitis remains largely unknown.

In this study, we found that GSDMD played an important role in exacerbating UPEC-induced acute orchitis and chronic experimental autoimmune orchitis (EAO). In addition, we showed that GSDMD-induced pyroptosis in macrophages enhanced antigen presentation, thereby amplifying T-cell immune responses during orchitis.

# Results

## GSDMD was highly activated in the testes of mice with UPEC-induced acute orchitis

UPEC infection can induce epididymo-orchitis in mice, and the epididymis is known to be vulnerable to bacterial infection and concomitant inflammation (Fijak et al, 2018; Klein et al, 2020; Klein et al, 2019; Michel et al, 2016). However, few studies have investigated the direct effects of UPEC on the testes. In the present study, we modified the UPEC infection model based on a previous study (Klein et al, 2020). Briefly, UPEC was directly injected into the upper pole of the testis, followed by ligation of the vas deferens to prevent the spread of the infection. After treatment, the testes exhibited swelling, and the testis–body weight ratio increased significantly (Fig. EV1A). Histopathological analysis (hematoxylin and eosin; H&E staining) showed inflammatory cell infiltration, damaged seminiferous tubules, and impaired spermatogenesis, characterized by decreased tubules containing elongated spermatids (ES), and increased tubules containing round

spermatids (RS), pachytene spermatocytes (PSc), spermatogonia (SG), or Sertoli cells only (SCO) (Fig. EV1B). Computer-assisted sperm analysis (CASA) consistently showed a reduction in sperm count and sperm motility, such as the ratio of motile sperms and progressive sperms, and velocity of motile sperms (average path velocity, VAP; curvilinear velocity, VCL; straight-line velocity, VSL) (Fig. EV1C). We also observed elevated expression of proinflammatory cytokines and chemokines, including Il-6, Tnfa, Ccl-2, and Cxcl-10 (Fig. EV1D). Thus, UPEC-induced acute orchitis was established successfully.

To investigate whether GSDMD was involved in the pathogenesis of UPEC-induced acute orchitis, we first determined the expression patterns of GSDMD and other inflammasome-related molecules in the testes using qRT-PCR, western blotting, and immunofluorescence analysis after orthotopic injection of UPEC (Fig. 1A). The expression of inflammasome-related molecules, such as NLRP3, AIM2, ASC, CASPASE-1, CASPASE-11 and IL-1β, and GSDMD, in the testes of the orchitic mice were significantly increased (Fig. 1B,C). Notably, the cleavage of CASPASE-1 and GSDMD also enhanced (Fig. 1C). Furthermore, immunofluorescence analysis revealed that NLRP3, CASPASE-1, and GSDMD were upregulated in the interstitial spaces (Fig. 1D). Collectively, these data suggested that GSDMD was involved in UPEC-induced testicular inflammation.

## GSDMD deficiency controlled UPEC-induced acute orchitis

To assess whether GSDMD plays a potential role in the testes under homeostatic conditions, we analyzed testicular structure and spermatogenesis in WT and $Gsdmd^{-/-}$ mice. $Nlrp3^{-/-}$, $Aim2^{-/-}$, $Asc^{-/-}$, and $Caspase-1/11^{-/-}$ mice were used as inflammasome controls. Unexpectedly, we found that the testes from $Gsdmd^{-/-}$ and $Asc^{-/-}$ mice were smaller than those from WT mice (Fig. EV2A). Furthermore, $Gsdmd^{-/-}$ and $Asc^{-/-}$ mice showed increase in seminiferous tubular diameter, and reduction in sperm count and number of tubules containing ES, although all mice exhibited slight increase in the number of tubules containing PSc and decrease in the ratio of motile sperms (Fig. EV2B-E). However, the expression of proinflammatory cytokines and chemokines in testes did not differ among the above knockout and WT mice (Fig. EV2F,G). Collectively, these data suggested that GSDMD or ASC deficiency slightly impaired spermatogenesis and seminiferous tubules under homeostatic conditions, which cannot be attributed to inflammation.

To directly investigate the effect of GSDMD on orchitis, UPEC was injected into the testes of age-matched $Gsdmd^{-/-}$ and WT mice in situ and their orchitis phenotypes were compared. $Nlrp3^{-/-}$, $Aim2^{-/-}$, $Asc^{-/-}$, and $Caspase-1/11^{-/-}$ mice were used as inflammasome controls. The testes from $Gsdmd^{-/-}$ mice were smaller than those from WT mice, and the testis-to-body weight ratio in $Gsdmd^{-/-}$ mice also decreased (Fig. 2A). H&E staining showed that GSDMD deficiency led to reduction in inflammatory cell infiltration and seminiferous tubule damage, characterized by decreased tubules containing SCO, SG, PSc, and RS, and increased tubules containing ES in response to UPEC (Figs. 2B and EV2H). Consistently, the results of CASA showed that sperm count, frequency of motile sperms, ratio of progressive sperms, and VAP, VSL, and VCL of motile sperms were significantly higher in $Gsdmd^{-/-}$ mice than those in WT mice (Fig. 2C,D), indicating that GSDMD deficiency significantly enhanced the quality of sperms during orchitis. We also analyzed the expression of

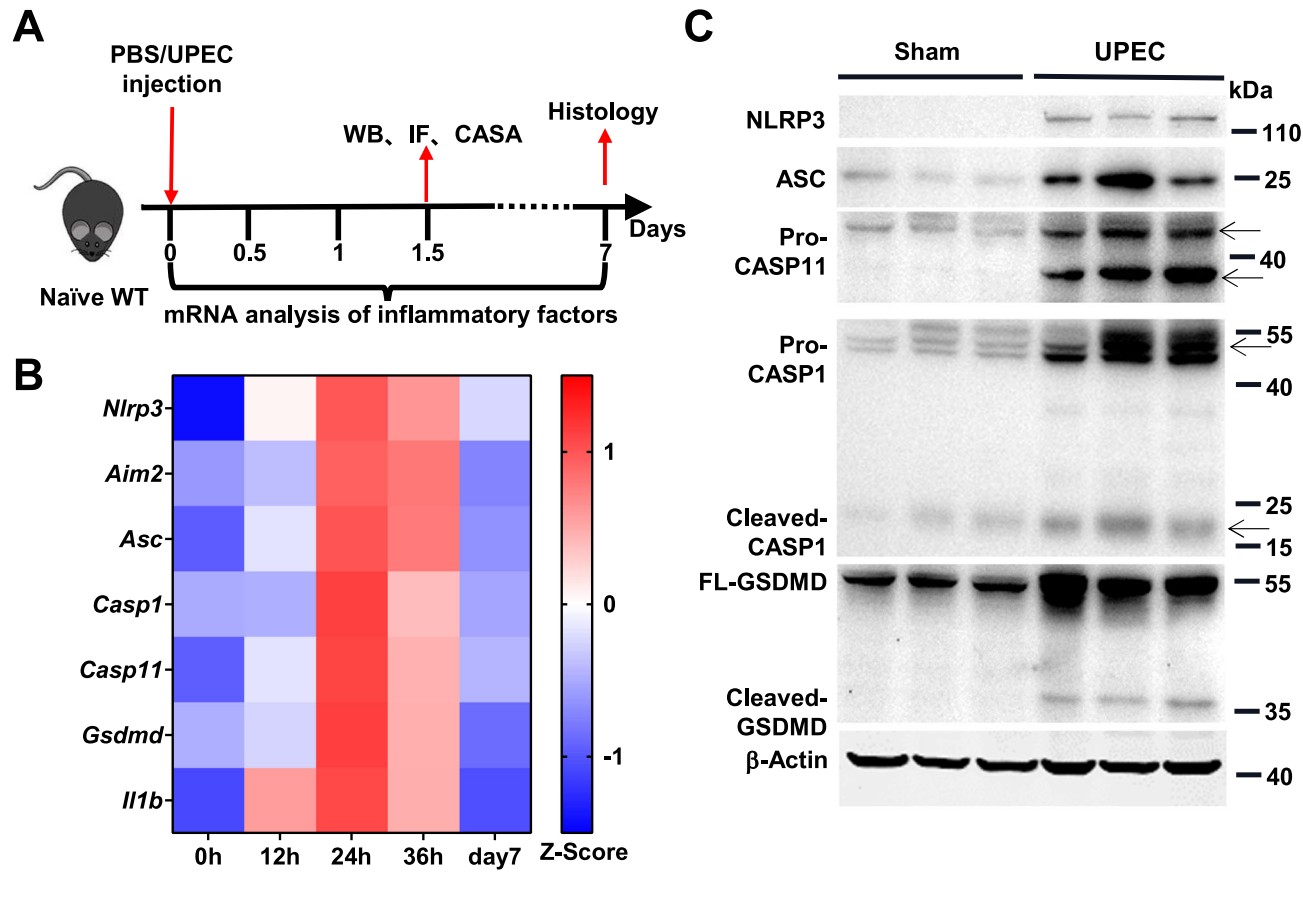

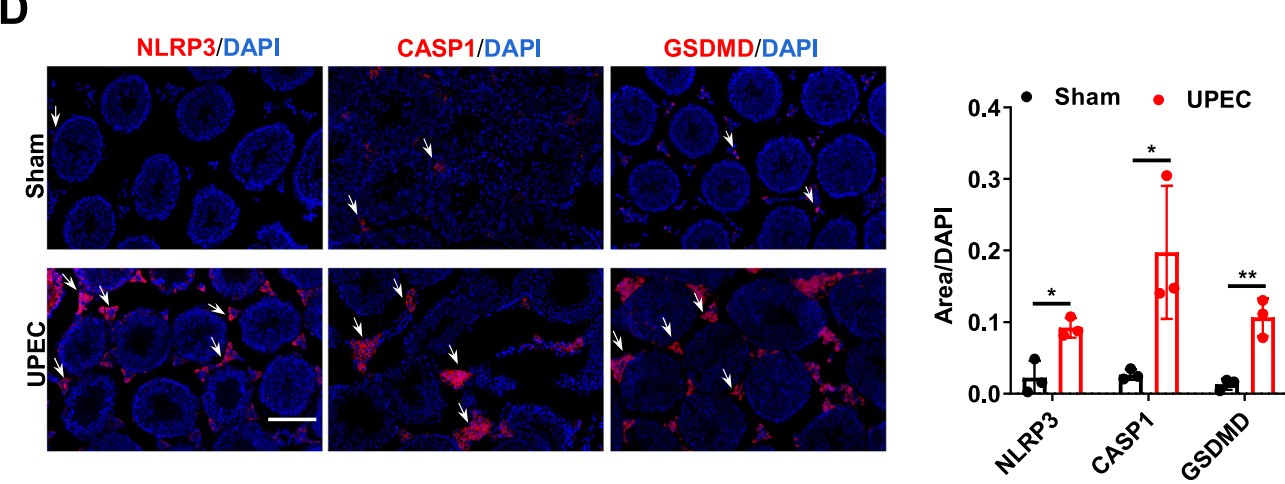

**Figure 1.  GSDMD was highly activated in the testes of UPEC-induced acute orchitic mice.**

(**A**) Schematic representation of the experiments performed using mice with uropathogenic *Escherichia coli* (UPEC)-induced acute orchitis. WB western blot, IF immunofluorescence, CASA computer-assisted sperm analysis. (**B**) Relative mRNA levels of *Nlrp3*, *Aim2*, *Asc*, *Casp1(Caspase-1)*, *Casp11(Caspase-11)*, *Gsdmd*, and *Il1b* in the testes of WT mice treated with UPEC ($1 \times 10^5$ CFU) for the indicated time points ($n = 5$ mice). (**C**) Immunoblots of NLRP3, ASC, pro-CASPASE-11, pro-CASPASE-1, full length (FL)-GSDMD, and β-actin (loading control) in testes from WT mice treated with UPEC ($1 \times 10^5$ CFU) for 36 h. The arrowheads indicate the target bands. (**D**) Immunofluorescent analysis of NLRP3, CASPASE-1, and GSDMD expression in the testes of WT mice injected with UPEC for 36 h. Scale bar, 100 μm. Data are presented as representative pictures (left) and quantified relative fluorescence intensity (right) ($n = 3$ mice per group). For quantifying immunofluorescence, the indicated fluorescence intensity of tissues was normalized to the intensity of DAPI. The white arrowheads indicate NLRP3-positive, CASPASE-1-positive, or GSDMD-positive cells. Data information: Data are pooled from three independent experiments for (**B**). Data are representative of three independent experiments for (**C**, **D**). Error bars show mean ± s.d. *$P < 0.05$, **$P < 0.01$. Two-tailed unpaired Student's *t* test was used unless otherwise stated. Source data are available online for this figure.

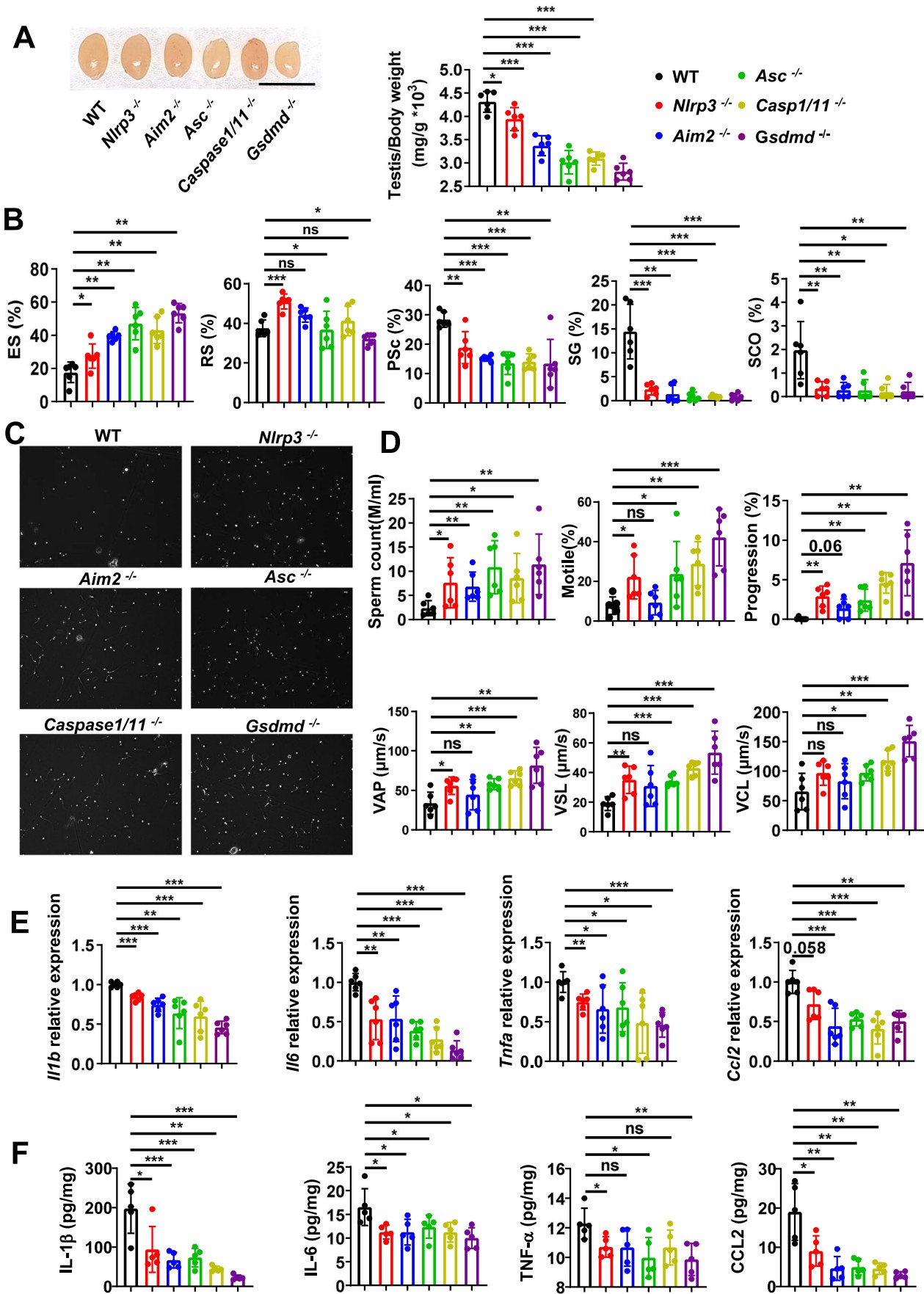

◄ **Figure 2. GSDMD deficiency significantly alleviated UPEC-induced acute orchitis.**

Age-matched male WT, *Nlrp3*⁻/⁻, *Aim2*⁻/⁻, *Asc*⁻/⁻, *Caspase-1/11*⁻/⁻, and *Gsdmd*⁻/⁻ mice were injected with UPEC (1 × 10⁵ CFU), and testicular tissues were collected for subsequent experiments. (A) Gross morphology of representative testes and relative weight of testes from UPEC-induced orchitic mice. Scale bars, 1 cm, (*n* = 6 mice per group). (B) Testicular histological changes in the testes in Fig. EV2H were calculated using the adaption of the classical Johnsen scoring system. ES elongated spermatids, RS round spermatids, PSc pachytene spermatocytes, SG spermatogonia, SCO Sertoli-cell only. (C) Representative images of CASA from the mice are shown in (A). (D) Sperm count, ratio of motile sperm and progressive sperm, and velocity of motile sperm (VAP, VSL, VCL) were detected using CASA (*n* = 6 mice per group). VAP mean average velocity, VCL curvilinear velocity, VSL straight linear velocity. (E) Relative mRNA levels of *Il1b*, *Il-6*, *Tnfa*, and *Ccl-2* in the testes of mice treated with UPEC for 24 h (*n* = 6 mice per group). (F) Enzyme-linked immunosorbent assay (ELISA) analysis of IL-1β, IL-6, TNF-α, and CCL-2 levels in the testes of mice treated with UPEC for 36 h (*n* = 6 mice per group). Data information: Data are pooled from three independent experiments for (A, C–F). Data are representative of three independent experiments for (B). Error bars show mean ± s.d. \**P* < 0.05, \*\**P* < 0.01, \*\*\**P* < 0.001, ns not significant. The two-tailed unpaired Student's *t* test was used unless otherwise stated. Mann–Whitney test for all comparisons of ES (%) in (B), for comparisons of RS (%) between WT and *Gsdmd*⁻/⁻ in (B), for comparisons of SG(%) between WT and *Aim2*⁻/⁻ in (B), and for comparisons of SCO(%) between WT and *Asc*⁻/⁻, WT and *Caspase-1/11*⁻/⁻, WT and *Gsdmd*⁻/⁻ in (B); Mann–Whitney test for all comparisons of progressive sperms (%) in (D); Mann–Whitney test for comparisons of relative expression of *Ccl-2* between WT and *Nlrp3*⁻/⁻, WT and *Gsdmd*⁻/⁻ in (E); Mann–Whitney test for comparisons of IL-1β level between WT and *Nlrp3*⁻/⁻ in (F). Source data are available online for this figure.

proinflammatory cytokines and chemokines in the testes of WT and *Gsdmd*⁻/⁻ orchitic mice using qRT-PCR and enzyme-linked immunosorbent assay (ELISA). GSDMD deficiency significantly inhibited the production of proinflammatory cytokines and chemokines, including IL-1β, IL-6, tumor necrosis factor (TNF)-α, CCL-2, and CXCL-10 (Figs. 2E,F and EV2I,J). In addition, *Nlrp3*⁻/⁻, *Aim2*⁻/⁻, *Asc*⁻/⁻, and *Caspase-1/11*⁻/⁻ mice showed similar but milder disease phenotypes than *Gsdmd*⁻/⁻ mice after UPEC infection (Figs. 2A–F and EV2H–J). Taken together, these data suggested that GSDMD is essential for the pathogenesis of orchitis.

## GSDMD in testicular macrophages promoted UPEC-induced acute orchitis

Expression analysis using a public database (Green et al, 2018) (Gene Expression Omnibus database with accession number GSE112393, 2018) showed that *Gsdmd* expression was higher in macrophages than in other testis cell types, including myoid cells, Leydig cells, meiotic spermatocytes (SCytes), post-meiotic haploid round spermatids (STids), Sertoli cells, endothelial cells, innate lymphoid cells, and ES (Fig. 3A). Furthermore, immunofluorescence analysis showed that GSDMD was upregulated and mainly co-localized with F4/80⁺ macrophages in the interstitial spaces after UPEC infection (Fig. 3B), indicating that GSDMD in testicular macrophages might be responsible for promoting UPEC-induced orchitis. To test this, we crossed *Gsdmd*ᶠˡ/ᶠˡ mice with *Cx3cr1*-cre mice to generate macrophage conditional knockout mice (*Gsdmd*ᶠˡ/ᶠˡ *Cx3cr1*-cre). We observed smaller testes and lower testis-to-body weight ratio in *Gsdmd*ᶠˡ/ᶠˡ*Cx3cr1*-cre mice than in *Cx3cr1*-cre mice after UPEC infection (Fig. 3C). Moreover, decrease in inflammatory cell infiltration and the number of tubules containing SCO, SG, PSc, and RS, and increase in the number of tubules containing ES in response to UPEC infection were observed in *Gsdmd*ᶠˡ/ᶠˡ *Cx3cr1*-cre mice (Figs. 3D and EV3A). The results of CASA showed that the sperm quality of *Gsdmd*ᶠˡ/ᶠˡ *Cx3cr1*-cre mice had improved (Fig. 3E,F). In addition, the expression of proinflammatory cytokines and chemokines markedly decreased in the testes of *Gsdmd*ᶠˡ/ᶠˡ *Cx3cr1*-cre mice (Fig. 4A,B). Next, we analyzed the immune cell population in the testes of mice treated with UPEC using flow cytometry (Fig. EV3B) and observed a significant decrease in the absolute numbers of immune cells (CD45⁺), macrophages (CD11b⁺F4/80⁺), and CD4⁺ and CD8⁺ T cells in the testes of *Gsdmd*ᶠˡ/ᶠˡ *Cx3cr1*-cre mice (Fig. 4C–E). In addition, the expression of MHC class II in macrophages, which present antigens

to T cells to facilitate T-cell antibacterial immunity (Harding et al, 2003), was markedly reduced in the testes of *Gsdmd*ᶠˡ/ᶠˡ *Cx3cr1*-cre mice (Fig. 4F,G). In agreement with these results, GSDMD deficiency in testicular macrophages resulted in downregulation of CD69 expression and TNF-α⁺ cells in CD4⁺ and CD8⁺ T cells (Fig. 4H–L). In addition, the absolute numbers of immune cells (CD45⁺), macrophages (CD11b⁺F4/80⁺), and CD4⁺ and CD8⁺ T cells did not differ between *Cx3cr1*-cre and *Gsdmd*ᶠˡ/ᶠˡ*Cx3cr1*-cre mice under physiological conditions (Fig. EV3C,D). Collectively, these data indicated that the specific loss of GSDMD in macrophages was sufficient to alleviate UPEC-induced acute orchitis and the immune response.

## GSDMD promoted antigen presentation and expression of inflammatory genes of testicular macrophages during UPEC-induced orchitis

To further elucidate the mechanism via which GSDMD in macrophages promoted orchitis, we performed RNA-sequencing (RNA-seq) of macrophages (CD45⁺CD11b⁺Ly6G⁻F4/80⁺) sorted from the testes of WT and *Gsdmd*⁻/⁻ mice treated with UPEC (Fig. EV4A). The expression profiles of macrophages of WT mice exhibited a gene-clustered architecture, which was distinct from that of macrophages of *Gsdmd*⁻/⁻ mice, as shown using principal component analysis (Fig. 5A). Venn diagram analysis revealed 1637 upregulated differentially expressed genes (DEGs) (fold change >1.2, *P* < 0.05) and 1888 downregulated DEGs in *Gsdmd*⁻/⁻ macrophages compared to that in WT macrophages (Fig. 5B). Reactome analysis further confirmed that the downregulated pathways in *Gsdmd*⁻/⁻ macrophages included the immune system, interleukin-1 signaling, inflammatory response, antigen processing-cross-presentation, and pyroptosis (Fig. 5C), which is in agreement with the results of a previous report showing that pyroptosis was impaired in the absence of GSDMD (Shi et al, 2015). Gene Ontology biological process (GO-BP) analysis indicated that the downregulated genes in *Gsdmd*⁻/⁻ macrophages were related to inflammatory responses (Fig. 5C). Gene Set Enrichment Analysis (GSEA) of GO and GO-BP gene sets further revealed that the inflammatory response and antigen processing presentation were significantly downregulated in *Gsdmd*⁻/⁻ macrophages (Fig. 5D). Heatmap and qRT-PCR analyses also showed significant reduction in the expression of genes associated with inflammatory response and antigen presentation, such as *H2-Dmb1*, *H2-Dma*, *H2-Ab1*, *Tap2*, *Psmb3*, *Tnfa*, *Il-6*, *Il1a*, and

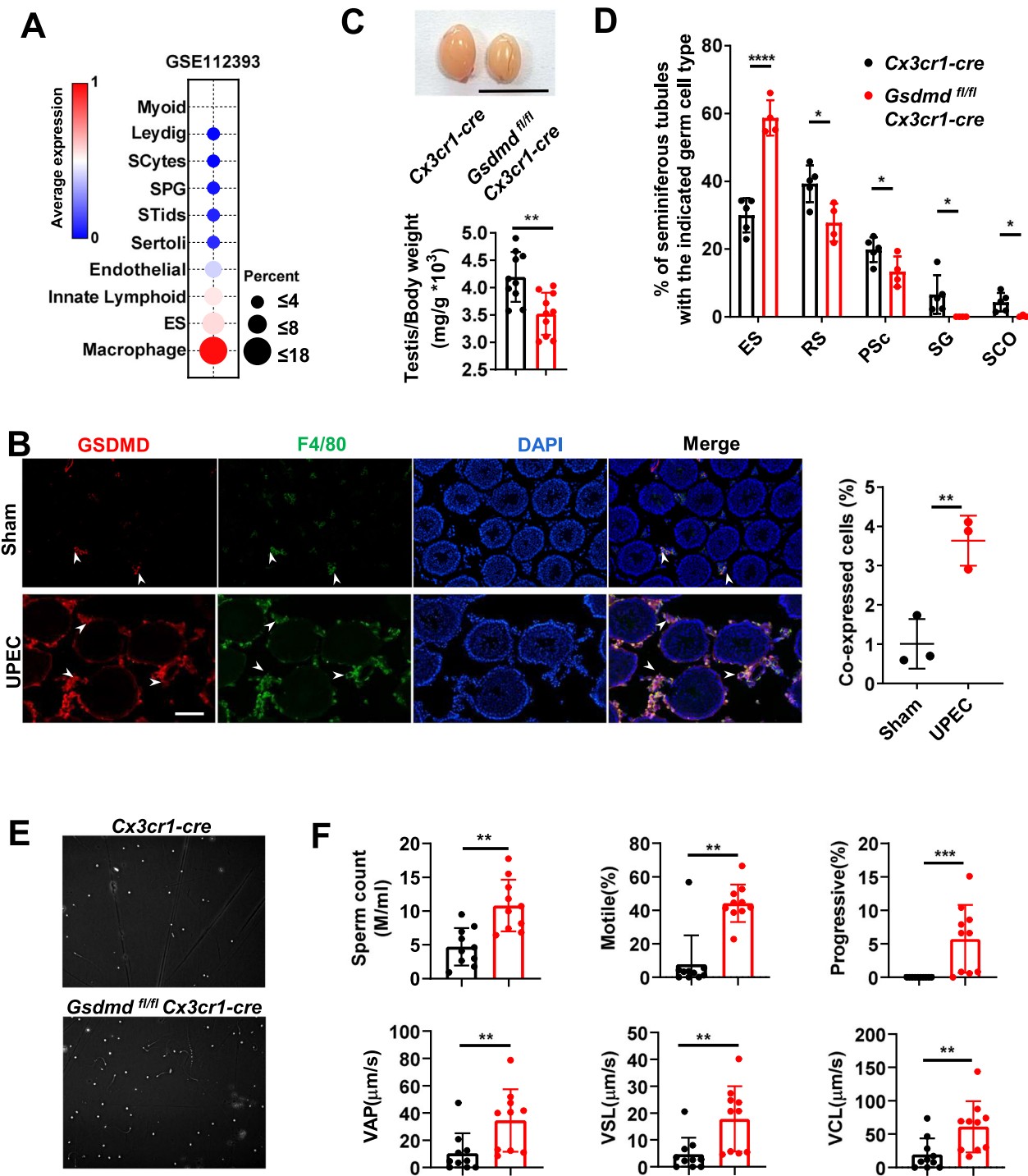

**Figure 3. GSDMD deficiency in macrophages improved spermatogenesis during UPEC-induced acute orchitis.**

(A) *Gsdmd* expression in different types of cells in mouse testis, including myoid cells, Leydig cells, meiotic spermatocytes (SCytes), spermatogonia (SPG), Sertoli, post-meiotic haploid round spermatids (STids), endothelial cells, innate lymphoid cells, elongating spermatids (ES), and macrophages, obtained from public databases (GSE112393). The percent values are shown in the bubble chart. (B) Immunofluorescence staining for GSDMD (red) and F4/80 (green) in testicular tissue of WT mice injected with UPEC for 36 h. The merge of GSDMD with F4/80 is indicated by white arrowheads. Scale bars, 100 μm. Data are presented as representative images (left) and quantified proportion of co-expressing cells (right) (n = 3 mice per group). (C) Gross morphology of representative testes and relative weight of testes from *Gsdmd*[fl/fl]*Cx3cr1*-cre and *Cx3cr1*-cre mice treated with UPEC for 36 h. Scale bars, 1 cm (n = 10 mice per group). (D) Testicular histological changes calculated using the adaption of the classical Johnsen scoring system (*Cx3cr1*-cre, n = 5; *Gsdmd*[fl/fl]*Cx3cr1*-cre, n = 4). (E) Representative picture of the mice in (C) obtained using CASA. (F) Sperm count, the ratio of motile and progressive sperms, velocity of motile sperms (VAP, VSL, VCL) in (C) were detected using CASA (n = 10 mice per group). Data information: Data are pooled from three independent experiments for (C–F). Data are representative of three independent experiments for (B). Error bars show mean ± s.d. *P < 0.05, **P < 0.01, ***P < 0.001, ****P <0.0001. Two-tailed unpaired Student's t test was used unless otherwise stated. Mann–Whitney test for comparisons of SG and SCO in (D). Source data are available online for this figure.

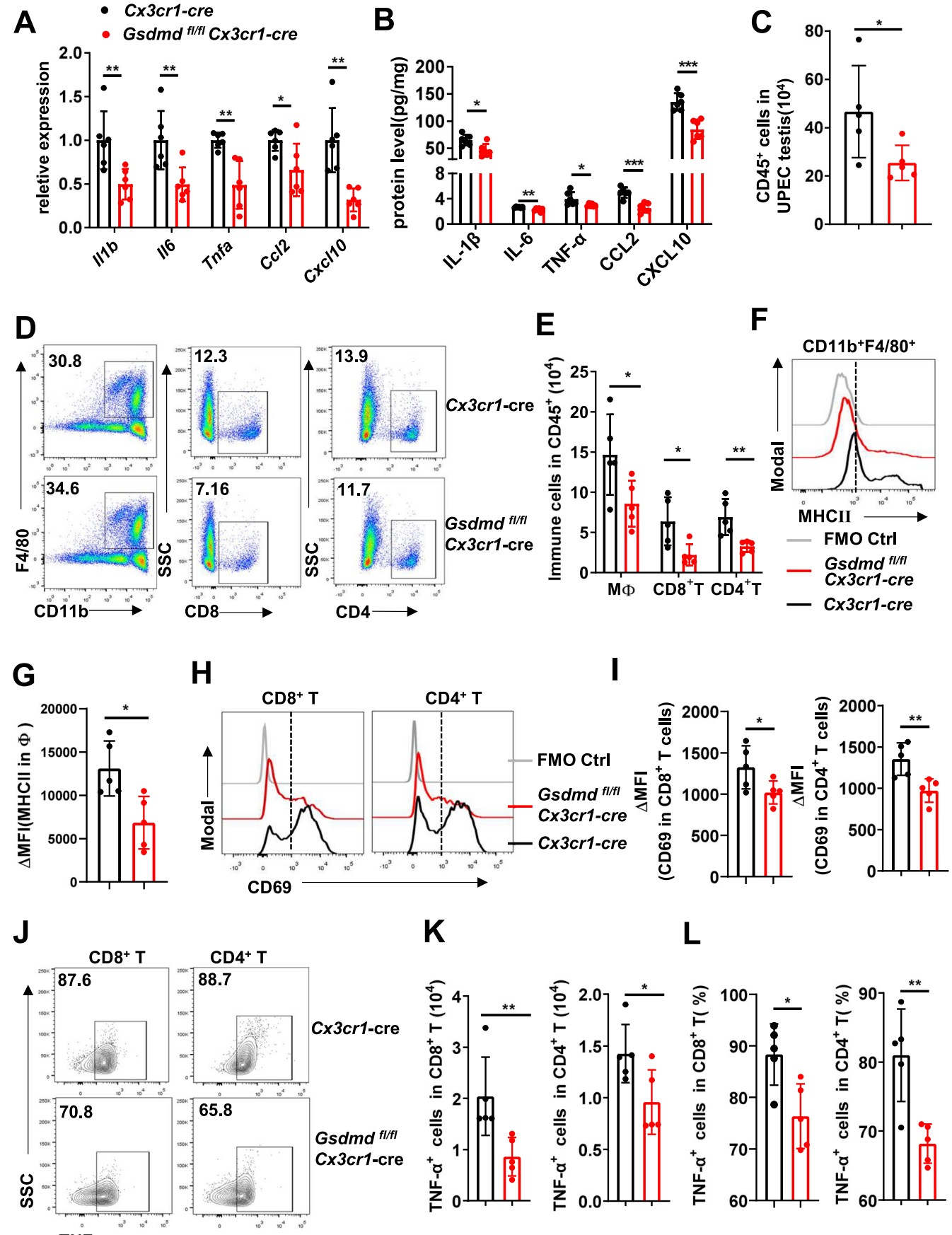

◄ **Figure 4. GSDMD deficiency in macrophages alleviated testicular inflammation during UPEC-induced acute orchitis.**

(A) Relative mRNA levels of *Il1b*, *Il-6*, *Tnfa*, *Ccl-2*, and *Cxcl-10* in the testes from *Gsdmd*^fl/fl^*Cx3cr1*-cre and *Cx3cr1*-cre mice treated with UPEC for 24 h (*n* = 6 mice per group). (B) ELISA analysis of IL-1β, IL-6, TNF-α, CCL-2, and CXCL-10 levels in the testes of mice treated with UPEC for 36 h (*n* = 6 mice per group). (C) The number of total immune cells in the testes of *Gsdmd*^fl/fl^*Cx3cr1*-cre and *Cx3cr1*-cre mice treated with UPEC for 36 h (*n* = 5 mice per group). (D, E) Representative flow cytometry plots (D) and the quantified absolute number (E) of the macrophages (CD11b⁺F4/80⁺), CD8⁺ T cells, CD4⁺ T cells in the testes of *Gsdmd*^fl/fl^*Cx3cr1*-cre and *Cx3cr1*-cre mice treated with UPEC for 36 h (*n* = 5 mice per group). (F, G) Representative flow cytometry histogram (F) and mean fluorescence intensity (MFI) (G) of MHC class II expression (G) in macrophages (*n* = 5 mice per group). (H, I) Representative flow cytometry histogram (H) and MFI (I) of CD69 expression in CD8⁺ and CD4⁺ T cells (*n* = 5 mice per group). (J–L) Representative flow cytometry plots (J), and the proportion (K), and quantified absolute number (L) of TNF-α⁺ cells in CD4⁺ and CD8⁺ T cells from *Gsdmd*^fl/fl^ *Cx3cr1*-cre and *Cx3cr1*-cre mice treated with UPEC for 36 h (*n* = 5 mice per group). Data information: Data are pooled from three independent experiments for (C–L). Data are representative of three independent experiments for (D, F, H, J). Error bars show mean ± s.d. *$P < 0.05$, **$P < 0.01$, ***$P < 0.001$. Two-tailed unpaired Student's *t* test was used unless otherwise stated. Mann–Whitney test for comparisons of CD8⁺ T cells in (E); Mann–Whitney test for comparisons of TNF-α⁺ cells in CD8⁺ T in (K). Source data are available online for this figure.

*Tnfaip3* (Fig. 5E,F). Taken together, these data indicated that GSDMD deficiency in testicular macrophages negatively regulated antigen presentation and inflammatory gene expression in UPEC-induced acute orchitis.

## GSDMD boosted T-cell response by enhancing antigen presentation of macrophages

Next, to investigate whether GSDMD in macrophages regulated antigen presentation and boosted the T-cell response, we co-cultured naive OT-1 CD8⁺ T cells with WT or *Gsdmd*⁻/⁻ bone marrow-derived macrophages (BMDMs) stimulated by lipopolysaccharide (LPS) plus nigericin and the chicken ovalbumin (OVA) peptide (Fig. 6A). The expression of CD86 in *Gsdmd*⁻/⁻ BMDMs was significantly lower than that in WT BMDMs (Fig. 6B), suggesting GSDMD deficiency resulted in impaired antigen presentation in macrophages. Moreover, when compared with co-culture with WT BMDMs, co-culture with *Gsdmd*⁻/⁻ BMDMs decreased the frequency of activated antigen-specific CD8⁺ T cells (CD8⁺ OVA⁺ CD44⁺) (Fig. 6C) and the proportion of interferon (IFN)-γ and TNF-α secreting antigen-specific CD8⁺ T cells (Fig. 6D,E). Altogether, these data indicated that GSDMD deficiency in macrophages impaired the T-cell response by negatively regulating antigen presentation, thereby controlling orchitis.

## Administration of a GSDMD inhibitor attenuated UPEC-induced acute orchitis

Dimethyl fumarate (DMF) suppresses the cleavage and activation of GSDMD (Humphries et al, 2020). To further confirm the role of GSDMD-mediated pyroptosis in orchitis and the interventional effect of targeted inhibition of GSDMD activation in UPEC-induced orchitis, we administered DMF to mice with UPEC-induced orchitis (Fig. 7A). We found that DMF treatment protected against UPEC infection and considerably attenuated the size of the testes and the testis–body weight ratio (Fig. 7B). H&E staining consistently showed decrease in inflammatory cell infiltration, increase in the number of tubules containing ES, and decreased tubules containing SCO, SG, and RS after DMF treatment (Figs. 7C and EV4D). DMF administration also improved sperm quality (Fig. 7D,E). In addition, DMF therapy markedly inhibited the expression of proinflammatory cytokines and chemokines (Fig. 7F,G). Overall, these data suggested that the inhibition of GSDMD-mediated pyroptosis can control UPEC-induced orchitis and indicated that GSDMD might be a potential therapeutic target for UPEC-induced acute orchitis.

## GSDMD in macrophages aggravated chronic autoimmune orchitis during EAO development

These results clearly showed that testicular macrophage-intrinsic GSDMD promoted antigen presentation, thereby inducing T-cell activation during UPEC-induced acute orchitis. To further investigate whether GSDMD in macrophages promoted the development of chronic autoimmune orchitis, we induced experimental autoimmune orchitis (EAO) in *Gsdmd*^fl/fl^*Cx3cr1*-cre and *Cx3cr1*-cre mice (Fig. 8A). The size of the testis and testis–body weight ratio were larger in *Gsdmd*^fl/fl^*Cx3cr1*-cre mice than in *Cx3cr1*-cre mice (Fig. 8B), indicative of improved testicular degeneration in GSDMD-deficient mice. Moreover, decreased tubules containing PSc and RS, and increased tubules containing ES were observed in *Gsdmd*^fl/fl^ *Cx3cr1*-cre mice using H&E staining (Figs. 8C and EV4E). The results of CASA showed that sperm quality have improved in *Gsdmd*^fl/fl^ *Cx3cr1*-cre mice (Fig. 8D,E). Consistently, flow cytometry analysis showed that the numbers of immune cells, testicular macrophages, CD8⁺ T cells, and CD4⁺ T cells in *Gsdmd*^fl/fl^ *Cx3cr1*-cre mice were markedly lower than those in *Cx3cr1*-cre mice (Fig. 8F,G). The expression of MHC class II molecules decreased significantly in the testicular macrophages of *Gsdmd*^fl/fl^ *Cx3cr1*-cre mice (Fig. 8H). Consistent with these results, GSDMD deficiency in testicular macrophages decreased the proportion of IFN-γ⁺ CD4⁺ and CD8⁺ T cells (Fig. 8I). These findings suggested that GSDMD deficiency in macrophages impaired the development of chronic orchitis.

# Discussion

Orchitis and epididymitis, common intrascrotal inflammatory diseases, are significant pathogenic factors that contribute to male infertility. UPEC is the most common pathogen isolated from patients with epididymitis and orchitis. Bacterial epididymitis and orchitis result in subinfertility or infertility in ~40% patients (Klein et al, 2020). The pathological effects of epididymitis on epididymal tissues have been well-documented in both human males and animal models (Klein et al, 2020; Pilatz et al, 2013). However, the details of the testicular consequences of bacterial infections are poorly understood. In this study, we injected a small number of bacteria into the testes rather than the vas deferens to assess the direct role of UPEC infection in male infertility. Thereafter, we comprehensively examined the effects of GSDMD on the testes in the context of UPEC-induced orchitis. We found that GSDMD deficiency in macrophages mitigated the inflammatory responses

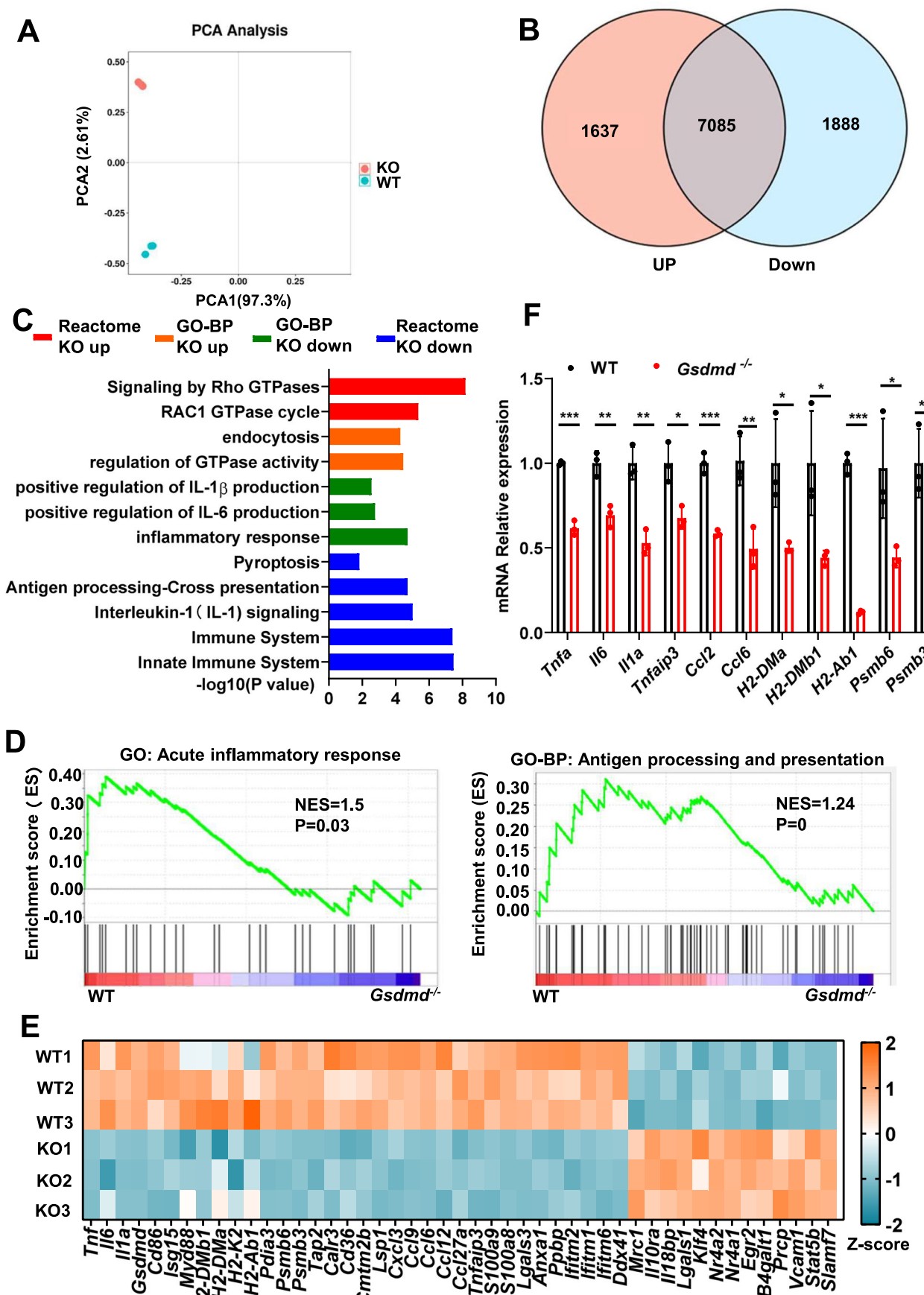

◄

**Figure 5. GSDMD enhanced antigen presentation and inflammatory gene expression in testicular macrophages.**

WT and *Gsdmd*⁻/⁻ mice were injected with UPEC for 36 h, and testicular macrophages were sorted for RNA-seq analysis. (A) Principal component analysis showing transcriptionally clustered architecture in testicular macrophages from WT and *Gsdmd*⁻/⁻ mice. (B) Venn diagram displaying the total number of genes detected, including upregulated differentially expressed genes (DEGs), downregulated DEGs, and not differentially expressed genes, in macrophages from WT and *Gsdmd*⁻/⁻ mice. DEGs were identified using fold change >1.2, P value < 0.05. (C) Gene Ontology biological process (GO-BP) and Reactome pathway analyses of the most significantly enriched signaling pathways using DEGs in testicular macrophages of WT and *Gsdmd*⁻/⁻ mice were performed using http://david.ncifcrf.gov, and graphs were generated using GraphPad Prism 8.0. (D) Gene Set Enrichment Analysis (GSEA) of the total genes associated with "antigen processing-cross presentation" and "inflammatory response" in macrophages using GSEA v3.0 based on the GO or GO-BP gene sets. (E) Heatmap was generated using DEGs (fold change >1.2, P value < 0.05) involved in the most significantly enriched signaling pathways obtained from RNA-seq analysis of testicular macrophages in Fig. 4B. The graphs were generated using GraphPad Prism 8.0. (F) qRT-PCR analysis of the indicated genes of testicular macrophages from WT and *Gsdmd*⁻/⁻ mice (n = 3). Data information: Data are pooled from three independent experiments for (F). Error bars show mean ± s.d. *P < 0.05, **P < 0.01, ***P < 0.001. Hypergeometric test for (C), Permutation test for (D), and two-tailed unpaired Student's *t* test for (F). Source data are available online for this figure.

and conferred protection against UEPC infection. In addition, we demonstrated the macrophage-intrinsic role of GSDMD in promoting EAO development.

In the past few decades, accumulating evidence has indicated the importance of innate immunological factors in the establishment of UPEC infections which trigger inflammatory responses, such as TLR4 (Hilbert, 2011), Myd88 (Bhushan et al, 2008), and NLRP3 (Li et al, 2019b). Consistent with this, we found that GSDMD, the key executioner of pyroptosis downstream of inflammasomes, contributes to UPEC-induced orchitis. Moreover, we found that GSDMD-mediated pyroptosis in testicular macrophages exacerbated the inflammatory response and impaired spermatogenesis and sperm quality following UPEC infection. In agreement with our results, UPEC has been reported to trigger pyroptosis in macrophages in vitro (Schaale et al, 2016; Zhang et al, 2022), even in bladder epithelial cells (Joshi et al, 2022; Lindblad et al, 2022), and renal epithelial cells (Gu et al, 2021). In addition, the flow cytometry plot of macrophages in the testes showed only one population under homeostatic conditions but two distinct groups after UPEC infection, one of which showed both F4/80^high and F4/80^low in CD11b^high cells. The expression of F4/80 by infiltrated monocytes was lower than that in tissue macrophages. Therefore, we speculated that a large proportion of the CD11b^high population may include infiltrated monocytes and macrophages, which may represent macrophages functionally distinct from the F4/80^high CD11b^low population.

GSDMD and ASC deficiency decreased sperm count and impaired sperm quality under physiological conditions but did not alter the expression of inflammatory factors. Data in public databases and previous studies have shown that *Gsdmd* and *Asc* are expressed in Sertoli cells (Green et al, 2018; Walenta et al, 2018). Thus, we speculated that GSDMD and ASC may regulate the function of Sertoli cells, thereby supporting spermatogenesis. The mechanisms underlying these regulatory effects should be investigated in future.

Notably, our RNA-seq analysis revealed that immune system, inflammatory response, and antigen processing and presentation were the major immune pathways downregulated in macrophages isolated from orchitic *Gsdmd*⁻/⁻ mice. Consistently, the heatmap and results of qRT-PCR showed that GSDMD deficiency considerably impaired the expression of genes involved in antigen presentation, including *H2-Dmb1*, *H2-Dma*, *H2-Ab1*, *Tap2*, *Cd86*, *Ifitm2*, *Isg15*, *Ifitm1*, and *S100a8*. Studies have shown that S00A8 can promote the expression of MHC and co-stimulatory molecules, thereby enhancing the antigen presentation capacity of antigen-presenting cells during arthritis and dermatitis (Kessel et al, 2013; Petersen et al, 2013). The importance of S100A8 in UPEC bladder and kidney infections has been reported (Spencer et al, 2015; Subashchandrabose and Mobley, 2015). However,

whether the role of GSDMD in regulating antigen presentation can be attributed to the promotion of S100A8 expression and how GSDMD enhances S100A8 expression during orchitis requires further investigation. In addition, our RNA-seq analysis revealed that GSDMD modulates antigen processing and cross-presentation of antigens. Therefore, by co-culturing OT-1 CD8⁺ T cells with macrophages, we demonstrated that *Gsdmd* is a cell-autonomous gene in macrophages that boosts the CD8⁺ T-cell immune response by enhancing the antigen presentation capacity. Consistent with this, CD8⁺ T cells are the dominant subset of T cells in the testis (Jing et al, 2021). Overall, our findings highlight GSDMD as a potential therapeutic target for the treatment of UPEC-induced orchitis. Furthermore, we directly evaluated the therapeutic effect of GSDMD inhibition on UPEC-induced orchitis in a mouse model by administering DMF, an inhibitor of GSDMD activation that has been approved for the treatment of multiple sclerosis (Narapureddy and Dubey, 2019). DMF treatment significantly ameliorated UPEC-induced orchitis in a mouse model.

UPEC is a common urinary tract pathogen that causes orchitis and is usually treated with antibiotics. However, the increase in antibiotic resistance among pathogens is a global cause for concern. Our findings provide a novel therapy for bacterial and chronic autoimmune orchitis. Increasing evidence suggests that orchitis can be caused by *Neisseria gonorrhoeae* and even viral infections (Fijak et al, 2018). However, whether GSDMD regulates orchitis pathogenesis in response to other bacterial and viral infections remains unclear, which also warrants further investigation.

UPEC initially affects the epididymis by ascending from the penile urethra, but ~60% of the infected patients experience orchitis (Klein et al, 2020). Furthermore, fertility impairment following epididymo-orchitis can originate from the direct exposure of testicular germ cells to pathogens, toxins, or pathogen-induced immune response (Klein et al, 2020). Reports have shown that UPEC ascended to the testis at day 7 post inoculation and spread into the lumen of the vas deferens (Michel et al, 2016). Therefore, despite the limitations in injecting UPEC into the testes, investigating the direct effects of UPEC on the testes is important.

In summary, we have demonstrated that GSDMD is activated during UPEC-induced orchitis. GSDMD deficiency in macrophages limits inflammation and impairs spermatogenesis in acute and chronic orchitis. Mechanistically, GSDMD ablation in macrophages impairs CD8⁺ T-cell responses by preventing antigen presentation. Administration of GSDMD inhibitors conferred protection against UPEC infection-induced orchitis (Fig. EV5). These results suggested that GSDMD is a potential therapeutic target for the treatment of UPEC-induced acute orchitis and chronic autoimmune orchitis.

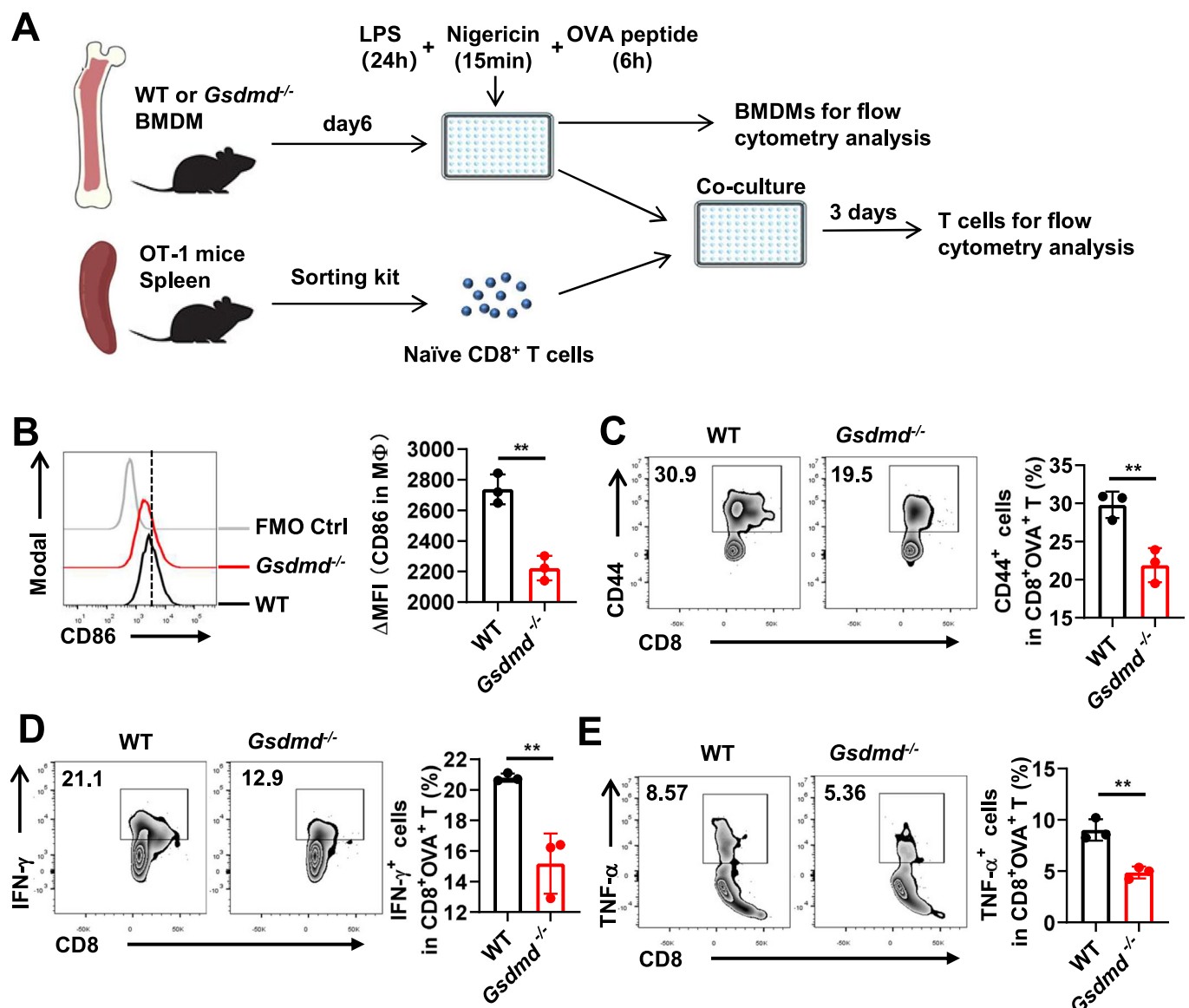

**Figure 6. GSDMD boosted T-cell response by enhancing antigen presentation of macrophages.**

Age-matched male WT and Gsdmd$^{-/-}$ mice were used for bone marrow-derived macrophage (BMDM) culture, and OT-1 mice were used to sort naive CD8$^+$ T cells. (**A**) Schematic representation of the co-culture experiments in (B–E). Briefly, BMDMs from WT and Gsdmd$^{-/-}$ were pre-stimulated with LPS (lipopolysaccharide, 100 ng/ml), nigericin (5 mM), and OVA (ovalbumin, 100 ng/ml) for indicated time points, and then BMDMs were washed three times with PBS, and co-cultured with naive CD8$^+$ T cells from OT-1 mice. (**B**) Flow cytometric analysis of CD86 expression in macrophages, and data are presented as a representative histogram (left) and summary graph of quantified MFI (right). ($n = 3$). (**C**) Flow cytometric analysis of CD44$^+$ cells in CD8$^+$ T cells, and data are presented as representative plots (left) and summary graph of quantified percentages (right) ($n = 3$). (**D, E**) Flow cytometric analysis of IFN-γ$^+$ (**D**) and TNF-α$^+$ cells (**E**) in CD8$^+$ T cells, and data are presented as representative plots (left) and summary graph of quantified percentages (right) ($n = 3$). Data information: Data are representative of two independent experiments. Error bars show mean ± s.d. **$P < 0.01$. Two-tailed unpaired Student's $t$ test was used unless otherwise stated. Source data are available online for this figure.

## Methods

### Mice

Nlrp3$^{-/-}$, Aim2$^{-/-}$, Asc$^{-/-}$, and Caspase-1/11$^{-/-}$ mice were a gift from Dr. V Dixit (Genentech, South San Francisco, CA, USA), and their generation and validation have been described previously (Kayagaki et al, 2011). Gsdmd$^{-/-}$ mice were provided by Dr. Feng Shao (National Institute of Biological Sciences, Beijing, China), and have been described previously (Shi et al, 2015). Cx3cr1-cre mice were donated by Dr. Jiawei Zhou (Institute of Neuroscience, Chinese Academy of Sciences, Shanghai, China), as described previously (Yona et al, 2013). Gsdmd$^{fl/fl}$ mice were generated as described previously (Li et al, 2019a). Gsdmd$^{fl/fl}$ mice were crossed with Cx3cr1-cre mice to generate macrophage-conditioned GSDMD knockout mice. All mice were bred and housed in a barrier facility and under controlled environmental conditions (12 h light/12 h darkness, temperature range of 20–23 °C, and relative humidity maintained at 50–60%), and all animal

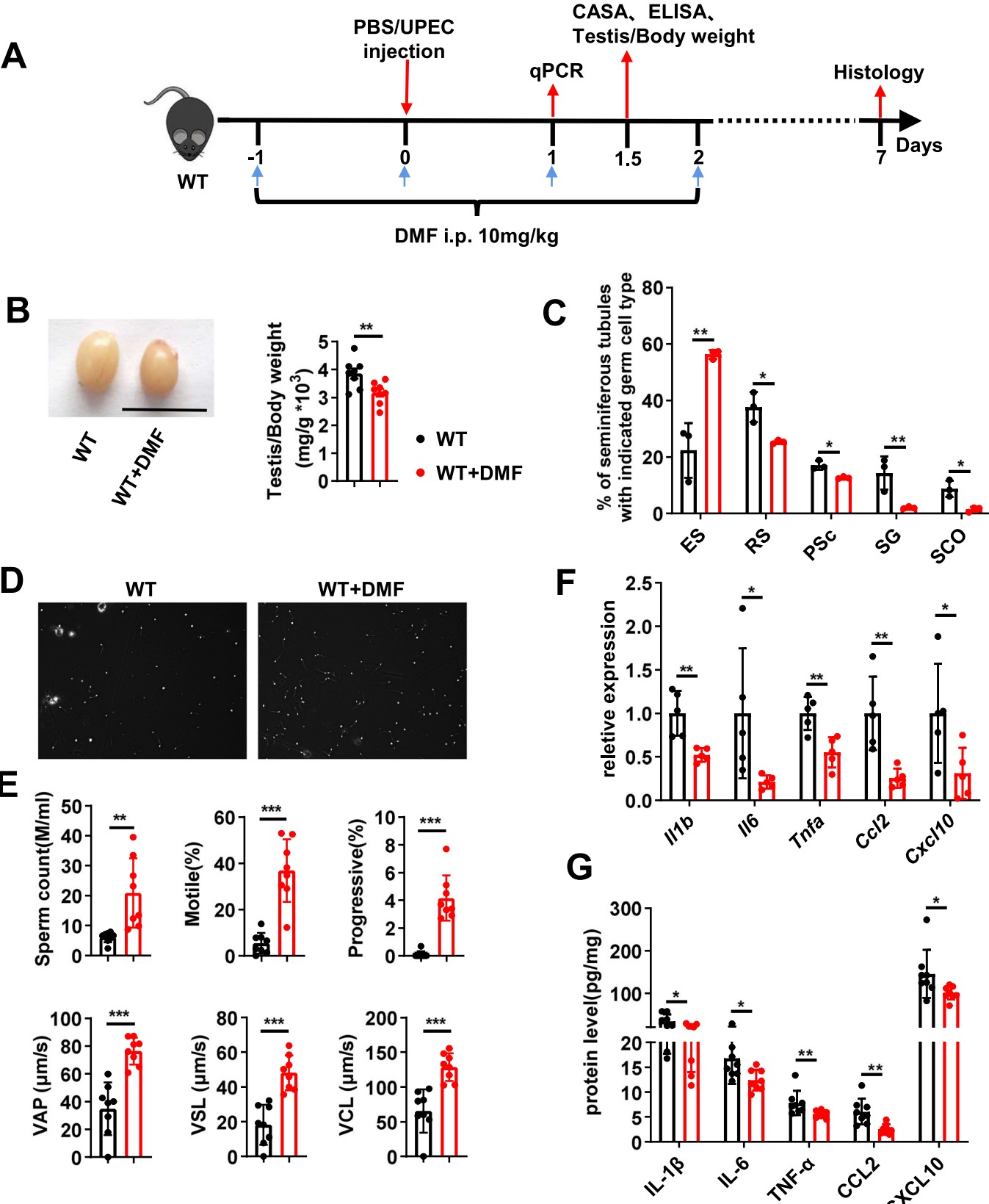

◀ **Figure 7.  Inhibitor targeting GSDMD attenuated UPEC-induced acute orchitis.**

Age-matched male WT mice were randomly divided into two groups, one of which was intraperitoneally (i.p.) injected with dimethyl fumarate (DMF). After 1 day, both the groups were treated with UPEC. Testicular tissues were collected at different time points for subsequent experiments. (A) Schematic representation of DMF medication pattern during UPEC infection in (B–F). (B) Testicular morphology and relative weight of testis. Scale bars, 1 cm ($n = 8$ mice per group). (C) Testicular histological changes of testes in Fig. EV4D were calculated using the adaption of the classical Johnsen scoring system ($n = 3$ mice per group). (D) Sperm count, ratio of motile and progressive sperm, and velocity of motile sperm (VAP, VSL, and VCL) of indicated mice were detected using CASA ($n = 8$ mice per group). (E) Relative mRNA levels of *Il1b*, *Il-6*, *Tnfa*, *Ccl-2*, and *Cxcl-10* in the testes of indicated mice ($n = 5$ mice per group). (F) IL-1β, IL-6, TNF-α, CCL-2, and CXCL-10 levels in the testes of mice from different groups measured using ELISA ($n = 8$ mice per group). Data information: Data are pooled from three independent experiments. Error bars show mean ± s.d. *$P < 0.05$, **$P < 0.01$, ***$P < 0.001$, ns not significant. Two-tailed unpaired Student's *t* test was used unless otherwise stated. Mann–Whitney test for comparisons of Progressive (%) in (E), and Mann–Whitney test for comparisons of levels of IL-6, TNF-α, and CXCL-10 in (G). Source data are available online for this figure.

experiments were performed in compliance with the Ethical Procedures for Experimental Animal Welfare of Nanjing Medical University and Nanjing University of Traditional Chinese Medicine (approval number, 1705038).

## Bacterial culture and animal treatment

The UPEC strain, CFT073, was a gift from Dr. Yongning Lu (Zhongshan Hospital, Fudan University, China) and was propagated as described previously (Lu et al, 2021). The UPEC strain was propagated overnight on agar plates, then a single clone of UPEC was isolated and inoculated in lysogeny broth (LB) agar medium until it grew to an exponential phase ($OD_{600} = 0.5-1.0$). The cells were centrifuged at $4500 \times g$ for 8 min at 4 °C, and the pellets were washed once with pre-cooled phosphate-buffered saline (PBS) and then stored in Dulbecco's modified Eagle's medium (DMEM). After gradient dilution, the cells were spread on the plate and the colonies were counted after culturing overnight at 37 °C. The UPEC concentration was calculated according to standard growth curves.

Experimental orchitis induced by UPEC in male mice was modified on the method described previously (Klein et al, 2020). Briefly, the mice were anesthetized by injecting tribromoethanol intraperitoneally (T48402, 180 mg/kg; Sigma), and the testes and vas deferens were exposed. Approximately $1 \times 10^5$ colony-forming units (CFU) of bacteria in 5 μl PBS were injected into the upper pole of the testis instead of the lumen between the ligation site and the cauda epididymis, as described by Klein et al, and 5 μl PBS was injected into the mice of the sham group. The vas deferens were then ligated by tying a knot using surgical sutures to prevent the spread of the infection. After the wound was sutured, the mice were kept warm until they recovered, and the wound sutures were disinfected twice daily.

## Induction of experimental autoimmune orchitis (EAO)

Age-matched male *Gsdmd*^fl/fl^*Cx3cr1*-cre and *Cx3cr1*-cre mice were used to induce EAO, as described previously (Nicolas et al, 2017). First, the testicular homogenate (TH) was obtained from decapsulated testes of adult syngeneic mice and homogenized in an equal volume of sterile PBS. Second, TH was mixed with incomplete Freund's adjuvant (IFA; Sigma-Aldrich, F5506) containing 1 mg/ml *Mycobacterium tuberculosis* (BD, 231141) in 1:1 ratio. Finally, the mice were immunized thrice every 14 days with the mixture, followed by an intraperitoneal injection of 100 ng pertussis toxin (List Biological Laboratories, lot#180) in 100 μl PBS. Each animal was immunized dorsally at four sites near the popliteal lymph nodes in a total volume of 200 μl. The testes were collected for further analysis 50 days after the first immunization.

## Histological analysis

For histological analysis, the isolated testes were fixed with modified Davidson's fixative (MDF; 1.2% formaldehyde, 15% ethyl alcohol, and 5% glacial acetic acid) at room temperature for 12 h. The testicular tissues were then cut in half, fixed with fresh fixative solution for 12 h, and mounted in paraffin according to standard protocols. The tissue sections were stained with H&E, and histology was scored using an adaptation of the classical Johnsen scoring system as described previously (Klein et al, 2020). Briefly, slides were prepared after cutting the tissue at the largest cross-sectional position of the testis. In each mouse, one microscope slide was evaluated if 150−250 seminiferous tubules per section were visible at ×40 magnification. Microscopy slides that met the criteria were assessed by evaluating all seminiferous tubules under an optical microscope (magnification, ×100). Approximately 10−15 fields of view in one section contained 150−250 seminiferous tubules; the most advanced germ cell type detected in each tubule cross-section was recorded as the representative germ cell stage, and the percentage of tubule cross-sections showing the respective germ cell stage was documented. For example, a single slide included 200 seminiferous tubules, of which only 10 tubules containing Sertoli cells were recorded. The frequency of tubules containing Sertoli cells was only 5%.

## Computer-assisted sperm analysis (CASA)

CASA (Hamilton Thorne IVOS) was used to evaluate the motility parameters of the epididymal semen. The epididymis was removed and incubated in preheated sterile PBS for 10 min, following which 10 μl of the sperm suspension was assessed for motility characteristics at 37 °C. At least five fields of view were selected for each sample. Six parameters were evaluated: sperm count (M/ml), motile cell count (%), progressive cell count (%), mean average velocity (VAP, mm/s), curvilinear velocity (VCL, mm/s), and straight linear velocity (VSL, mm/s).

## Flow cytometry

Testicular tissue was prepared as single-cell suspensions, as described previously (Wang et al, 2021). Briefly, the testes were excised and digested with 10 U/mL DNase I (Sigma, D4257) and 0.5 mg/mL collagenase I (Gibco, 2441928) in RPMI 1640 at 37 °C under agitation conditions (200 rpm) for 30 min. Digested tissues were filtered through a 70-mm filter to obtain single-cell suspensions. Subsequently, cell suspensions were centrifuged over

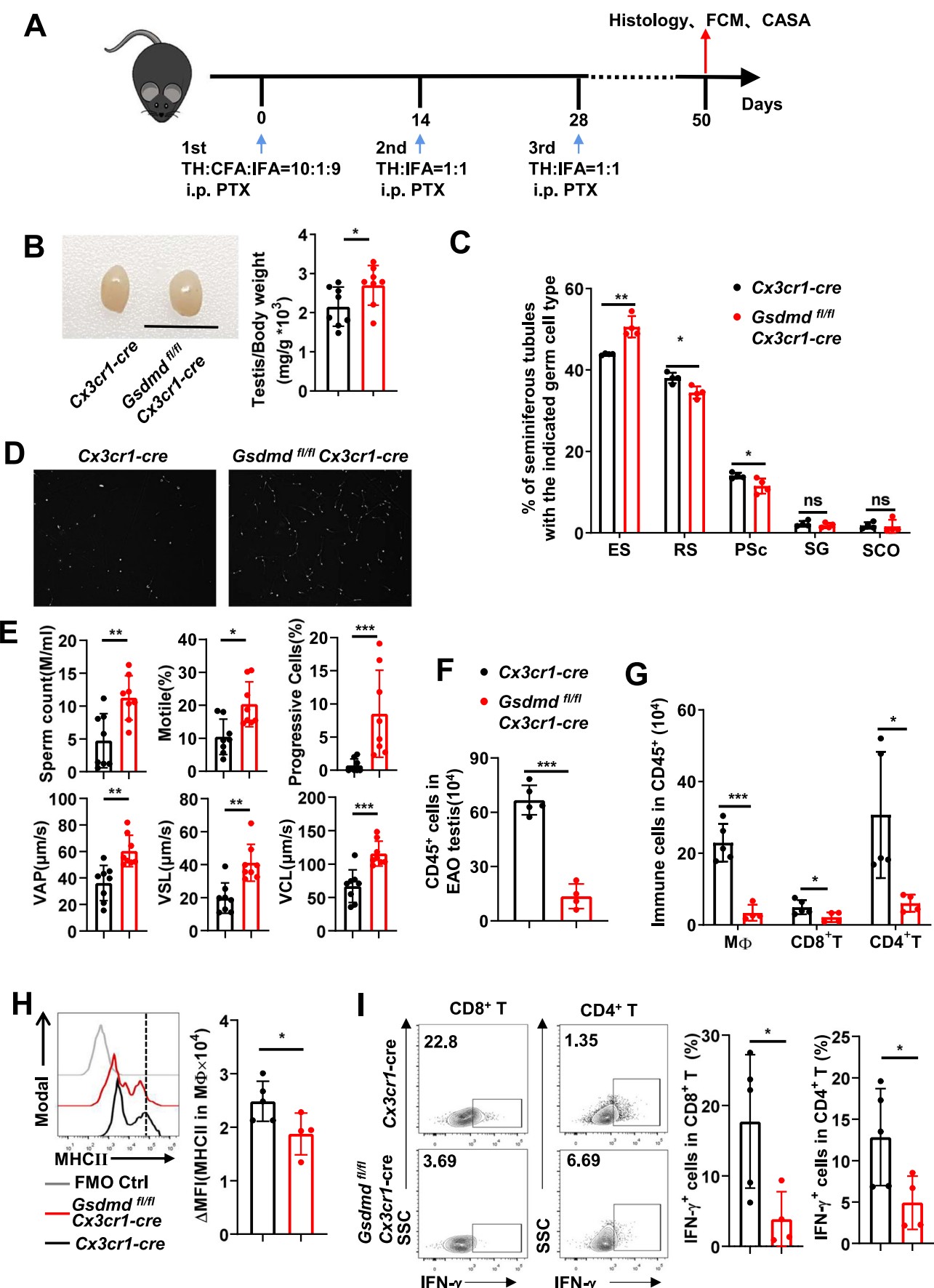

**Figure 8.   GSDMD in testicular macrophages promoted the development of EAO.**

Age-matched *Gsdmd*^fl/fl^*Cx3cr1*-cre and *Cx3cr1*-cre mice were induced with experimental autoimmune orchitis (EAO). Testicular tissues were collected 50 days after the first immunization for subsequent experiments. (A) Schematic representation of the experimental procedures for EAO in (B–F). TH testicular homogenate, CFA complete Freund's adjuvant, IFA incomplete Freund's adjuvant, PTX pertussis toxin, FCM flow cytometry. (B) Testicular morphology and relative weight of testes of *Gsdmd*^fl/fl^*Cx3cr1*-cre and *Cx3cr1*-cre EAO mice. Scale bars, 1 cm (*n* = 8 mice per group). (C) Testicular histological changes of testes in Fig. EV4E calculated using the adaption of the classical Johnsen scoring system (*n* = 4 mice per group). (D, E) CASA of sperm count, ratio of motile and progressive sperm, and velocity of motile sperm (VAP, VSL, and VCL) of indicated mice, and data are presented as representative picture (D) and summary graph (E) (*n* = 8 mice per group). (F–I) The testes from *Gsdmd*^fl/fl^*Cx3cr1*-cre and *Cx3cr1*-cre EAO mice were collected for flow cytometric analysis: the number of total immune cells (F), the number of macrophages, CD4^+^ T cells, and CD8^+^ T cells (G); MFI of MHC class II in macrophages (H); proportion of IFN-γ^+^ cells in CD4^+^ and CD8^+^ T cells (I), (*Cx3cr1*-cre, *n* = 5, *Gsdmd*^fl/fl^*Cx3cr1*-cre, *n* = 4). Data information: Data are pooled from three independent experiments. Error bars show mean ± s.d. \**P* < 0.05, \*\**P* < 0.01, \*\*\**P* < 0.001, ns not significant. Two-tailed unpaired Student's *t* test was used unless otherwise stated. Mann–Whitney test for comparisons of Motile (%), Progressive (%), and VSL in (E), Mann–Whitney test for comparisons of SCO (%) in (C), and Mann–Whitney test for comparisons of CD4^+^ T cells in (G). Source data are available online for this figure.

the Percoll (GE Healthcare) density gradient, and mononuclear cells were isolated by collecting the interface fractions between 40 and 80% Percoll. Single-cell suspensions were stained with FVD eFluor 506 (eBioscience, 65-0866-14), anti-CD16/32 (eBioscience, 14-0161-85), anti-CD45-AlexaFluor 700 (eBioscience, 56-0451-82), anti-CD4-APC (eBioscience, 17-0041-82), anti-CD4-PE-Cy7 (eBioscience, 25-0042-82), anti-CD69-FITC (Biolegend, 104505), anti-CD8-PE (eBioscience, 12-0081-83), anti-F4/80-APC (eBioscience, 17-4801-82), anti-MHCII-APC-Cy7 (Biolegend, 107627), anti-CD86-PE-Cy7 (eBioscience, 25-0862-82) in PBS containing 2% fetal bovine serum (FBS) to analyze the surface markers. For intracellular cytokine staining, cells were stimulated with phorbol 12-myristate 13-acetate (PMA, Multi Science), ionomycin (Multi Science), and brefeldin A (BFA, Invitrogen) for five hours. After stimulation, cells were stained with FVD eFluor 506 (eBioscience, 65-0866-14), anti-CD45-FITC (eBioscience, 11-045182), anti-CD4-APC-Cy7 (Biolegend, 100414), anti-CD8a-PE (eBioscience, 12-0081-83), anti-IFNγ-PerCP-Cy5.5 (eBioscience, 85-45-7311-82), anti-TNFα-APC (eBioscience,17-7321-82) according to the manufacturer's instructions. For antigen-specific T cells assay, T cells were stained with FVD eFluor 506, anti-CD44, anti-IFNγ, anti-TNFα and H-2K(b)/SIINFEKL Tetramer-PE (HELIXGEN) for flow cytometry analysis. Flow cytometry was performed on an Attune NxT flow cytometer (Thermo Fisher Scientific) or BDVerse, and data were analyzed by FlowJo 10.0.7 software.

## Macrophage sorting and RNA-seq

CD45^+^ CD11b^+^ Ly6g^−^ F4/80^+^ macrophages were isolated from the testes of WT and *Gsdmd*^−/−^ mice 36 h after UPEC infection by sorting on a BD FACSAria. Total RNA was extracted using the TRIzol reagent (Thermo, 15596018). cDNA library construction and RNA-seq were performed by LC-Bio Technology Co., Ltd. using an Illumina Novaseq6000. Clean reads were mapped to the mouse genome using HISAT2. Matched reads were calculated and normalized to fragments per kilobase of transcript per million mapped reads (FPKM). The fold change was calculated for all possible comparisons, and a 1.2-fold cutoff (*P* < 0.05) was used to select genes with significant changes in expression. GO-BP and Reactome pathway analyses were performed using the online tool, DAVID Bioinformatics Resources 6.8 (http://david.ncifcrf.gov), and significant DEGs (fold change >1.2, *P* value < 0.05) as target genes. GSEA was performed using GSEA v3.0. Raw data and processed files were deposited in the NCBI Sequence Read Archive database under the accession code PRJNA985126.

## Antigen presentation assay

BMDMs and T cells were co-cultured as described previously (Gojkovic et al, 2021). Briefly, for BMDM culture, bone marrow cells from the femur and tibia of WT and *Gsdmd*^−/−^ male mice (8−12 weeks) were cultured for 6 days in DMEM supplemented with 10% fetal bovine serum (FBS) and 10% L929 conditioned medium. The primary BMDMs were plated on 96-well round bottom plates (50,000 cells/well) and grown for 2 h. Subsequently, the BMDMs were pre-stimulated with 100 ng/ml LPS (Enzolife Sciences, ALX-581-0) for 24 h, treated with nigericin (Invivogen, tlrl-nig-5) for 15 min, washed thrice with PBS, and cultured in DMEM supplemented with 10% FBS and 100 ng/ml OVA peptide (GenScript, RP10611) for 6 h. The expression of antigen presentation-associated genes in BMDMs was analyzed using flow cytometry. Naive CD8^+^ T cells were isolated from the spleen and lymph nodes of OT-I mice (8−12 weeks) using a cell isolation kit (STEMCELL, 19858). For co-culturing, 20,000 T cells were added to the BMDMs in a 96-well plate after 3 days, and the activation and function of antigen-specific T cells were detected using flow cytometry.

## Western blot analysis

Testicular tissues were collected from mice 36 h after UPEC injection and homogenized in radioimmunoprecipitation assay buffer containing a protease inhibitor cocktail (Sigma, P8340), followed by incubation on a shaker at 4 °C for 1 h. The lysates were centrifuged at 12,000×*g* for 15 min at 4 °C, and then the supernatants were collected to assay protein concentrations using the bicinchoninic acid protein assay. Equal quantities of the samples were resolved using sodium dodecyl sulfate-polyacrylamide gel electrophoresis, transferred onto nitrocellulose membranes, and incubated with the indicated antibodies. The antibodies and working dilutions are listed as follows: anti-NLRP3 (1:1000, Adipogen, AG-20B-0014), anti-ASC (1:1000, Santa Cruz, SC-22514-R), anti-CASPASE-1 (1:1000, Adipogen, AG-20B-0042), anti-CASPASE-11 (1:1000, Novus, NB120-10454), anti-GSDMD (1:1000, Abcam, ab209845), β-actin (1:1000, Sigma, A1978), IRDye 680RD anti-mouse (1:5000, LI-COR Biosciences, 926-68070), IRDye 800CW anti-rabbit (1:5000, LI-COR Biosciences, 926-32211), and goat anti-rabbit-HRP (1:3000, Thermo Fisher, 31402).

## RNA extraction and quantitative reverse-transcriptase PCR

Total RNA was extracted from the testes using TRIzol reagent (Thermo, 15596018) following the manufacturer's instructions.

**The paper explained**

**Problem**

Orchitis, the inflammation of the testis resulting from infection or autoimmune disorders, contributes significantly to male infertility. Existing treatments involving antibiotics and broad-spectrum anti-inflammatory drugs often fail to fully restore fertility and are accompanied by undesirable side effects. Therefore, there's a pressing need to delve into the pathogenesis of orchitis and explore alternative therapeutic approaches for more effective treatment.

**Results**

We found that Gasdermin D (GSDMD) is activated during uropathogenic *Escherichia coli* (UPEC)-induced orchitis. GSDMD deficiency in macrophages limits inflammation and spermatogenesis impairment during UPEC-induced acute orchitis and chronic autoimmune orchitis. On a mechanistic level, GSDMD ablation in macrophage impairs CD8 + T-cell response by impeding antigen presentation capacities. The use of GSDMD inhibitors demonstrated protective effects in mice subjected to UPEC-induced orchitis.

**Impact**

Our study demonstrates the role of GSDMD in driving orchitis. In addition, the study suggests that targeting GSDMD may be a potential therapeutic avenue to treat both UPEC-induced acute orchitis and chronic autoimmune orchitis.

cDNA synthesis kit (Vazyme, R222-01) was used to synthesize the cDNA according to the manufacturer's protocols. Quantitative reverse-transcriptase PCR (qRT-PCR) was conducted using SYBR Green Supermix (Vazyme, Q111-02). The following primers were used:

| | F (5′ → 3′) | R (5′ → 3′) |
|---|---|---|
| Hprt | GTCCCAGCGTCGTGATTAGC | TGGCCTCCCATCTCCTTCA |
| Gsdmd | CCATCGGCCTTTGAGAAAGTG | ACACATGAATAACGGGGTTTCC |
| Nlrp3 | ATTACCCGCCCGAGAAAGG | TCGCAGCAAAGATCCACACAG |
| Aim2 | GTCACCAGTTCCTCAGTTGTG | CACCTCCATTGTCCCTGTTTTAT |
| Asc | CTTGTCAGGGGATGAACTCAAAA | GCCATACGACTCCAGATAGTAGC |
| Casp11 | CTTCAGGCACATCAAACACCA | GGGGCATCACACTCAGAATAGT |
| Casp1 | ACAAGGCACGGGACCTATG | ACAAGGCACGGGACCTATG |
| Il1b | ATGCCACCTTTTGACAGTGATG | GTTGATGTGCTGCTGCGAGA |
| Il-6 | CTTGGGACTGATGCTGGTGAC | GCCATTGCACAACTCTTTTCTC |
| Tnfa | TACTGAACTTCGGGGTGATCG | TCCTCCACTTGGTGGTTTGC |
| Ccl-2 | TTAAAAACCTGGATCGGAACCAA | GCATTAGCTTCAGATTTACGGGT |
| Cxcl-10 | CCAAGTGCTGCCGTCATTTTC | GGCTCGCAGGGATGATTTCAA |
| Il1a | CGAAGACTACAGTTCTGCCATT | GACGTTTCAGAGGTTCTCAGAG |
| Tnfaip3 | GAACAGCGATCAGGCCAGG | CCTCCGTGACTGATGACAAGAT |
| Ccl6 | GCTGGCCTCATACAAGAAATGG | GCTTAGGCACCTCTGAACTCTC |
| H2-Ab1 | AGCCCCATCACTGTGGAGT | GATGCCGCTCAACATCTTGC |
| H2-DMa | CTCGAAGCATCTACACCAGTG | TCCGAGAGCCCTATGTTGGG |
| H2-DMb1 | ACCCCACAGGACTTCACATAC | GGATACAGCACCCCAAATTCA |
| Psmb6 | GCCTTAGCTGTTCGTCGAG | TAGAACCACGCCCCCATTAAA |
| Psmb3 | CAGCGTCTCAAGTTCCGACTG | CGCTTCTCATACAGGAGGTTG |

## ELISA

Testicular tissues were collected from mice at 36 h after UPEC injection and homogenized in RIPA buffer containing protease inhibitor cocktail (Sigma, P8340), followed by incubation on a shaker at 4 °C for 1 h. The lysates were centrifuged at $12,000 \times g$ for 15 min at 4 °C, the supernatant was collected by centrifugation. Then the protein levels of IL-1β, IL-6, TNF-α, CCL-2, and CXCL-10 were measured according to the manufacturer's instructions.

## Immunofluorescence of testicular tissues

Freshly isolated testes were fixed with periodate-lysine-paraformaldehyde (PLP) at 4 °C for 12 h, following which the testicular tissues were cut in half and fixed with a new fixative solution for 12 h, dehydrated, and mounted in paraffin according to standard protocols. For frozen sections, testicular tissues were fixed in 4% paraformaldehyde for 48 h, placed in 30% sucrose solution for dehydration, and embedded in OCT. For staining of CASPASE-1, the frozen sections were blocked in 5% goat serum for 1 h and then incubated with primary antibodies diluted in 0.2% Triton X-100 in PBS at 4 °C for 12 h. For staining of NLRP3 and GSDMD, and co-staining of GSDMD with F4/80, the paraffin sections were restored with sodium citrate antigen repair solution. The antibodies and working dilutions used were rabbit anti-GSDMD (1:500), rat anti-F4/80 (1:200), rabbit anti-CASPASE-1 (1:200), and mouse anti-NLRP3 (1:200). Goat anti-rabbit antibodies conjugated to Alexa 555 (1:500, Invitrogen, A21429), goat anti-mouse antibodies conjugated to Alexa 488 (1:500, Invitrogen, A-21422), and goat anti-rat antibodies conjugated to Alexa 488 (1:500, Invitrogen, A11006) were used as secondary antibodies. Images were captured using a Nikon 50i fluorescence microscope and processed using the ImageJ and AdobePhotoshopCS6 software.

## Statistical analysis

Statistical analyses were performed using GraphPad Prism 8.0. The normality of the data was analyzed using the Shapiro–Wilk test. For normally distributed data, comparisons were performed using a two-tailed unpaired $t$ test or multiple $t$ tests. Continuous variable data with non-normal distributions were analyzed using nonparametric tests, such as the Mann–Whitney test. Data are expressed as the means ± standard deviation (s.d.). *$P < 0.05$, **$P < 0.01$, and ***$P < 0.001$ were considered significant.

## For more information

Proteomics Database in Reproductive Medicine: http://reprod.njmu.edu.cn/.

## Data availability

Data of flow cytometry analyses are available upon request, and RNA-seq data are available in the NCBI Sequence Read Archive database under the accession code PRJNA985126.

## Peer review information

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

## Acknowledgements

We thank Dr. V Dixit (Genentech, South San Francisco, CA, USA) for providing *Nlrp3⁻/⁻*, *Aim2⁻/⁻*, *Asc⁻/⁻*, and *Caspase-1/11⁻/⁻* mice, thank Dr. Feng Shao (National Institute of Biological Sciences, Beijing, China) for providing *Gsdmd⁻/⁻* mice, and thank Dr. Jiawei Zhou (Institute of Neuroscience, Chinese Academy of Sciences, Shanghai, China) for donating *Cx3cr1*-cre mice. We thank Dr. Yongning Lu (Reproductive Medicine Centre, Zhongshan Hospital, Fudan University) for providing UPEC strain CFT073 as a gift. This work was supported by the National Key R&D Program of China (2022YFA1303900 to SY and CM), the National Natural Science Foundation of China (32270921 and 82070567 to SY, 81802393 to BW, 81901227 and 82270539 to CM), the Open Project of State Key Laboratory of Reproductive Medicine of Nanjing Medical University (SKLRM-2023A2 to SY), the talent cultivation project of "Organized scientific research" of Nanjing Medical University (NJMURC20220014 to SY), the Open Project of Chinese Materia Medica First-Class Discipline of Nanjing University of Chinese Medicine (No.2020YLXK017 to BW), the Natural Science Foundation of Jiangsu Province (BK20221352 to BW), the Priority Academic Program Development of Jiangsu Higher Education Institutions (to BW), the Natural Science Excellent Youth Foundation of Jiangsu Province (SBK2022030080 to CM), China Postdoctoral Science Foundation (2023M741787 to YY).

## Author contributions

**Chunmei Ma**: Data curation; Formal analysis; Funding acquisition; Methodology; Writing—original draft. **Jiajia Huang**: Data curation; Formal analysis; Methodology. **Yuying Jiang**: Formal analysis; Methodology. **Lu Liu**: Formal analysis. **Na Wang**: Data curation. **Shaoqiong Huang**: Formal analysis. **Honghui Li**: Formal analysis. **Xiangyu Zhang**: Methodology. **Shuang Wen**: Project administration. **Bingwei Wang**: Funding acquisition; Methodology; Writing—review and editing. **Shuo Yang**: Conceptualization; Data curation; Formal analysis; Supervision; Funding acquisition; Methodology; Project administration; Writing—review and editing.

## Disclosure and competing interests statement

The authors declare no competing interests.

# Expanded View Figures

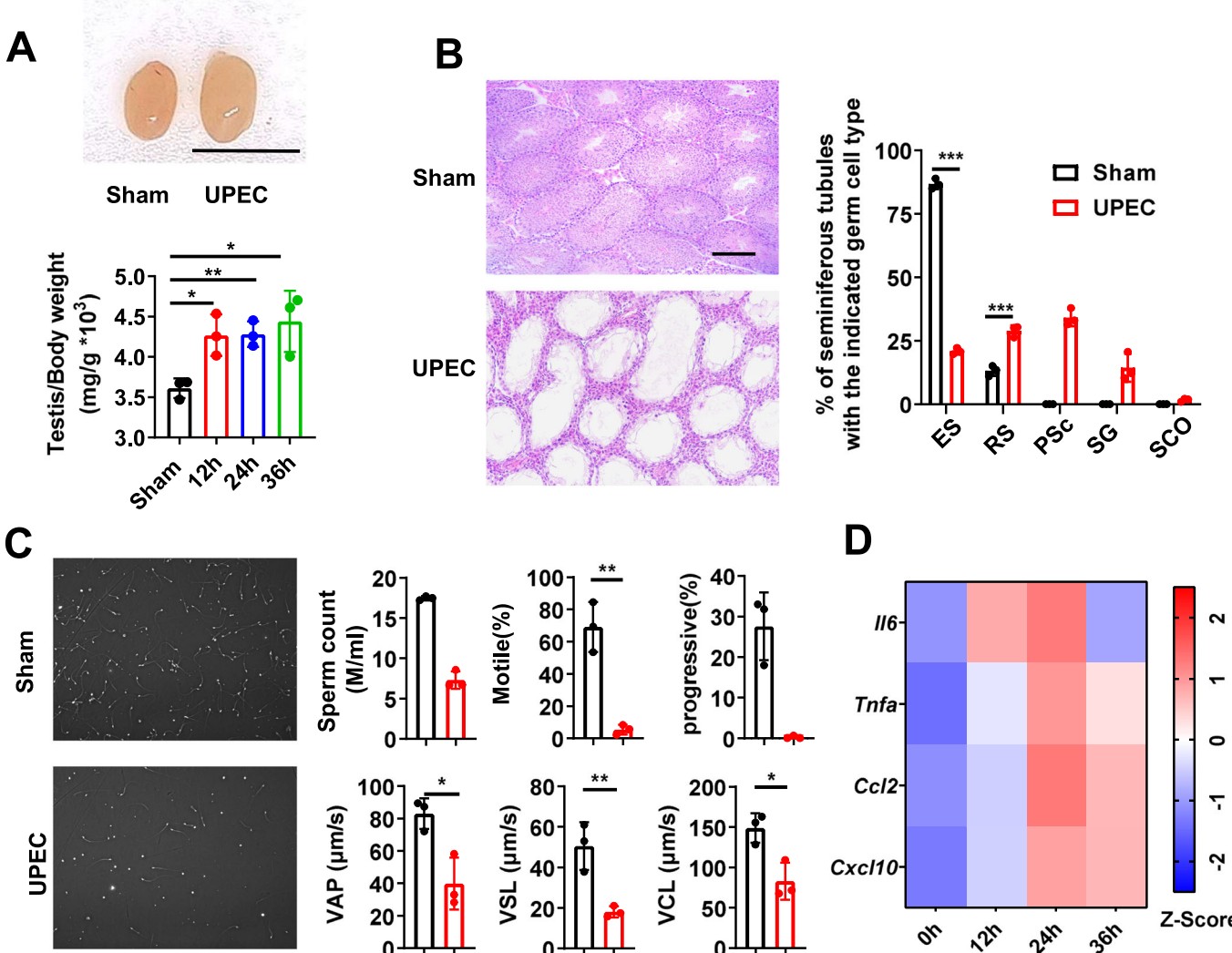

**Figure EV1. UPEC can successfully induce acute orchitis.**

(A) Gross morphology of representative testes from sham control mice and UPEC-induced orchitic mice, and relative weight of testis (testis weight (mg)/body weight (g) *10³). Scale bars, 1 cm, ($n = 3$ mice per group). (B) Testicular histological changes were detected using H&E staining and the adaption of classical Johnsen scoring system, elongated spermatids (ES), round spermatids (RS), pachytene spermatocytes (PSc), spermatogonia (SG), Sertoli-cell-only (SCO). Scale bars, 100 μm, ($n = 3$ mice per group). (C) Sperm count, ratio of motile and progressive sperm, and velocity of motile sperm (VAP, average path velocity; VSL, straight-line velocity; VCL, curvilinear velocity) were detected using computer-assisted sperm analysis (CASA), ($n = 3$ mice per group). (D) Relative mRNA levels of *Il-6*, *Tnfa*, *Ccl-2*, and *Cxcl-10* in the testes of mice treated with UPEC ($1 \times 10^5$ CFU) for indicated time ($n = 3$ mice per group). Data information: Data are pooled from two independent experiments. Error bars show mean ± s.d. *$P < 0.05$, **$P < 0.01$. Two-tailed unpaired Student's $t$ test was used unless otherwise stated.

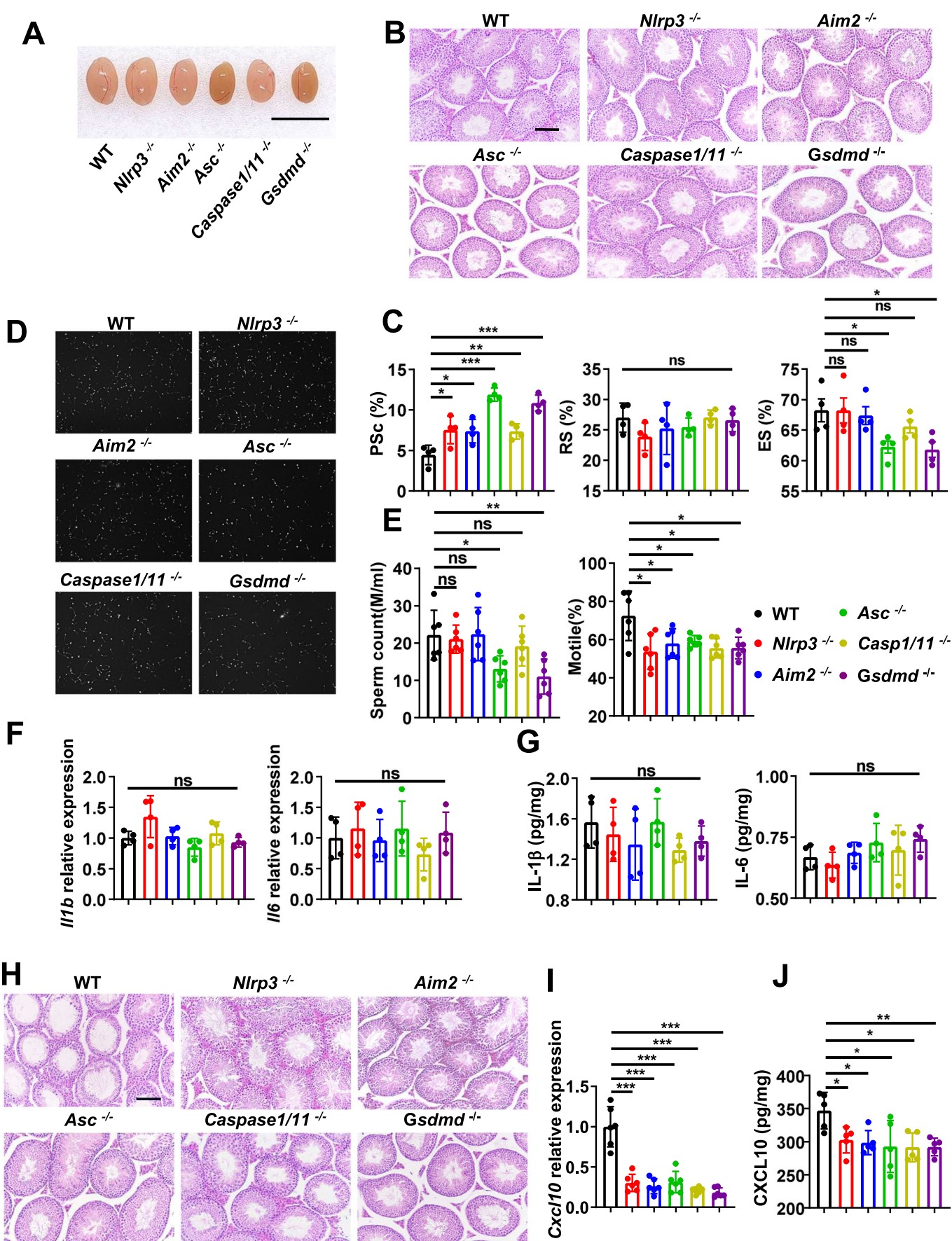

◄ **Figure EV2. Deletion of GSDMD did not induce spontaneous orchitis.**

Testicular tissues from age-matched male WT, *Nlrp3*$^{-/-}$, *Aim2*$^{-/-}$, *Asc*$^{-/-}$, *Caspase-1/11*$^{-/-}$, and *Gsdmd*$^{-/-}$ mice were collected for subsequent experiments. (**A**) Testicular morphology. Scale bars, 1 cm. (**B**) Representative images of H&E staining of testes from the mice in (**A**). (**C**) Testicular histological changes of testes in (**B**) were calculated using the adaption of classical Johnsen scoring system. Scale bars, 100 μm, ($n = 4$ mice per group). (**D**) Representative CASA picture of sperm status of indicated mice. (**E**) Sperm count, ratio of motile sperm of indicated mice were detected using CASA ($n = 6$ mice per group). (**F**) Relative mRNA levels of *Il1b* and *Il-6* in the testes of indicated mice ($n = 4$ mice per group). (**G**) ELISA analysis of IL-1β and IL-6 levels in the testes of the indicated mice ($n = 4$ mice per group). (**H**) Representative images of H&E staining of testes from the indicated mice treated with UPEC 7 days. (**I**) Relative mRNA levels of *Cxcl-10* in the testes from the indicated mice treated with UPEC for 24 h ($n = 6$ mice per group). (**J**) ELISA analysis of CXCL-10 in the testes from the indicated mice treated with UPEC for 36 h ($n = 6$ mice per group). Data information: Data are pooled from three independent experiments for (**C, E–G, I, J**). Data are representative of three independent experiments for (**A, B, D**), and (**H**) error bars show mean ± s.d. *$P < 0.05$, **$P < 0.01$, ***$P < 0.001$, ns, not significant. Two-tailed unpaired Student's *t* test was used unless otherwise stated. Mann–Whitney test for comparisons of relative expression of *Il-6* between WT and *Caspase-1/11*$^{-/-}$ in (**F**), and Mann–Whitney test for comparisons of CXCL-10 level between WT and *Caspase-1/11*$^{-/-}$ in (**J**).

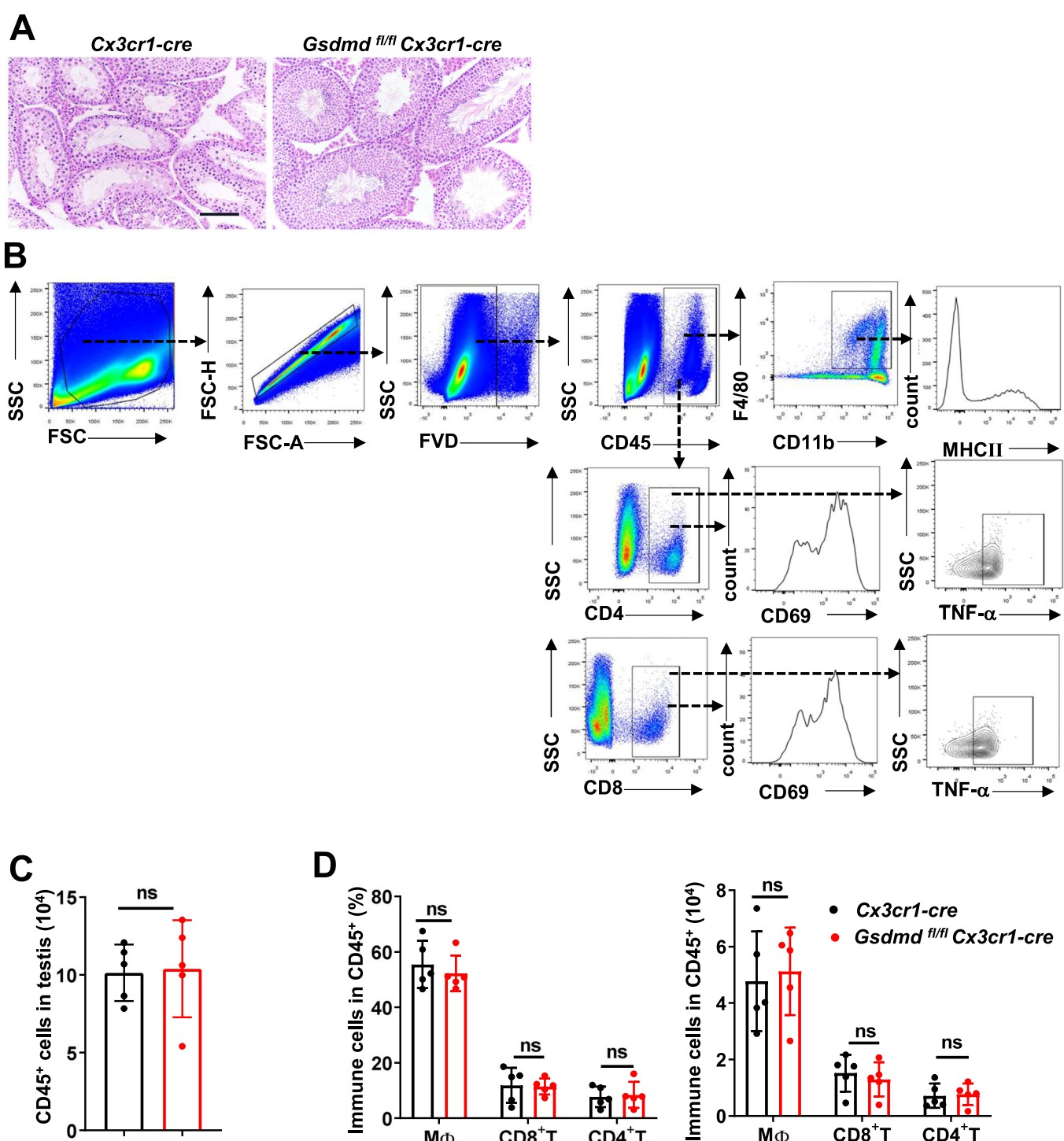

**Figure EV3. GSDMD deficiency in macrophages alleviated testicular inflammation during UPEC-induced acute orchitis.**

(A) Representative images of H&E staining of testes from $Gsdmd^{fl/fl}Cx3cr1$-cre and $Cx3cr1$-cre mice treated with UPEC for 7 days. (B) Flow cytometric gating strategy for Fig. 4. (C, D) Testes from $Gsdmd^{fl/fl}Cx3cr1$-cre and $Cx3cr1$-cre mice aged at 10–12 weeks were collected for flow cytometric analysis. The number of total immune cells (C) and the quantified absolute number (D) of the macrophages (CD11b⁺F4/80⁺), CD8⁺ T cells, CD4⁺ T cells, ($n = 5$ mice per group). Data information: Error bars show mean ± s.d. ns, not significant. Two-tailed unpaired Student's $t$ test was used unless otherwise stated.

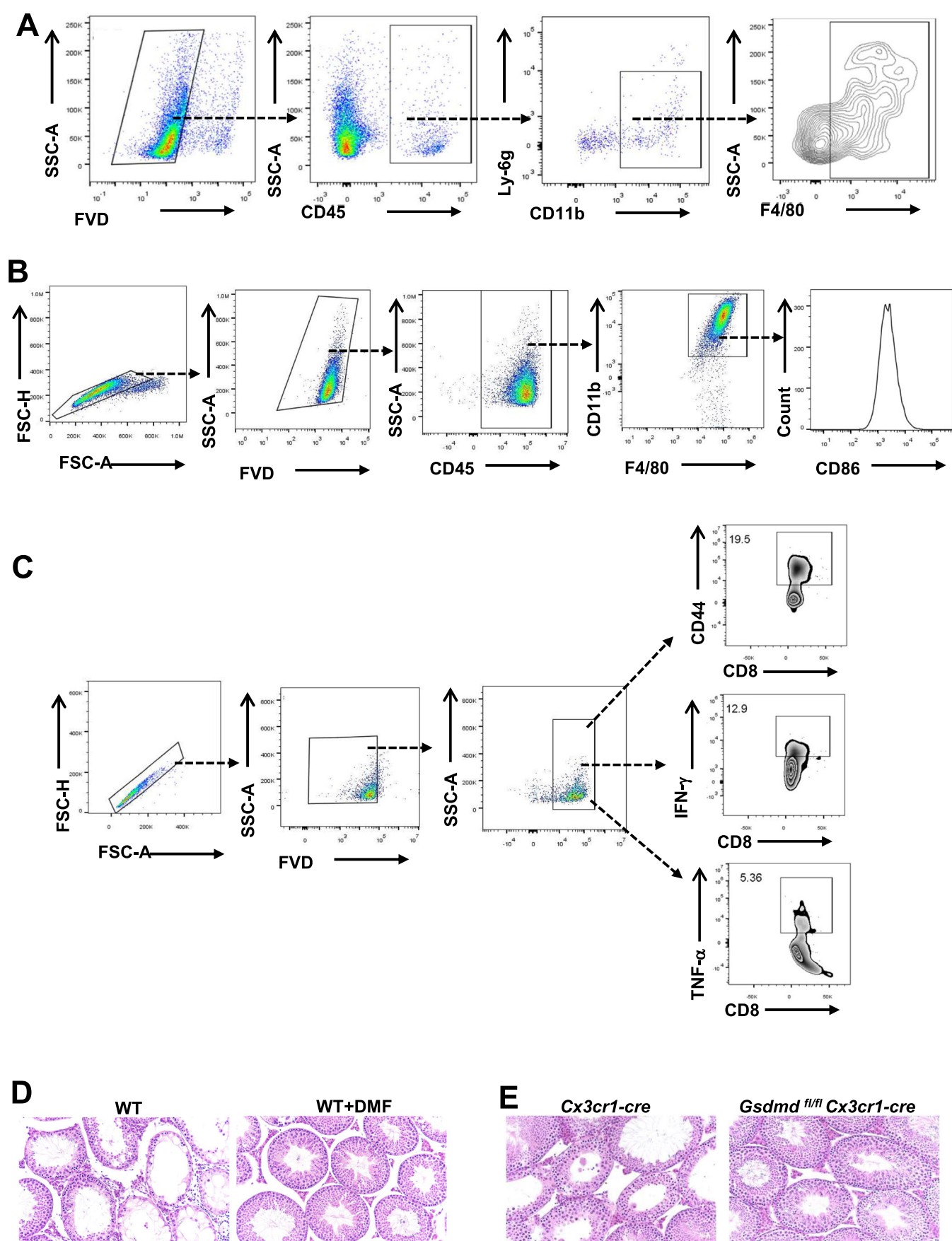

◀ **Figure EV4. FACS gating strategy.**

(A) FACS gating strategy for sorting testicular macrophage in Fig. 5. (B, C) FCM gating strategy for analysis of BMDMs (B) and T cells (C) in Fig. 6. (D) Testicular histological changes of testes in UPEC-induced orchitic WT mice injected vehicle or DMF were detected by H&E staining. Scale bars, 100 μm. (E) Testicular histological changes of testes in *Gsdmd*<sup>fl/fl</sup>*Cx3cr1*-cre and *Cx3cr1*-cre EAO mice were detected using H&E staining. Scale bars, 100 μm.

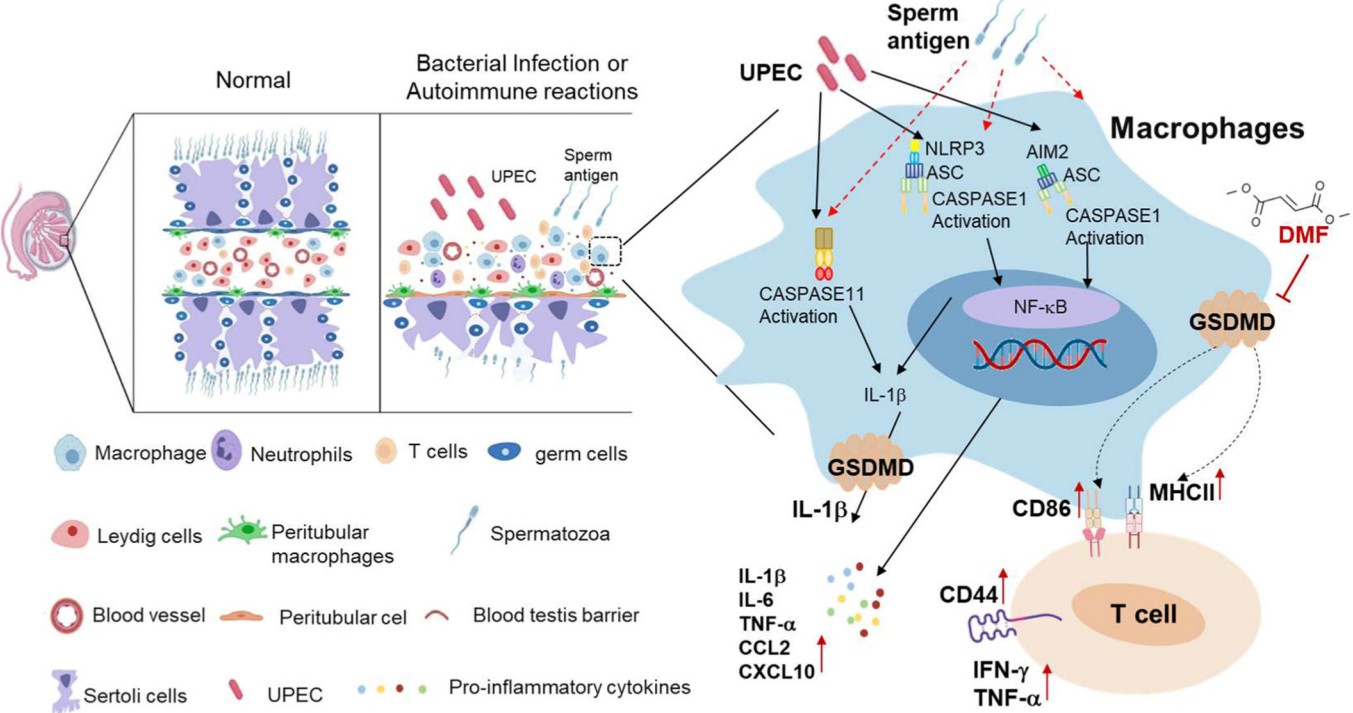

**Figure EV5.  Model showing how GSDMD in macrophages drives orchitis.**

During orchitis, GSDMD and other inflammasome-related molecules are activated by UPEC or sperm antigens, thereby promoting antigen presentation and inflammatory responses in testicular macrophages to enhance T-cell responses, which driven the development of acute and chronic orchitis. Administration of GSDMD inhibitors conferred protection against orchitis in mice.

