## [Peer Review File · EMBO Molecular Medicine]

Gasdermin D in macrophages drives orchitis by regulating inflammation and antigen presentation processes

Shuo Yang, Chunmei Ma, Jiajia Huang, Yuying Jiang, Lu Liu, Na Wang, Shaoqiong Huang, Honghui Li, Xiangyu Zhang, Shuang Wen, and Bingwei Wang

DOI: [10.15252/emmm.202318267](https://doi.org/10.15252/emmm.202318267)

Corresponding authors: Shuo Yang (shuoyang01@njmu.edu.cn) , Bingwei Wang (bingweiwang@njucm.edu.cn)

Review Timeline:

Submission Date:	4th Jul 23
Editorial Decision:	21st Jul 23
Appeal:	28th Jul 23
Editorial Decision:	29th Aug 23
Revision Received:	5th Oct 23
Editorial Decision:	9th Nov 23
Revision Received:	28th Nov 23
Accepted:	1st Dec 23

Editor: Zeljko Durdevic

Transaction Report:

21st Jul 2023

Dear Prof. Yang,

Thank you for the submission of your manuscript to EMBO Molecular Medicine. We have now received feedback from the two reviewers who agreed to evaluate your manuscript.

As you will see from their reports pasted below, while they recognize interest of your study, they also raise serious concerns, particularly regarding the lack of evidence to support the main conclusions of the study. As clear and conclusive insight into a novel, clinically relevant observation is crucial for publication in EMBO Molecular Medicine, and together with the fact that we only accept papers that receive enthusiastic support upon initial review, I am afraid that we cannot offer to consider the manuscript further.

I am sorry that I could not bring better news this time and hope that the referee comments are helpful in your continued work in this area.

Yours sincerely,

Zeljko Durdevic

***** Reviewer's comments *****

Referee #1 (Comments on Novelty/Model System for Author):

The model system of this paper is correct adequate.

Referee #1 (Remarks for Author):

Summary:

The manuscript provides the first demonstration of GSDMD's role in driving UPEC-induced orchitis and detailed delineation of the underlying mechanism. In addition, it was found that the administration of GSDMD inhibitors conferred the mice protection against UPEC infection induced orchitis. These results suggested that targeting GSDMD may be an efficient therapeutic target to treat UPEC-induced acute orchitis and chronic autoimmune orchitis. It provides a promising transformation therapy for bacterial orchitis.

General remarks:

The paper deals with a topic that is relevant to this journal. In general, the paper is completely structured and the logic is clear. However, I have some critical remarks for the authors to consider before being able to recommend the paper be accepted.

Major improvement suggestions:

1. The layout of illustrations in the article needs to be improved:

(1) The boundary between B/C in Figure S1, E/D in Figure 3 and C/D in Figure S3 is not clear;

(2) Figure 2 is slightly crowded;

(3) In the "GSDMD in testicular macrophages promotes UPEC-induced acute orchitis" part, some experimental results are scattered in two figures, the author may consider merging them into one, such as "(Fig 3E, Fig S3A)"; "(Fig 3H, I, Fig S3B)" and "(FIG3J, 3K, FIG3B)".

2. Some elements in the figure are wrong or should be explained in detail:

(1) The upper arrow of the "UPEC-DAPI" group in Figure 3-B is not pointed accurately;

(2) "WT" appears in both Figure 1 and Figure 2, however, "WT" in Figure 1 refers to wild-type mice, and "WT" in Figure 2 refers to wild-type mice infected by UPEC;

(3) The abbreviations in Figure 7-A should be explained in detail in figure legends, such as "TH" and "PTX".

3. "GSDMD deficiency in macrophages induces inflammation and damaged spermatogenesis during acute and chronic orchitis" in "Abstract" is ideologically contradictory to the full text.

4. It is more logical to consider switching the order of the "GSDMD deficiency controls UPEC-induced acute orchitis" part and

the "GSDMD in testicular macrophages promotes UPEC-induced acute orchitis" part.

5. Why choose "CXCL10" as the typical index of pro-inflammatory chemokines in the "GSDMD deficiency controls UPEC-induced acute orchitis" part?

6. What is the difference between the "Gsdmd^{-/-} mice" in "GSDMD deficiency controls UPEC-induced acute orchitis" part and the "Gsdmd^{fl/fl}Cx3cr1-cre mice" in "GSDMD in testicular macrophages promotes UPEC-induced acute orchitis" part?

7. Please explain the function of "LPS plus nigericin" in the "GSDMD boosts T cell response by enhancing antigen presentation of macrophages" part.

8. In the "GSDMD in macrophages aggravates chronic autoimmune orchitis during EAO development" part, why are the testicular size and testis-body weight ratio of GSDMD deficient mice larger, which indicates that their testicular degeneration has been improved?

Referee #2 (Comments on Novelty/Model System for Author):

FOR IMMUNOHISTOCHEMISTRY: all of the images in this manuscript show quite large gaps between tubules and where the interstitial space should be. This is resultant from an error in fixation or tissue processing and the structure of the tissue is not intact. I would recommend repeating these experiments to obtain higher quality staining and imaging. If it cannot be repeated, I recommend taking more images to see whether better imaging can be obtained, or removing this from the manuscript. If the authors feel the IHC is essential to the manuscript, then perhaps removing it to supplementary would be more appropriate. In any case, quantification of positive staining should be performed for all figures. The authors rely on staining to draw many conclusions, therefore it would be appropriate to have quantitative data rather than qualitative, as is currently provided.

FOR FLOW CYTOMETRY: the axis tick values need to be added back into ALL FIGURES containing flow cytometry. Some of the positive populations appear to be gated very close to what I assume is the zero tick on both the x and y axes, which makes the flow cytometry data questionable, although the separation of 'positive populations' does appear distinct. Showing the full gating strategies will help clear this issue up. Further, replicate values for the flow cytometry data can be graphed, and statistical analysis performed, to add some quantitative data to the argument being made, rather than heavily relying on representative and qualitative data.

FOR DATA PRESENTATION AND STATISTICS:

The authors have used SEM, which should only be used in population statistics; here standard deviation (SD) would be appropriate.

Some graphs, or even some bars within graphs, have bidirectional error bars and some only have unidirectional error bars, and this needs to be made consistent, preferably with bidirectional bars.

Many of the axis labels are not descriptive enough, lack units, or are not labelled at all (Figure 4, which also lacks individual data points in F).

Additionally, many of the graphs throughout the manuscript that have been given * or ** have a great deal of overlapping data between compared groups. This makes the statistical analysis incorrect. For my own interest, I replicated Figure 6F, in Prism, as indicated. I ran normality tests, and additional outlier analysis as the CXCL10 data has a lot of spread. I could not generate the same statistical significance that the authors have found, as expected with data that overlaps as much as seen here. A review of the data presentation, tests used, and significance outcome is required.

Referee #2 (Remarks for Author):

Summary: This manuscript investigates the role of GSDMD and GSDMD-depletion in UPEC-mediated orchitis. While this is an interesting topic and is generally a well thought out study, the manuscript contains many grammatical/syntax errors and would benefit from a thorough proof read, and the manuscript contains numerous experimental flaws.

PAPER EXPLAINED and ABSTRACT:

This reviewer feels that the authors should rework the latter half of the abstract to reflect any changes made consequent to review, being careful not to oversell the mechanistic aspect of the study. Similarly, the authors should ensure they do not overreach for the conclusion that GSDMD makes a feasible therapeutic target for treating orchitis, as while the study may provide a foundation for that study, the current study does not investigate the necessary

INTRODUCTION:

The authors indicate that 'approximately 60%' of cases of male infertility related to infection and inflammation are attributed to UPEC. However, the literature is much more complex than this, and rates vary greatly between studies. This could be due to type of testing, demographics, geographics, etc. but it is insufficient to cite only one article, from 2008, as the foundation for this statement. Please provide an unbiased and up-to-date comment on the role of UPEC in male infertility.

Further, the reference given (Wiles, Kulesus et al., 2008) does not contain the information given. This reference does not

mention a 60% value, infertility, or orchitis that this reviewer could find after skimming the article and searching for keywords. <https://academic.oup.com/molehr/article/26/4/215/5721557> this article cites a 60% value on a similar topic, but it is not the same statistic that you have given.

A thorough review of all references is required.

At the start of the second page of the introduction, the authors indicate that during orchitis, testicular macrophages 'transform' into pro-inflammatory macrophages that highly express proinflammatory cytokines. This is inaccurate, and the authors need to review the appropriate literature to understand the origin of inflammatory cytokines in the testes. This should include the role of recruited monocytes as key proinflammatory mediators.

Similarly, the APC ability of testicular macrophages is different to standard macrophages in dogma, so please review your statements and conduct a more thorough update of the literature to understand these two topics. Please ensure that the inflammation pathway information is specific to testicular macrophages in line with your updated reading.

RESULTS:

GENERAL NOTE FOR IMMUNOHISTOCHEMISTRY: all of the images in this manuscript show quite large gaps between tubules and where the interstitial space should be. This seems to be resultant from an error in fixation or tissue processing and the structure of the tissue no longer appears intact. If possible, this reviewer would recommend repeating these experiments to obtain higher quality staining and imaging. If it cannot be repeated, I recommend taking more images to see whether better imaging can be obtained, or removing this from the manuscript. If the authors feel the IHC is essential to the manuscript, then perhaps removing it to supplementary would be more appropriate. In any case, quantification of positive staining should be performed for all figures. The authors rely on staining to draw many conclusions, therefore it would be appropriate to have quantitative data rather than qualitative, as is currently provided. Finally, please specify how many fields of view your images are representative of in the figure legend or methods, once quantification as been completed.

Figure 1:

The Figure is well laid out and the panels are clear, there are several issues however.

B, Z-score should be Z-score.

In Figure 1c, multiple bands are observed in the Western blot of Pro CASP11 and others, which is fine, but the blots are cropped, which means we cannot see other potential bands and this is not acceptable by the journal and today's standards.

Either in the figure or as a supplementary, full sized, uncropped blots must be provided.

The immunostaining images in Figure 1D appear to be out of focus in panels for NLRP3 and GSDMD.

The Sham-GSDMD staining in Figure 1D vs Figure 3B is quite different, with quite a lot more staining present in the latter. What is the reason for this? Have the authors ensured that, in both cases, representative images have been provided? If so, why are these so different?

In the figure legend, please clarify whether $n = 5$ is the total number of mice used, or whether the three independent experiments conducted were each $n = 5$, giving a total of $n = 15$.

Please define all of the abbreviations used in the figure, in the figure legend. E.g. CASP1.

Figure 2:

The figure legend indicates that Sham as well as UPEC infected mice were assessed, however, I cannot see any data from Sham mice in this figure. Please address this, most appropriately would be to include the Sham data in this figure for side-by-side comparisons of KOs.

For the testis/Body weight graph in A, please provide a unit of measure, e.g. mg.

In the figure legend, please define all abbreviations, e.g. SCO, SG, M/mL

Authors state that the Johnsen Criteria method was used for assessing the seminiferous tubules in C, however, my understanding is that this method results in a score from 1 to 10, not a percentage, so please explain clearly in the methods/legend how this percentage has been obtained.

Please also clarify how tubules were staged, in order to obtain the percentages for round and elongated spermatocytes, for example, as I currently cannot see if this one performed, to standardize the counting (i.e. the number of these types of cells is dependent on the stage, was this accounted for).

In Figure 2/S1, there is a white line/spot observed on the testis, is this just light reflecting on damp tissue at the time the photograph was taken, or something else that the authors have annotated on the image?

For D, the image are really too small to be useful. Suggest moving these to supplementary if the authors feel they are necessary to the manuscript. Alternatively, enlarge the images and condense some of the graphical information. Condensing the graphical information would also help improve the readability of the figure.

For E and F, please clarify your n values. Is this $n = 2$ mice repeated 3 times, to give $n = 6$ in total? By pooled data, do you mean to say that your triplicate independent experiments are shown on a single graph, or do you mean that the actual samples were pooled?

In Figure S2A, the color of the testis appears different in Gsdmd^{-/-} and Asc^{-/-} mice. The reason for this color difference could be expanded upon if space permits, or if not relevant, please explain in response.

In Figure S2B, the H&E staining of Gsdmd^{-/-} and Asc^{-/-} mice shows an increase in interstitial space. The authors claim that the testis size is reduced in these mice, but it is unclear how they relate the testis size to the increase in interstitial space.

Figure 3:

B - comments as above about quality of sections and destruction of tissue structure. These images are not appropriate to draw conclusions from.

The authors claim to have generated macrophage conditional knockout mice of GSDMD (*Gsdmdfl/flCx3cr1-cre*), but they have not shown any data demonstrating GSDMD deletion in macrophages.

The decrease in immune cells after UPEC infection is shown in *Gsdmdfl/flCx3cr1-cre* mice, but the immune cell population in *Gsdmdfl/flCx3cr1-cre* mice under physiological conditions is not addressed.

In Figure 3I, the y-axis represents immune cells in number (10⁴) in CD45⁺ cells. However, it is generally known that macrophages contribute around 60% of the immune cell population. This information is not reflected in the graph. Please justify why your results are widely different to existing literature.

In Figure S3B, In the gating strategy is not clearly written or shown which parent population were taken.

In Figure S3B, the flow cytometry plot depicting macrophages reveals the presence of two distinct populations. It is possible that one of these populations corresponds to infiltrated monocytes rather than true macrophages. Notably, only a small proportion of the F4/80^{high}CD11b^{low} population is identified as macrophages. This raises the question of how this particular plot is observed in the absence of UPEC infection.

As a general comment for ALL FLOW CYTOMETRY: the axis tick values need to be added back into ALL FIGURES containing flow cytometry. Some of the positive populations appear to be gated very close to what I assume is the zero tick on both the x and y axes, which makes the flow cytometry data questionable, although the separation of 'positive populations' does appear distinct. Showing the full gating strategies will help clear this issue up. Further, replicate values for the flow cytometry data can be graphed, and statistical analysis performed, to add some quantitative data to the argument being made, rather than heavily relying on representative and qualitative data.

Figure 4:

In Figure 4F, the individual data points are not shown, which limits the ability to assess the variability and significance of the results. Y axis label is missing

What is the difference in genes displayed in the Venn diagram vs the heatmap, is the Venn diagram the total number of genes detected whereas the heatmap is only the significantly dysregulated genes? If so, please make this clear in the figure legend and results. If not, and the UP and Down represent significantly altered genes, why was the selection displayed in the heatmap chosen?

The pooling statement in the legend needs to be clarified as per previous comments. A brief overview of the analytical pipeline needs to be provided. E.g. DEGs were identified using Heatmap was generated using in the legend also. The program used for generating graphs and statistics needs to be added to every figure legend.

Figure 5: changes as per previous comments. Additionally, 21% interferon gamma expression is quite high, as IFN γ is de novo assembled after stimulation. There does not appear to be clear separation between the positive and negative cells in this figure for most markers, please amend these plots as per previous comments and display a gating strategy.

Figure 6/7: comments as per previous

GENERAL COMMENTS ON DATA PRESENTATION AND STATISTICS:

The authors have used SEM, which should only be used in population statistics; here standard deviation (SD) would be appropriate.

Some graphs, or even some bars within graphs, have bidirectional error bars and some only have unidirectional error bars, and this needs to be made consistent, preferably with bidirectional bars.

Many of the axis labels are not descriptive enough, lack units, or are not labelled at all (Figure 4, which also lacks individual data points in F).

Additionally, many of the graphs throughout the manuscript that have been given * or ** have a great deal of overlapping data between compared groups. This makes the statistical analysis seem less valid. For my own interest, I replicated Figure 6F, in Prism, as indicated. I ran normality tests, and additional outlier analysis as the CXCL10 data has a lot of spread. I could not generate the same statistical significance that the authors have found, as expected with data that overlaps as much as seen here. A review of the data presentation, tests used, and significance outcome is required.

DISCUSSION: Similar comments to the introduction about fact-checking some statements in the opening paragraph of the discussion.

The limitations of the study need to be discussed. Injecting UPEC into the testes does not represent a natural method of infection, which would presumably be by ascending infection from the penile urethra. The testes and sperm health are susceptible to damage through systemic inflammation, much of recent COVID-19 literature displays this well.

MATERIALS AND METHODS:

The animal ethics committee that provided approval for this study needs to be named, and the approval number provided.

Original publications showing the generation and validation of various mouse strains needs to be included.

"Experimental orchitis induced by UPEC in male mice was improved on previously described method(Jing, Cao et al., 2021a).

Briefly, the mice were anesthetized and the testes and vas deferens were exposed. Approximately 1×10^5 colony-forming units (CFU) of bacteria in 5 μ l PBS were injected into the upper pole of the testis, and 5 μ l PBS was injected into the sham group mice. Then the vas deferens were ligated to prevent the spread of infection." - This part of the methods requires more details, e.g. type of anaesthesia given, details of 'improvements' made, surgery details for vas ligation. Additionally, there seems to have been invasive surgery performed here, so details for pain relief and any other measures to ensure the health of the mice should be provided.

Product details, e.g. MDF, are missing in multiple places, these need to be provided.

As a service to authors, EMBO provides authors with the possibility to transfer a manuscript that one journal cannot offer to publish to another EMBO publication. The full manuscript and if applicable, reviewers reports are automatically sent to the receiving journal to allow for fast handling and a prompt decision on your manuscript. For more details of this service, and to transfer your manuscript to another EMBO title please click on Link Not Available

Dear Dr. Zeljko Durdevic

Re: Appeal of Decision relating to EMM-2023-18267

I am writing to respectfully appeal the decision to reject the above manuscript entitled “**Gasdermin D in macrophages drives orchitis by regulating inflammation and antigen presentation processes**”. On behalf all of the authors, we believe that the decision is too severe, given the Reviewer comments. In particular, Reviewer 1 provided positive feedback on our study, stating, “*The paper deals with a topic that is relevant to this journal. In general, the paper is completely structured and the logic is clear*”. Additionally, Reviewer 1 mainly suggested improvement regarding the arrangement of graph and text annotation, which we fully prepared to address.

Specially, we feel regretful that the decision may have been predominantly based on the Reviewer 2. However, many questions raised by reviewer 2 due to inappropriate data presentation or insufficiently clear description, which led to misunderstanding.

For example, the following comments are typical of Reviewer 2 in their misunderstanding because of inappropriate data presentation: “*the axis tick values need to be added back into ALL FIGURES containing flow cytometry. Some of the positive populations appear to be gated very close to what I assume is the zero tick on both the x and y axes, which makes the flow cytometry data questionable, although the separation of 'positive populations' does appear distinct. Showing the full gating strategies will help clear this issue up. Further, replicate values for the flow cytometry data can be graphed, and statistical analysis performed, to add some quantitative data to the argument being made, rather than heavily relying on representative and qualitative data*”.

In response to this comment, we apologize for not showing flow cytometry data more clearly. The reason why positive populations appear to be gated very close to the zero tick is that the scale of x and y axes were Biex mode. In this mode, the position closed to zero tick does not necessarily correspond 10^1 , but it could be 10^3 (see representative picture below). We will include the axis values into flow cytometry data and show the full gating strategies in the revised manuscript. Furthermore, every flow cytometry data in manuscript showed corresponding statistical analysis graph, some of which were placed in the supplementary figure, but the corresponding statistical results were in the main figure. We apologize for not describing these data more clearly, and we will describe these legends with more clarity in the revised manuscript. Finally, we can provide all raw data of flow cytometry if required.

For example, the following comments are typical of Reviewer 2 in their misunderstanding because of insufficiently clear description: “*Additionally, many of the graphs throughout the manuscript*

*that have been given * or ** have a great deal of overlapping data between compared groups. This makes the statistical analysis is incorrect. For my own interest, I replicated Figure 6F, in Prism, as indicated. I ran normality tests, and additional outlier analysis as the CXCL10 data has a lot of spread. I could not generate the same statistical significance that the authors have found, as expected with data that overlaps as much as seen here. A review of the data presentation, tests used, and significance outcome is required”.*

In response to this comment, we apologize for not describing the data statistical analysis. In fact, we have applied all data to Shapiro–Wilk test to assess data distribution. For data with normal distribution, comparisons were performed using the unpaired t-test or multiple t-tests. Continuous variables data with non-normal distributions were analyzed by non-parametric tests such as the Mann-Whitney test. However, we unintentionally omitted this information from the manuscript. Indeed, CXCL10 data did not exhibit normal distribution, and we performed the Mann-Whitney test for statistical analysis, yielding a p-value of 0.0148 (*), indicating statistical significance. Regrettably, the figure legend of CXCL10 data incorrectly stated the use of t-test. We will correct these mistakes in the revised manuscript.

As another example: *“FOR IMMUNOHISTOCHEMISTRY: all of the images in this manuscript show quite large gaps between tubules and where the interstitial space should be. This is resultant from an error in fixation or tissue processing and the structure of the tissue is not intact. I would recommend repeating these experiments to obtain higher quality staining and imaging. If it cannot be repeated, I recommend taking more images to see whether better imaging can be obtained, or removing this from the manuscript. If the authors feel the IHC is essential to the manuscript, then perhaps removing it to supplementary would be more appropriate. In any case, quantification of positive staining should be performed for all figures”.*

In response to this comment, we would like to clarify that we performed HE and immunofluorescent staining but not immunohistochemistry (IHC).

For HE staining, we would like to highlight that a large proportion of images in our manuscript were testicular staining from **orchitic** mice. Under inflammatory conditions, seminiferous tubules are destroyed and interstitial spaces between seminiferous tubules were enlarged, which is consistent with other studies[1, 2]. Furthermore, the images of HE staining from no-treated ASC and GSDMD knockout mice showed that gaps between seminiferous tubules were large in Fig S2B, which may be related to the smaller size of testicular organ in knockout mice. We speculate that the diameter of seminiferous tubules in knockout mice is decreased, leading to an increase in interstitial spaces. Finally, in Fig S1B and Fig S2B, the images of HE staining from sham-operated or no-treated mice showed that the interstitial spaces between seminiferous tubules were normal, similar to previous studies[1, 3].

For immunofluorescent staining, we used PFA (paraformaldehyde) fixation solution, which has been reported to result in worse morphologic details in testes[3]. However, due to the need for GSDMD/NLRP3/F4/80 antibody staining, we had to use this fixation solution. This might have led to large gaps between the seminiferous tubules in immunofluorescence images. Moreover, we will include the quantified relative integral optical density (IOD) of NLRP3, CASPASE1, and GSDMD, and quantified number of GSDMD-positive and F4/80-positive cells. Finally, we believe that immunofluorescence data, together with a comprehensive body of data from immunoblot, and public databases, are solid and in support of our conclusions.

Some questions raised by reviewer 2 due to our oversights or minor mistake when prepared the manuscript. For example, “*some bars within graphs, have bidirectional error bars and some only have unidirectional error bars*”, which we are in position to fully address all of these points.

We acknowledge the importance of the reviewer feedback in enhancing our manuscript and ensuring its quality. We sincerely appreciate the time and effort invested by the reviewers in providing constructive comments. Specifically, we plan to address the reviewer's suggestion to improve the order of graphs and text annotations promptly.

In conclusion, considering the relevance of our study to your journal's scope, the positive feedback provided by Reviewer 1, and Reviewer 2's misunderstanding because of our inappropriate data presentation and insufficiently clear description, we kindly request the opportunity to submit a revised version of our manuscript for further evaluation. We believe that with the proposed revisions, our manuscript will meet the standards and expectations of the journal.

Thank you for your reconsideration of our appeal. We look forward to the possibility of resubmitting our revised manuscript to contribute to the scientific discourse in your esteemed journal.

Responses to the individual points raised by each of the Reviewers are attached.

Kind Regards

Shuo Yang

Corresponding Author

***** Reviewer's comments *****

Referee #1 (Comments on Novelty/Model System for Author):

The model system of this paper is correct adequate.

Referee #1 (Remarks for Author):

Summary:

The manuscript provides the first demonstration of GSDMD's role in driving UPEC-induced orchitis and detailed delineation of the underlying mechanism. In addition, it was found that the administration of GSDMD inhibitors conferred the mice protection against UPEC infection induced orchitis. These results suggested that targeting GSDMD may be an efficient therapeutic target to treat UPEC-induced acute orchitis and chronic autoimmune orchitis. It provides a promising transformation therapy for bacterial orchitis.

General remarks:

The paper deals with a topic that is relevant to this journal. In general, the paper is completely structured and the logic is clear. However, I have some critical remarks for the authors to consider before being able to recommend the paper be accepted.

Response: We welcome the Reviewer 1's positive comments on our study "*The paper deals with a topic that is relevant to this journal. In general, the paper is completely structured and the logic is clear*". We would be happy to address the specific points raised by this Reviewer to strengthen our study.

Major improvement suggestions:

1. The layout of illustrations in the article needs to be improved:

(1) The boundary between B/C in Figure S1, E/D in Figure 3 and C/D in Figure S3 is not clear;

Response : We apologize for not showing these data more clearly. We will label these data with more clarity in the revised manuscript.

(2) Figure 2 is slightly crowded;

Response : In light of this suggestion, we will move the results of QPCR to supplementary figure in the revised manuscript.

(3) In the "GSDMD in testicular macrophages promotes UPEC-induced acute orchitis" part, some experimental results are scattered in two figures, the author may consider merging them into one, such as "(Fig 3E, Fig S3A)"; "(Fig 3H, I, Fig S3B)" and "(FIG3J, 3K, FIG3B)".

Response : In light of this suggestion, we will rearrange the results of Figure3 and Figure S3 in the

revised manuscript.

2. Some elements in the figure are wrong or should be explained in detail:

(1) The upper arrow of the "UPEC-DAPI" group in Figure 3-B is not pointed accurately;

Response : We apologize for this oversight, and we will omit arrow from the "UPEC-DAPI" group in the revised manuscript.

(2) "WT" appears in both Figure 1 and Figure 2, however, "WT" in Figure 1 refers to wild-type mice, and "WT" in Figure 2 refers to wild-type mice infected by UPEC;

Response : We apologize for not labeling these Figures more clearly, and we will change "WT" in Figure2 into "WT+UPEC" in the revised manuscript.

(3) The abbreviations in Figure 7-A should be explained in detail in figure legends, such as "TH" and "PTX".

Response : In light of this suggestion, we will add the full name of the abbreviations in Figure legends. For example, testicular homogenate (TH), pertussis toxin (PTX).

3. "GSDMD deficiency in macrophages induces inflammation and damaged spermatogenesis during acute and chronic orchitis" in "Abstract" is ideologically contradictory to the full text.

Response : We apologize for this oversight, and we will omit " deficiency " from this sentence in the revised manuscript.

4. It is more logical to consider switching the order of the "GSDMD deficiency controls UPEC-induced acute orchitis" part and the "GSDMD in testicular macrophages promotes UPEC-induced acute orchitis" part.

Response : We firstly observed the phenotype of GSDMD knockout mice, and then we explored the specific cell type in which GSDMD functions as a positive regulator in UPEC-induced orchitis. Therefore, the order of article is more logical.

5. Why choose "CXCL10" as the typical index of pro-inflammatory chemokines in the "GSDMD deficiency controls UPEC-induced acute orchitis" part?

Response : CXCL10 is categorized as an inflammatory chemokine, and it induces "homing" functions to chemoattract CXCR3⁺ cells, including macrophages, monocytes, and activated T lymphocytes toward infected and inflamed areas. It has been well reported that alterations in CXCL10 expression levels are associated with infectious diseases and autoimmune diseases[4, 5]. Furthermore, CXCL10 is one of major chemokines expressed by testicular macrophages when the testes are challenged by pathogens[6]. Taken together, we chosen CXCL10 as the typical index of pro-inflammatory chemokines.

6. What is the difference between the "Gsdmd^{-/-} mice" in "GSDMD deficiency controls UPEC-induced acute orchitis" part and the "Gsdmd^{fl/fl}Cx3cr1-cre mice" in "GSDMD in testicular macrophages promotes UPEC-induced acute orchitis" part?

Response :Gsdmd^{-/-} mice are conventional knockout mice where GSDMD has been deleted in both immune and non-immune cells. We firstly observed the phenotype of GSDMD knockout mice, and then we explored the specific cell type in which GSDMD functions as a positive regulator in UPEC-induced orchitis. To this end, we analyzed GSDMD expression in different cell type of testes, and found that GSDMD is predominantly expressed in testicular macrophages under physiological state. Upon UPEC treatment, GSDMD expression significantly increased and almost co-localized with macrophages. Therefore, we hypothesized that GSDMD in macrophages promotes orchitis. To validate this hypothesis, we used macrophage-specific GSDMD knockout mice (Gsdmd^{fl/fl}Cx3cr1-cre) and found that macrophage-specific GSDMD knockout mice showed the similar phenotype with conventional knockout mice.

7. Please explain the function of "LPS plus nigericin" in the "GSDMD boosts T cell response by enhancing antigen presentation of macrophages" part.

Response :The function of "LPS plus nigericin" was to activate NLRP3 inflammasome, leading to the activation of GSDMD and subsequently pyroptosis. UPEC is a Gram-negative bacterium that releases LPS during infection and can activate NLRP3 and pyroptosis, and thus using "LPS plus nigericin" was to mimic UPEC infection activating GSDMD and better demonstrated the function GSDMD-mediated pyroptosis in enhancing antigen presentation to boost T cell response.

8. In the "GSDMD in macrophages aggravates chronic autoimmune orchitis during EAO development" part, why are the testicular size and testis-body weight ratio of GSDMD deficient mice larger, which indicates that their testicular degeneration has been improved?

Response : Unlike an acute UPEC infection model, EAO is a chronic autoimmune orchitis, characteristic of the pathological testicular atrophy. Thus, the reason why the testes in Gsdmd^{fl/fl}Cx3cr1-cre mice showed larger is that GSDMD deficiency in macrophage impaired CD8⁺ T cells response by preventing the antigen presentation capacities, thereby attenuating the impairment of testicular tissue and atrophy.

Referee #2 (Comments on Novelty/Model System for Author):

FOR IMMUNOHISTOCHEMISTRY: all of the images in this manuscript show quite large gaps between tubules and where the interstitial space should be. This is resultant from an error in fixation or tissue processing and the structure of the tissue is not intact. I would recommend

repeating these experiments to obtain higher quality staining and imaging. If it cannot be repeated, I recommend taking more images to see whether better imaging can be obtained, or removing this from the manuscript. If the authors feel the IHC is essential to the manuscript, then perhaps removing it to supplementary would be more appropriate. In any case, quantification of positive staining should be performed for all figures.

Response : In response to this comment, we would like to clarify that we performed HE and immunofluorescent staining but not immunohistochemistry (IHC).

For HE staining, we would like to highlight that a large proportion of images in our manuscript were testicular staining from **orchitic** mice. Under inflammatory conditions, seminiferous tubules are destroyed and interstitial spaces between seminiferous tubules were enlarged, which is consistent with other studies[1, 2]. Furthermore, the images of HE staining from no-treated ASC and GSDMD knockout mice showed that gaps between seminiferous tubules were large in Fig S2B, which may be related to the smaller size of testicular organ in knockout mice. We speculate that the diameter of seminiferous tubules in knockout mice is decreased, leading to an increase in interstitial spaces. Finally, in Fig S1B and Fig S2B, the images of HE staining from sham-operated or no-treated mice showed that the interstitial spaces between seminiferous tubules were normal, similar to previous studies[1, 3].

For immunofluorescent staining, we used PFA (paraformaldehyde) fixation solution, which has been reported to result in worse morphologic details in testes[3]. However, due to the need for GSDMD/NLRP3/F4/80 antibody staining, we had to use this fixation solution. This might have led to large gaps between the seminiferous tubules in immunofluorescence images. Moreover, we will include the quantified relative integral optical density (IOD) of NLRP3, CASPASE1, and GSDMD, and quantified number of GSDMD-positive and F4/80-positive cells. Finally, we believe that immunofluorescence data, together with a comprehensive body of data from immunoblot, and public databases, are solid and in support of our conclusions.

FOR FLOW CYTOMETRY: the axis tick values need to be added back into ALL FIGURES containing flow cytometry. Some of the positive populations appear to be gated very close to what I assume is the zero tick on both the x and y axes, which makes the flow cytometry data questionable, although the separation of 'positive populations' does appear distinct. Showing the full gating strategies will help clear this issue up. Further, replicate values for the flow cytometry data can be graphed, and statistical analysis performed, to add some quantitative data to the argument being made, rather than heavily relying on representative and qualitative data.

Response : We apologize for not showing flow cytometry data more clearly. The reason why positive populations appear to be gated very close to the zero tick is that the scale of x and y axes were Biex mode. In this mode, the position closed to zero tick does not necessarily correspond 10^1 , but it could be 10^3 (see representative picture below). We will include the axis values into flow cytometry data and show the full gating strategies in the revised manuscript. Furthermore, every flow cytometry data in manuscript showed corresponding statistical analysis graph, some of which were placed in the supplementary figure, but the corresponding statistical results were in the main figure. We apologize for not describing these data more clearly, and we will describe these legends with more clarity in the revised manuscript. Finally, we can provide all raw data of flow cytometry

if required.

FOR DATA PRESENTATION AND STATISTICS:

The authors have used SEM, which should only be used in population statistics; here standard deviation (SD) would be appropriate.

Response : In light of this suggestion, we will change SEM into SD in the revised manuscript.

Some graphs, or even some bars within graphs, have bidirectional error bars and some only have unidirectional error bars, and this needs to be made consistent, preferably with bidirectional bars.

Response : In light of this suggestion, we will change all error bars into bidirectional error bars in the revised manuscript.

Many of the axis labels are not descriptive enough, lack units, or are not labelled at all (Figure 4, which also lacks individual data points in F).

Response : We apologize for this oversight. We will include “ $\mu\text{g/g}$ ” in graph of testis-body weight ratio, add “relative expression” in Figure 4F, and add “Z score” in Figure 4G. We will also change histogram into dot histogram in Figure 4F.

Additionally, many of the graphs throughout the manuscript that have been given * or ** have a great deal of overlapping data between compared groups. This makes the statistical analysis incorrect. For my own interest, I replicated Figure 6F, in Prism, as indicated. I ran normality tests, and additional outlier analysis as the CXCL10 data has a lot of spread. I could not generate the same statistical significance that the authors have found, as expected with data that overlaps as much as seen here. A review of the data presentation, tests used, and significance outcome is required.

Response : We apologize for not describing the data statistical analysis. In fact, we have applied all data to Shapiro–Wilk test to assess data distribution. For data with normal distribution, comparisons were performed using the unpaired t-test or multiple t-tests. Continuous variables data with non-normal distributions were analyzed by non-parametric tests such as the

Mann-Whitney test. However, we unintentionally omitted this information from the manuscript. Indeed, CXCL10 data did not exhibit normal distribution, and we performed the Mann-Whitney test for statistical analysis, yielding a p-value of 0.0148 (*), indicating statistical significance. Regrettably, the figure legend of CXCL10 data incorrectly stated the use of t-test. We will correct these mistakes in the revised manuscript.

Referee #2 (Remarks for Author):

Summary: This manuscript investigates the role of GSDMD and GSDMD-depletion in UPEC-mediated orchitis. While this is an interesting topic and is generally a well thought out study, the manuscript contains many grammatical/syntax errors and would benefit from a thorough proof read, and the manuscript contains numerous experimental flaws.

Response : We welcome the Reviewer's positive comments on our study "this is an interesting topic and is generally a well thought out study". We are also thankful for the Reviewer's meaningful suggestion, and happy to address the points raised by this Reviewer.

In light of the Reviewer's suggestion, we will recruit a native English speaker to edit the manuscript to ensure the grammar and language are corrected.

In response to experimental flaws, we believe that misunderstanding may have been caused by inadequate description in the manuscript. We have detailed our responses to the individual points raised by each the Reviewers.

PAPER EXPLAINED and ABSTRACT:

This reviewer feels that the authors should rework the latter half of the abstract to reflect any changes made consequent to review, being careful not to oversell the mechanistic aspect of the study. Similarly, the authors should ensure they do not overreach for the conclusion that GSDMD makes a feasible therapeutic target for treating orchitis, as while the study may provide a foundation for that study, the current study does not investigate the necessary

Response : In light of this suggestion, we will rewrite abstract in the revised manuscript.

INTRODUCTION:

The authors indicate that 'approximately 60%' of cases of male infertility related to infection and inflammation are attributed to UPEC. However, the literature is much more complex than this, and rates vary greatly between studies. This could be due to type of testing, demographics, geographics, etc. but it is insufficient to cite only one article, from 2008, as the foundation for this statement. Please provide an unbiased and up-to-date comment on the role of UPEC in male infertility.

Further, the reference given (Wiles, Kulesus et al., 2008) does not contain the information given. This reference does not mention a 60% value, infertility, or orchitis that this reviewer could find after skimming the article and searching for keywords.

<https://academic.oup.com/molehr/article/26/4/215/5721557> this article cites a 60% value on a similar topic, but it is not the same statistic that you have given.

A thorough review of all references is required.

Response :We apologize for this oversight, 'approximately 60%' came from the previous study [7].

It said that approximately 15% of male infertility cases are related to the inflammation of the reproductive system, and approximately 60% of such cases are caused by uropathogenic Escherichia Coli (UPEC), which cited one article from Wiles, Kulesus et al [8]. Wiles, Kulesus et al said that these bacteria are the primary cause of community-acquired urinary tract infections (UTI) (70–95%) and a large portion of nosocomial UTIs (50%), accounting for substantial medical costs and morbidity worldwide. We are so sorry that we have summarized the content of this paper incorrectly, and we will change this sentence into “UPEC can be isolated from urine and semen samples of 50 to 95% of genitourinary infectious patients”.

In light of this suggestion, we will check thoroughly all references.

At the start of the second page of the introduction, the authors indicate that during orchitis, testicular macrophages 'transform' into pro-inflammatory macrophages that highly express proinflammatory cytokines. This is inaccurate, and the authors need to review the appropriate literature to understand the origin of inflammatory cytokines in the testes. This should include the role of recruited monocytes as key proinflammatory mediators.

Response : Indeed, the origin of inflammatory cytokines in the testes are mainly from circulating monocytes and macrophages differentiated from them, with a smaller portion originating from resident pro-inflammatory testicular macrophages. Thus, we will correct this description in the revised manuscript.

Similarly, the APC ability of testicular macrophages is different to standard macrophages in dogma, so please review your statements and conduct a more thorough update of the literature to understand these two topics. Please ensure that the inflammation pathway information is specific to testicular macrophages in line with your updated reading.

Response : Thank you for your suggestion. Indeed, the APC ability of testicular macrophages is not entirely the same as standard macrophages in dogma. Although it is tempting to speculate that peritubular testicular macrophages might have the ability of presentation of antigens to T cells, based on their high levels of MHCII[9], but our statements are also inappropriate. We will modify these statements in the revised manuscript.

RESULTS:

GENERAL NOTE FOR IMMUNOHISTOCHEMISTRY: all of the images in this manuscript

show quite large gaps between tubules and where the interstitial space should be. This seems to be resultant from an error in fixation or tissue processing and the structure of the tissue no longer appears intact. If possible, this reviewer would recommend repeating these experiments to obtain higher quality staining and imaging. If it cannot be repeated, I recommend taking more images to see whether better imaging can be obtained, or removing this from the manuscript. If the authors feel the IHC is essential to the manuscript, then perhaps removing it to supplementary would be more appropriate. In any case, quantification of positive staining should be performed for all figures. The authors rely on staining to draw many conclusions, therefore it would be appropriate to have quantitative data rather than qualitative, as is currently provided.

Response : In response to this comment, we would like to clarify that we performed HE and immunofluorescent staining but not immunohistochemistry (IHC).

For HE staining, we would like to highlight that a large proportion of images in our manuscript were testicular staining from **orchitic** mice. Under inflammatory conditions, seminiferous tubules are destroyed and interstitial spaces between seminiferous tubules were enlarged, which is consistent with other studies[1, 2]. Furthermore, the images of HE staining from no-treated ASC and GSDMD knockout mice showed that gaps between seminiferous tubules were large in Fig S2B, which may be related to the smaller size of testicular organ in knockout mice. We speculate that the diameter of seminiferous tubules in knockout mice is decreased, leading to an increase in interstitial spaces. Finally, in Fig S1B and Fig S2B, the images of HE staining from sham-operated or no-treated mice showed that the interstitial spaces between seminiferous tubules were normal, similar to previous studies[1, 3].

For immunofluorescent staining, we used PFA (paraformaldehyde) fixation solution, which has been reported to result in worse morphologic details in testes[3]. However, due to the need for GSDMD/NLRP3/F4/80 antibody staining, we had to use this fixation solution. This might have led to large gaps between the seminiferous tubules in immunofluorescence images. Moreover, we will include the quantified relative integral optical density (IOD) of NLRP3, CASPASE1, and GSDMD, and quantified number of GSDMD-positive and F4/80-positive cells. Finally, we believe that immunofluorescence data, together with a comprehensive body of data from immunoblot, and public databases, are solid and in support of our conclusions.

Finally, please specify how many fields of view your images are representative of in the figure legend or methods, once quantification as been completed.

Response : We apologize for not providing the detailed HE analysis. For HE analysis, we applied the categorization of Johnsen criteria method to assess seminiferous tubules and quantify seminiferous tubules as previously described [10, 11]. In brief, after cutting the tissue at the largest cross-sectional position of testis, we start sectioning from this location for slide preparation. In each mouse, one microscopy slide would be evaluated if 150-250 seminiferous tubules in a section were visible at 40X magnification. Then, the microscopy slides that met the criteria were assessed by evaluating all seminiferous tubules under optical microscope (magnification, 100X). Approximately, 10-15 fields of view in one section contain 150-250 seminiferous tubules. We will provide the detailed analysis in the revised manuscript.

Figure 1:

The Figure is well laid out and the panels are clear, there are several issues however.

In Figure 1B, Z-score should be Z-score.

Response : We apologize for this mistake, and we will correct this word in Fig 1B and Fig S1D.

In Figure 1c, multiple bands are observed in the Western blot of Pro CASP11 and others, which is fine, but the blots are cropped, which means we cannot see other potential bands and this is not acceptable by the journal and today's standards. Either in the figure or as a supplementary, full sized, uncropped blots must be provided.

Response : Procaspase-11 has two isoforms 43 and 38 kDa, and we apologize for the incomplete showing of Procaspase-11 bands. The correction will be included and the specific target bands will be indicated by arrows in the revised manuscript. Additionally, we now have provided full sized and uncropped blots (see figure below).

The immunostaining images in Figure 1D appear to be out of focus in panels for NLRP3 and GSDMD.

Response : Indeed, the immunostaining image for NLRP3 appear unclear because the image was not saved properly but not out of focus. We will replace the image in the revised manuscript. However, the immunostaining image for GSDMD appear fine.

The Sham-GSDMD staining in Figure 1D vs Figure 3B is quite different, with quite a lot more staining present in the latter. What is the reason for this? Have the authors ensured that, in both cases, representative images have been provided? If so, why are these so different?

Response : The difference of Sham-GSDMD staining between Figure 1D and Figure 3B was due to variations in exposure time when taking a picture. To address the concerns, we will include the quantitative analysis of 3-4 mice per group from three independent experiments in the revised manuscript.

In the figure legend, please clarify whether n = 5 is the total number of mice used, or whether the three independent experiments conducted were each n = 5, giving a total of n = 15.

Response : We will clarify that N=5 is the total number of mice used in the revised manuscript.

Please define all of the abbreviations used in the figure, in the figure legend. E.g. CASP1.

Response : We will define all of the abbreviations used in the figure, in the figure legend. For example, testicular homogenate (TH), pertussis toxin (PTX), caspase1(CASP1), caspase11(CASP11).

Figure 2:

The figure legend indicates that Sham as well as UPEC infected mice were assessed, however, I cannot see any data from Sham mice in this figure. Please address this, most appropriately would be to include the Sham data in this figure for side-by-side comparisons of KOs.

Response : We apologize for this oversight, and we will remove the describing “Sham” from the figure legend.

For the testis/Body weight graph in A, please provide a unit of measure, e.g. mg.

In the figure legend, please define all abbreviations, e.g. SCO, SG, M/mL

Response : In light of this suggestion, we will include “ $\mu\text{g/g}$ ” in graph of testis-body weight ratio, and define all of the abbreviations used in the figure, in the figure legend. ES= elongated spermatids, RS = round spermatids, PSc= pachytene spermatocytes, SG=spermatogonia, SCO= Sertoli-cell only.

Authors state that the Johnsen Criteria method was used for assessing the seminiferous tubules in C, however, my understanding is that this method results in a score from 1 to 10, not a percentage, so please explain clearly in the methods/legend how this percentage has been obtained. Please also clarify how tubules were staged, in order to obtain the percentages for round and elongated spermatocytes, for example, as I currently cannot see if this one performed, to standardize the counting (i.e. the number of these types of cells is dependent on the stage, was this accounted for).

Response : We apologize for not describing the quantified method more clearly. In fact, we only utilized Johnsen criteria qualitative method, which categorized seminiferous tubules into different types including elongated spermatids (ES), round spermatids (RS), pachytene spermatocytes (PSc),

spermatogonia (SG), and Sertoli-cell only (SCO), based on the presence or absence of the main cell types arranged in the order of maturity[12]. Next, we applied the categorization of Johnsen criteria method to assess seminiferous tubules and quantify seminiferous tubules as previously described [10, 11]. Briefly, 150-250 seminiferous tubules per mouse were evaluated, and the most advanced germ cell type detectable in each tubule cross-section were recorded, and then the percentage of tubule cross-sections showing the respective germ cell stage were finally documented. For example, a single slide includes total 200 seminiferous tubules, and among them 10 tubules only containing Sertoli cells were recorded. The frequency of tubules with Sertoli cells only would be 5%. We will provide the more detailed information in the revised manuscript.

In Figure 2/S1, there is a white line/spot observed on the testis, is this just light reflecting on damp tissue at the time the photograph was taken, or something else that the authors have annotated on the image?

Response : A white line/spot observed on the testis is just light on damp tissue when taking the photograph.

For D, the image are really too small to be useful. Suggest moving these to supplementary if the authors feel they are necessary to the manuscript. Alternatively, enlarge the images and condense some of the graphical information. Condensing the graphical information would also help improve the readability of the figure.

Response : In light of this suggestion, we will move the results of QPCR to supplementary figure in the revised manuscript.

For E and F, please clarify your n values. Is this n = 2 mice repeated 3 times, to give n = 6 in total? By pooled data, do you mean to say that your triplicate independent experiments are shown on a single graph, or do you mean that the actual samples were pooled?

Response : We will clarify that N=5 is the total number of mice used in the revised manuscript.

For pooled data, we showed triplicate independent experiments on a single graph.

In Figure S2A, the color of the testis appears different in *Gsdmd*^{-/-} and *Asc*^{-/-} mice. The reason for this color difference could be expanded upon if space permits, or if not relevant, please explain in response.

Response : The reason for difference color of the testes from *Gsdmd*^{-/-} and *Asc*^{-/-} mice may be attributed to pathological changes induced by knockout, which was an interesting phenotype. However, this fascinating finding requires further exploration to elucidate the underlying mechanism.

In Figure S2B, the H&E staining of *Gsdmd*^{-/-} and *Asc*^{-/-} mice shows an increase in interstitial space. The authors claim that the testis size is reduced in these mice, but it is unclear how they

relate the testis size to the increase in interstitial space.

Response : We speculate that the diameter of seminiferous tubules in *Gsdmd*^{-/-} and *Asc*^{-/-} mice is decreased, leading to an increase in interstitial spaces. However, the specific mechanisms by which ASC and GSDMD regulate the diameter of seminiferous tubules and the size of testes observation requires further exploration.

Figure 3:

B - comments as above about quality of sections and destruction of tissue structure. These images are not appropriate to draw conclusions from.

Response : For immunofluorescent staining, we used PFA (paraformaldehyde) fixation solution, which has been reported to result in worse morphologic details in testes. However, due to the need for GSDMD/NLRP3/F4/80 antibody staining, we had to use this fixation solution. This may have led to large gaps between the seminiferous tubules in immunofluorescence images. Moreover, we will include the quantified relative integral optical density (IOD) of NLRP3, CASPASE1, and GSDMD, and quantified number of GSDMD-positive and F4/80-positive cells. Finally, we believe that immunofluorescence data, together with a comprehensive body of data from immunoblot, and public databases, are solid and in support of our conclusions.

*The authors claim to have generated macrophage conditional knockout mice of GSDMD (*Gsdmd*^{fl/fl}*Cx3cr1-cre*), but they have not shown any data demonstrating GSDMD deletion in macrophages.*

Response The GSDMD deletion of *Gsdmd*^{fl/fl}*Cx3cr1-cre* mice had been demonstrated using single-cell RNA-seq in our previous study[13].

The decrease in immune cells after UPEC infection is shown in *Gsdmd*^{fl/fl}*Cx3cr1-cre* mice, but the immune cell population in *Gsdmd*^{fl/fl}*Cx3cr1-cre* mice under physiological conditions is not addressed.

Response : We will detect the immune cells in *Cx3cr1-cre* and *Gsdmd*^{fl/fl}*Cx3cr1-cre* mice under physiological conditions in the revised manuscript.

In Figure 3I, the y-axis represents immune cells in number (10⁴) in CD45+ cells. However, it is generally known that macrophages contribute around 60% of the immune cell population. This information is not reflected in the graph. Please justify why your results are widely different to existing literature.

Response : We agree that macrophages contribute around 60% of the immune cell population in testes under physiological conditions. However, we showed immune cells of testes from UPEC-induced orchitic mice in Figure 3I, and it is generally known that the composition of testicular immune cells changes dramatically under inflammatory condition, because a large number of immune cells migrate into inflamed tissue.

In Figure S3B, In the gating strategy is not clearly written or shown which parent population were taken.

Response : We will include all gating strategy of flow cytometry in the revised manuscript.

In Figure S3B, the flow cytometry plot depicting macrophages reveals the presence of two distinct populations. It is possible that one of these populations corresponds to infiltrated monocytes rather than true macrophages. Notably, only a small proportion of the F4/80^{high}CD11b^{low} population is identified as macrophages. This raises the question of how this particular plot is observed in the absence of UPEC infection.

Response : In fact, the expression of F4/80 by monocytes is relatively lower than macrophages, however, the two distinct groups of cells showed similar expression of F4/80 but different expression levels of CD11b. Therefore, we speculate that these two groups may represent functionally distinct macrophages. Furthermore, we will explore whether macrophages showed two distinct populations in the absence of UPEC treatment.

As a general comment for ALL FLOW CYTOMETRY: the axis tick values need to be added back into ALL FIGURES containing flow cytometry. Some of the positive populations appear to be gated very close to what I assume is the zero tick on both the x and y axes, which makes the flow cytometry data questionable, although the separation of 'positive populations' does appear distinct. Showing the full gating strategies will help clear this issue up. Further, replicate values for the flow cytometry data can be graphed, and statistical analysis performed, to add some quantitative data to the argument being made, rather than heavily relying on representative and qualitative data.

Response : We apologize for not showing flow cytometry data more clearly. The reason why positive populations appear to be gated very close to the zero tick is that the scale of x and y axes were Biex mode. In this mode, the position closed to zero tick does not necessarily correspond 10^1 tick, but it could be 10^3 tick (see representative picture below). We will include the axis values into flow cytometry data and show the full gating strategies in the revised manuscript. Furthermore, every flow cytometry data in manuscript showed corresponding statistical analysis graph, some of which were placed in the supplementary figure, but the corresponding statistical results were in the main figure. We apologize for not describing these data more clearly, and we will describe these legends with more clarity in the revised manuscript. Finally, we can provide all raw data of flow cytometry if required.

Figure 4:

In Figure 4F, the individual data points are not shown, which limits the ability to assess the variability and significance of the results. Y axis label is missing

Response : We apologize for this oversight. We will add “relative expression”, and change histogram into dot histogram in Figure 4F.

What is the difference in genes displayed in the Venn diagram vs the heatmap, is the Venn diagram the total number of genes detected whereas the heatmap is only the significantly dysregulated genes? If so, please make this clear in the figure legend and results. If not, and the UP and Down represent significantly altered genes, why was the selection displayed in the heatmap chosen?

Response : We apologize for not describing these data more clearly. The Venn diagram showed the total number of genes detected, and the heatmap showed the representative genes of the significantly dysregulated pathways after GSDMD deficiency, and we will clarify these details in the revised manuscript.

The pooling statement in the legend needs to be clarified as per previous comments. A brief overview of the analytical pipeline needs to be provided. E.g. DEGs were identified using Heatmap was generated using in the legend also. The program used for generating graphs and statistics needs to be added to every figure legend.

Response : Thank you for your suggestion. We will modify the text and add details of analytical pipeline, including p values and fold changes in Figure legend.

Figure 5: changes as per previous comments. Additionally, 21% interferon gamma expression is quite high, as IFN γ is de novo assembled after stimulation. There does not appear to be clear separation between the positive and negative cells in this figure for most markers, please amend these plots as per previous comments and display a gating strategy.

Response : We will include all gating strategy of flow cytometry in the revised manuscript. Notably, the proportion of interferon gamma was consistent with previous studies[14, 15].

Figure 6/7: comments as per previous

Response : Thank you for your suggestion, our answers were described above.

GENERAL COMMENTS ON DATA PRESENTATION AND STATISTICS:

The authors have used SEM, which should only be used in population statistics; here standard deviation (SD) would be appropriate.

Response : Thank you for your suggestion, we will change SEM into SD in the revised manuscript.

Some graphs, or even some bars within graphs, have bidirectional error bars and some only have unidirectional error bars, and this needs to be made consistent, preferably with bidirectional bars.

Response : Thank you for your suggestion, we will change all error bars into bidirectional error bars in the revised manuscript.

Many of the axis labels are not descriptive enough, lack units, or are not labelled at all (Figure 4, which also lacks individual data points in F).

Response : We apologize for this oversight. We will include “ $\mu\text{g/g}$ ” in graph of testis-body weight ratio, and add “relative expression” in Figure 4F, and add “Z score” in Figure 4G. We will also change histogram into dot histogram in Figure 4F.

Additionally, many of the graphs throughout the manuscript that have been given * or ** have a great deal of overlapping data between compared groups. This makes the statistical analysis seem less valid. For my own interest, I replicated Figure 6F, in Prism, as indicated. I ran normality tests, and additional outlier analysis as the CXCL10 data has a lot of spread. I could not generate the same statistical significance that the authors have found, as expected with data that overlaps as much as seen here. A review of the data presentation, tests used, and significance outcome is required.

Response : In response to this comment, we apologize for not describing the data statistical analysis. In fact, we have applied all data to Shapiro–Wilk test to assess data distribution. For data with normal distribution, comparisons were performed using the unpaired t-test or multiple t-tests. Continuous variables data with non-normal distributions were analyzed by non-parametric tests such as the Mann-Whitney test. However, we unintentionally omitted this information from the manuscript. Indeed, CXCL10 data did not exhibit normal distribution, and we performed the Mann-Whitney test for statistical analysis, yielding a p-value of 0.0148 (*), indicating statistical significance. Regrettably, the figure legend of CXCL10 data incorrectly stated the use of t-test. We will correct these mistakes in the revised manuscript.

DISCUSSION: Similar comments to the introduction about fact-checking some statements in the opening paragraph of the discussion.

Response : In light of this suggestion, we will modify some statements of discussion in the revised manuscript.

The limitations of the study need to be discussed. Injecting UPEC into the testes does not represent a natural method of infection, which would presumably be by ascending infection from the penile urethra. The testes and sperm health are susceptible to damage through systemic inflammation, much of recent COVID-19 literature displays this well.

Response : Thank you for your suggestion, we will include the limitations of the study about UPEC-induced orchitis mouse model, and we will discuss orchitis induced by systemic infection and inflammation in the revised manuscript. It is worth noting that although UPEC initially causes epididymis by ascending infection from the penile urethra, approximately in 60% of infected patients experience orchitis[10]. Furthermore, fertility impairment following epididymo-orchitis can originate from direct exposure of testicular germ cells to pathogens, toxins or pathogens-induced immune response[10]. Additionally, there are rare studies on the direct effects of UPEC on testis. Finally, in addition to exploring orchitis induced by UPEC, we also investigated autoimmune orchitis. Taken together, we injected UPEC into the testes in UPEC infection model.

MATERIALS AND METHODS:

The animal ethics committee that provided approval for this study needs to be named, and the approval number provided.

Response :All animal procedures were approved by the Ethical Review Committee for Laboratory Animal Welfare of Nanjing Medical University, and approval number was 1705038. We will include this information in the revised manuscript.

Original publications showing the generation and validation of various mouse strains needs to be included.

Response : Thank you for your suggestion, we will include original publications showing the generation and validation of all mouse strains used in our study.

"Experimental orchitis induced by UPEC in male mice was improved on previously described method(Jing, Cao et al., 2021a). Briefly, the mice were anesthetized and the testes and vas deferens were exposed. Approximately 1×10^5 colony-forming units (CFU) of bacteria in 5 μ l PBS were injected into the upper pole of the testis, and 5 μ l PBS was injected into the sham group mice. Then the vas deferens were ligated to prevent the spread of infection." - This part of the methods requires more details, e.g. type of anaesthesia given, details of 'improvements' made, surgery details for vas ligation. Additionally, there seems to have been invasive surgery performed here, so details for pain relief and any other measures to ensure the health of the mice should be provided.

Product details, e.g. MDF, are missing in multiple places, these need to be provided.

Response : The method of anesthesia used in our study was intraperitoneal injection of tribromoethanol (180mg/kg). For vas ligation, the vas deferens were directly tied a knot using surgical sutures. After the wound was sutured, these rats were kept warm until recovery. Furthermore, the wound suture was disinfected with iodine twice daily. We will include these descriptions and products details in the revised manuscript.

Given the above responses and our capacity to address all of the Reviewers comments, we hope you to re-consider your decision to reject the manuscript and permit us to make a re-submission.

Kind Regards

Shuo Yang

Corresponding Author

Reference

1. Wang, M.-W., et al., *Activation of PTH1R alleviates epididymitis and orchitis through Gq and β -arrestin-1 pathways*. Proceedings of the National Academy of Sciences, 2021. **118**(45): p. e2107363118.
2. Wang, M., et al., *Two populations of self-maintaining monocyte-independent macrophages exist in adult epididymis and testis*. Proceedings of the National Academy of Sciences, 2021. **118**(1): p. e2013686117.
3. Wang, H., et al., *Different fixative methods influence histological morphology and TUNEL staining in mouse testes*. Reproductive Toxicology, 2016. **60**: p. 53-61.
4. Liu, M., et al., *CXCL10/IP-10 in infectious diseases pathogenesis and potential therapeutic implications*. Cytokine & growth factor reviews, 2011. **22**(3): p. 121-130.
5. Lee, E.Y., Z.-H. Lee, and Y.W. Song, *CXCL10 and autoimmune diseases*. Autoimmunity reviews, 2009. **8**(5): p. 379-383.
6. Guazzone, V.A., et al., *Cytokines and chemokines in testicular inflammation: a brief review*. Microscopy research and technique, 2009. **72**(8): p. 620-628.
7. Su, Y., et al., *Prokineticin 2 via calcium-sensing receptor activated NLRP3 inflammasome pathway in the testicular macrophages of uropathogenic Escherichia coli-induced orchitis*. Frontiers in Immunology, 2020. **11**: p. 570872.
8. Wiles, T.J., R.R. Kulesus, and M.A. Mulvey, *Origins and virulence mechanisms of*

uropathogenic Escherichia coli. Experimental and molecular pathology, 2008. **85**(1): p. 11-19.

9. Mossadegh-Keller, N., et al., *Developmental origin and maintenance of distinct testicular macrophage populations*. Journal of Experimental Medicine, 2017. **214**(10): p. 2829-2841.
10. Klein, B., et al., *Differential tissue-specific damage caused by bacterial epididymo-orchitis in the mouse*. Molecular human reproduction, 2020. **26**(4): p. 215-227.
11. Rathi, R., et al., *Germ cell fate and seminiferous tubule development in bovine testis xenografts*. Reproduction, 2005. **130**(6): p. 923-929.
12. Johnsen, S.G., *Testicular biopsy score count—a method for registration of spermatogenesis in human testes: normal values and results in 335 hypogonadal males*. Hormone Research in Paediatrics, 1970. **1**(1): p. 2-25.
13. Jiang, Y., et al., *Gasdermin D restricts anti-tumor immunity during PD-L1 checkpoint blockade*. Cell Reports, 2022. **41**(4).
14. Hong, J.Y., et al., *Long-term programming of CD8 T cell immunity by perinatal exposure to glucocorticoids*. Cell, 2020. **180**(5): p. 847-861. e15.
15. Blagih, J., et al., *Cancer-specific loss of p53 leads to a modulation of myeloid and T cell responses*. Cell reports, 2020. **30**(2): p. 481-496. e6.

29th Aug 2023

Dear Prof. Yang,

Thank you for your response to the editorial decision on your manuscript entitled "Gasdermin D in macrophages drives orchitis by regulating inflammation and antigen presentation processes". I have now carefully examined the arguments provided in your letter and discussed them with the other members of our editorial team. Additionally, I have sent your point-by-point response to the referees for the expert opinion.

I am pleased to inform you that we decided to re-consider our initial decision and to invite major revision of your manuscript. Please provide detailed responses to the referee concerns and appropriately amend the manuscript to strengthen main message of the study.

Additional experiments that further strengthen the main conclusions of the study are of course appreciated. We would welcome the submission of a revised version within three months for further consideration. Please let us know if you require longer to complete the revision.

I look forward to receiving your revised manuscript.

Yours sincerely,

Zeljko Durdevic

We require:

- 1) A .docx formatted version of the manuscript text (including legends for main figures, EV figures and tables). Please make sure that the changes are highlighted to be clearly visible.
- 2) Individual production quality figure files as .eps, .tif, .jpg (one file per figure). For guidance, download the 'Figure Guide PDF': (<https://www.embopress.org/page/journal/17574684/authorguide#figureformat>).
- 3) A .docx formatted letter INCLUDING the reviewers' reports and your detailed point-by-point responses to their comments. As part of the EMBO Press transparent editorial process, the point-by-point response is part of the Review Process File (RPF), which will be published alongside your paper.
- 4) A complete author checklist, which you can download from our author guidelines (<https://www.embopress.org/page/journal/17574684/authorguide#submissionofrevisions>). Please insert information in the checklist that is also reflected in the manuscript. The completed author checklist will also be part of the RPF.

6) It is mandatory to include a 'Data Availability' section after the Materials and Methods. Before submitting your revision, primary datasets produced in this study need to be deposited in an appropriate public database, and the accession numbers and database listed under 'Data Availability'. Please remember to provide a reviewer password if the datasets are not yet public (see <https://www.embopress.org/page/journal/17574684/authorguide#dataavailability>).

13) Author contributions: You will be asked to provide CRediT (Contributor Role Taxonomy) terms in the submission system.

These replace a narrative author contribution section in the manuscript.

14) A Conflict of Interest statement should be provided in the main text.

Please note: When submitting your revision you will be prompted to enter your funding and payment information. This will allow Wiley to send you a quote for the article processing charge (APC) in case of acceptance. This quote takes into account any reduction or fee waivers that you may be eligible for. Authors do not need to pay any fees before their manuscript is accepted and transferred to the publisher.

EMBO Press participates in many Publish and Read agreements that allow authors to publish Open Access with reduced/no publication charges. Check your eligibility: <https://authorservices.wiley.com/author-resources/Journal-Authors/open-access/affiliation-policies-payments/index.html>

***** Reviewer's comments *****

Dear Dr. Zeljko Durdevic

Re: Re-Submission of EMM-2023-18267

Please find submitted a revised Manuscript entitled “Gasdermin D in macrophages drives orchitis by regulating inflammation and antigen presentation processes” (EMM-2023-18267) for consideration for publication in the EMBO Molecular Medicine. We appreciate your invitation to revise our manuscript in response to the editor’s and reviewer’s comments and we believe that the revision completes with new data and addresses all of the substantive concerns raised. We appreciate the constructive feedback from each of the editors and reviewers and feel that your comments and suggestions have greatly enhanced the quality of the manuscript. We hope that you now consider the manuscript worthy of publication in EMBO Molecular Medicine.

Detailed responses to the individual points raised by each of the reviewers are outlined below:

***** Reviewer's comments *****

Referee #1 (Comments on Novelty/Model System for Author):

The model system of this paper is correct adequate.

Referee #1 (Remarks for Author):

Summary:

The manuscript provides the first demonstration of GSDMD's role in driving UPEC-induced orchitis and detailed delineation of the underlying mechanism. In addition, it was found that the administration of GSDMD inhibitors conferred the mice protection against UPEC infection induced orchitis. These results suggested that targeting GSDMD may be an efficient therapeutic target to treat UPEC-induced acute orchitis and chronic autoimmune orchitis. It provides a promising transformation therapy for bacterial orchitis.

General remarks:

The paper deals with a topic that is relevant to this journal. In general, the paper is completely structured and the logic is clear. However, I have some critical remarks for the authors to consider before being able to recommend the paper be accepted.

***Response:** We are grateful for the Reviewer 1's positive comments on our study "The paper deals with a topic that is relevant to this journal. In general, the paper is completely structured and the logic is clear". We would be happy to address the specific points raised by this Reviewer to strengthen our study.*

Major improvement suggestions:

1. The layout of illustrations in the article needs to be improved:

(1) The boundary between B/C in Figure S1, E/D in Figure 3 and C/D in Figure S3 is not clear;

***Response :** We apologize for not showing these data more clearly. We have rearranged Figure S1, Figure 3 and Figure s3, and labeled these data with more clarity in the revised manuscript. For example, the new Figure S1 was presented below.*

(2) Figure 2 is slightly crowded;

Response : In light of this suggestion, we have moved representative H&E staining results and some of QPCR and ELISA results to supplementary figure in the revised manuscript.

(3) In the "GSDMD in testicular macrophages promotes UPEC-induced acute orchitis" part, some experimental results are scattered in two figures, the author may consider merging them into one, such as "(Fig 3E, Fig S3A)"; "(Fig 3H, I, Fig S3B)" and "(FIG3J, 3K, FIG3B)".

Response : In light of this suggestion, we have rearranged the results of Figure3 and Figure S3 in the revised manuscript, and put results from the same experiment into one Figure. For example, Fig 3E and Fig S3A have been labelled Fig 3E and Fig 3F, and Fig 3H, I, Fig S3B have been labelled Fig 4D, 4E, 4F in the revised manuscript.

2. Some elements in the figure are wrong or should be explained in detail:

(1) The upper arrow of the "UPEC-DAPI" group in Figure 3-B is not pointed accurately;

Response : We apologize for this oversight, and we have omitted arrow from the "UPEC-DAPI" group in the revised manuscript.

(2) "WT" appears in both Figure 1 and Figure 2, however, "WT" in Figure 1 refers to wild-type mice, and "WT" in Figure 2 refers to wild-type mice infected by UPEC;

Response : We apologize for not labeling these Figures more clearly, and we have changed "WT" in Figure1 into "naïve WT" in the revised manuscript.

(3) The abbreviations in Figure 7-A should be explained in detail in figure legends, such as "TH" and "PTX".

Response : In light of this suggestion, we have added the full name of the abbreviations in Figure legends. For example, testicular homogenate (TH), pertussis toxin (PTX).

3. "GSDMD deficiency in macrophages induces inflammation and damaged spermatogenesis during acute and chronic orchitis" in "Abstract" is ideologically contradictory to the full text.

Response : We apologize for this oversight, and we will omit " deficiency " from this sentence in the revised manuscript.

4. It is more logical to consider switching the order of the "GSDMD deficiency controls UPEC-induced acute orchitis" part and the "GSDMD in testicular macrophages promotes UPEC-induced acute orchitis" part.

Response : We welcome the opportunity to clarify the logic of the order of results. We first observed the phenotype of GSDMD knockout mice, and then we explored the specific cell type in which GSDMD functions as a positive regulator in UPEC-induced orchitis. Therefore, the order of article is more logical.

5. Why choose "CXCL10" as the typical index of pro-inflammatory chemokines in the "GSDMD deficiency controls UPEC-induced acute orchitis" part?

Response : CXCL10 is categorized as an inflammatory chemokine, and it induces "homing" functions to chemoattract CXCR3⁺ cells, including macrophages, monocytes, and activated T lymphocytes toward infected and inflamed areas. It has been well reported that alterations in CXCL10 expression levels are associated with infectious diseases and autoimmune diseases[1, 2]. Furthermore, CXCL10 is one of major chemokines expressed by testicular macrophages when the testes are challenged by pathogens[3]. Taken together, we chosen CXCL10 as the typical index of pro-inflammatory chemokines.

6. What is the difference between the "Gsdmd^{-/-} mice" in "GSDMD deficiency controls UPEC-induced acute orchitis" part and the "Gsdmd^{fl/fl}Cx3cr1-cre mice " in "GSDMD in testicular macrophages promotes UPEC-induced acute orchitis" part?

Response : We welcome the opportunity to clarify the difference between *Gsdmd*^{-/-} mice and result of *Gsdmd*^{fl/fl}*Cx3cr1-cre* mice. *Gsdmd*^{-/-} mice are conventional knockout mice where GSDMD has

been deleted in both immune and non-immune cells. We firstly observed the phenotype of GSDMD knockout mice, and then we explored the specific cell type in which GSDMD functions as a positive regulator in UPEC-induced orchitis. To this end, we analyzed GSDMD expression in different cell type of testes, and found that GSDMD is predominantly expressed in testicular macrophages under physiological state. Upon UPEC treatment, GSDMD expression significantly increased and almost co-localized with macrophages. Therefore, we hypothesized that GSDMD in macrophages promotes orchitis. To validate this hypothesis, we used macrophage-specific GSDMD knockout mice (*Gsdmd^{fl/fl}Cx3cr1-cre*) and found that macrophage-specific GSDMD knockout mice showed the similar phenotype with conventional knockout mice.

7. Please explain the function of "LPS plus nigericin" in the "GSDMD boosts T cell response by enhancing antigen presentation of macrophages" part.

Response :The function of “LPS plus nigericin” was to activate NLRP3 inflammasome, leading to the activation of GSDMD and subsequently pyroptosis. UPEC is a Gram-negative bacterium that releases LPS during infection and can activate NLRP3 and pyroptosis, and thus using “LPS plus nigericin” was to mimic UPEC infection activating GSDMD and better demonstrated the function GSDMD-mediated pyroptosis in enhancing antigen presentation to boost T cell response.

8. In the "GSDMD in macrophages aggravates chronic autoimmune orchitis during EAO development " part, why are the testicular size and testis-body weight ratio of GSDMD deficient mice larger, which indicates that their testicular degeneration has been improved?

Response : Unlike an acute UPEC infection model, EAO is a chronic autoimmune orchitis, characteristic of the pathological testicular atrophy. Thus, the reason why the testes in *Gsdmd^{fl/fl}Cx3cr1-cre* mice showed larger is that GSDMD deficiency in macrophage impaired CD8⁺ T cells response by preventing the antigen presentation capacities, thereby attenuating the impairment of testicular tissue and atrophy.

Referee #2 (Comments on Novelty/Model System for Author):

FOR IMMUNOHISTOCHEMISTRY: all of the images in this manuscript show quite large gaps between tubules and where the interstitial space should be. This is resultant from an error in fixation or tissue processing and the structure of the tissue is not intact. I would recommend repeating these experiments to obtain higher quality staining and imaging. If it cannot be repeated, I recommend taking more images to see whether better imaging can be obtained, or removing this from the manuscript. If the authors feel the IHC is essential to the manuscript, then perhaps removing it to supplementary would be more appropriate. In any case, quantification of positive staining should be performed for all figures.

Response : We welcome the opportunity to clarify the images of HE and IF, and there were no the images of IHC in our manuscript.

For HE staining, we would like to highlight that a large proportion of images in our manuscript were testicular staining from **orchitic** mice. Under inflammatory conditions, seminiferous tubules are destroyed and interstitial spaces between seminiferous tubules were enlarged, which is consistent with other studies[4, 5]. Furthermore, the images of HE staining from no-treated ASC and GSDMD knockout mice showed that gaps between seminiferous tubules were large in Fig S2B, which may be related to the smaller size of testicular organ in knockout mice. We speculate that the diameter of seminiferous tubules in knockout mice is decreased, leading to an increase in interstitial spaces. Finally, in Fig S1B and Fig S2B, the images of HE staining from sham-operated or no-treated mice showed that the interstitial spaces between seminiferous tubules were normal, similar to previous studies[4, 6].

For immunofluorescent staining, we used PFA (paraformaldehyde) or PLP (paraformaldehyde

lysine periodate) fixation solution, which resulted in worse morphologic details in testes, compared to MDF fixation solution[6]. However, due to the need for GSDMD/CASPASE1/NLRP3/F4/80 antibody staining, we had to use this fixation solution. This might have led to large gaps between the seminiferous tubules in immunofluorescence images. Moreover, the relative integral optical density (IOD) of NLRP3, CASPASE1, and GSDMD, and the number of GSDMD-positive and F4/80-positive cells were quantified and were statistically analyzed using unpaired t-test (see new **Fig. 1D** and **Fig. 3B** or **Figure I** below). Finally, we believe that immunofluorescence data, together with a comprehensive body of data from immunoblot, and public databases, are solid and in support of our conclusions.

In addition, although the above clarification, in light of the Reviewer's suggestion we also have removed the images of HE staining to supplementary Figure.

Figure I: A. The quantified relative fluorescence intensity of NLRP3, CASPASE1, and GSDMD in the testes from Sham and UPEC-treated mice (n=3 mice per group). For immunofluorescent quantification, the indicated fluorescence intensity of tissues was normalized to the intensity of DAPI. **B.** The quantified proportion of GSDMD and F4/80 double positive cells (co-expressed cells) in the testes from Sham and UPEC-treated mice (n=3 mice per group). Error bars show mean \pm sd. *P < 0.05, **P < 0.01. Two-tailed unpaired student's t-test.

FOR FLOW CYTOMETRY: the axis tick values need to be added back into ALL FIGURES containing flow cytometry. Some of the positive populations appear to be gated very close to what I assume is the zero tick on both the x and y axes, which makes the flow cytometry data questionable, although the separation of 'positive populations' does appear distinct. Showing the

full gating strategies will help clear this issue up. Further, replicate values for the flow cytometry data can be graphed, and statistical analysis performed, to add some quantitative data to the argument being made, rather than heavily relying on representative and qualitative data.

Response : We apologize for not showing flow cytometry data more clearly. The reason why positive populations appear to be gated very close to the zero tick is that the scale of x and y axes were Bix mode. In this mode, the position closed to zero tick does not necessarily correspond 10^1 , but it could be 10^3 (see **Figure II** below). We have included the axis values into flow cytometry data and show the full gating strategies in the revised manuscript (see **Figure II** below). Furthermore, every flow cytometry data in manuscript showed corresponding statistical analysis graph, some of which were placed in the supplementary figure, but the corresponding statistical results were in the main figure. We have rearranged the results of Figure3 and Figure S3, and put representative flow cytometry plot and corresponding statistical analysis graph into one Figure in the revised manuscript.

Figure II: A. Flow Cytometry (FCM) gating strategy for analysis of different immune cell types in testes from *Gsdmd^{fl/fl}Cx3cr1-cre* and *Cx3cr1-cre* treated with UPEC. **B, C.** FCM gating strategy for analysis of BMDMs (**B**) and T cells (**C**) from co-culturing system.

FOR DATA PRESENTATION AND STATISTICS:

The authors have used SEM, which should only be used in population statistics; here standard deviation (SD) would be appropriate.

Response : In light of this suggestion, we have changed SEM into SD in the revised manuscript.

Some graphs, or even some bars within graphs, have bidirectional error bars and some only have unidirectional error bars, and this needs to be made consistent, preferably with bidirectional bars.

Response : In light of this suggestion, we have changed all error bars into bidirectional error bars in the revised manuscript.

Many of the axis labels are not descriptive enough, lack units, or are not labelled at all (Figure 4, which also lacks individual data points in F).

Response : We apologize for this oversight. We have included “ $\mu\text{g/g}\cdot 10^3$ ” in graph of testis-body weight ratio, add “relative expression” in Figure 4F, and add “Z score” in Figure 4G. We also have changed histogram into dot histogram in Figure 4F.

Additionally, many of the graphs throughout the manuscript that have been given * or ** have a

great deal of overlapping data between compared groups. This makes the statistical analysis incorrect. For my own interest, I replicated Figure 6F, in Prism, as indicated. I ran normality tests, and additional outlier analysis as the CXCL10 data has a lot of spread. I could not generate the same statistical significance that the authors have found, as expected with data that overlaps as much as seen here. A review of the data presentation, tests used, and significance outcome is required.

Response : The reviewer raises a valuable point. We apologize for not applying the data statistical method properly. We have applied all data to Shapiro–Wilk test to assess data distribution. For data with normal distribution, comparisons were performed using the unpaired t-test or multiple t-tests. Continuous variables data with non-normal distributions were analyzed by non-parametric tests such as the Mann-Whitney test. Indeed, there are some discrepancies between the p value obtained from Mann-Whitney test and unpaired t-test for some data with non-parametric tests, however, it does not affect the main conclusions in our manuscript. For example, CXCL10 data did not exhibit normal distribution, and we performed the Mann-Whitney test for statistical analysis, yielding a p-value of 0.0148 (*), indicating statistical significance. We have added the detailed statistical method in the figure legends, and have provided the original data for every graph.

Referee #2 (Remarks for Author):

Summary: This manuscript investigates the role of GSDMD and GSDMD-depletion in UPEC-mediated orchitis. While this is an interesting topic and is generally a well thought out study, the manuscript contains many grammatical/syntax errors and would benefit from a thorough proof read, and the manuscript contains numerous experimental flaws.

Response : We welcome the Reviewer’s positive comments on our study “this is an interesting topic and is generally a well thought out study”. We are also thankful for the Reviewer’s meaningful suggestion, and happy to address the points raised by this Reviewer.

In light of the Reviewer’s suggestion, we will recruit a native English speaker to edit the

manuscript to ensure the grammar and language are corrected.

In response to experimental flaws, we believe that misunderstanding may have been caused by inadequate description in the manuscript. We have detailed our responses to the individual points raised by each the Reviewers.

PAPER EXPLAINED and ABSTRACT:

This reviewer feels that the authors should rework the latter half of the abstract to reflect any changes made consequent to review, being careful not to oversell the mechanistic aspect of the study. Similarly, the authors should ensure they do not overreach for the conclusion that GSDMD makes a feasible therapeutic target for treating orchitis, as while the study may provide a foundation for that study, the current study does not investigate the necessary

Response : In light of this suggestion, we have rewritten the latter half abstract in the revised manuscript. Furthermore, pharmacological inhibition of GSDMD alleviates the symptoms of UPEC-induced acute orchitis. Collectively, these findings provide the first demonstration of GSDMD's role in driving orchitis, and GSDMD may be a potential therapeutic target for treating orchitis.

INTRODUCTION:

The authors indicate that 'approximately 60%' of cases of male infertility related to infection and inflammation are attributed to UPEC. However, the literature is much more complex than this, and rates vary greatly between studies. This could be due to type of testing, demographics, geographics, etc. but it is insufficient to cite only one article, from 2008, as the foundation for this statement. Please provide an unbiased and up-to-date comment on the role of UPEC in male infertility.

Further, the reference given (Wiles, Kulesus et al., 2008) does not contain the information given. This reference does not mention a 60% value, infertility, or orchitis that this reviewer could find after skimming the article and searching for keywords.

<https://academic.oup.com/molehr/article/26/4/215/5721557> this article cites a 60% value on a

similar topic, but it is not the same statistic that you have given.

A thorough review of all references is required.

Response :We apologize for this oversight, 'approximately 60%' came from the previous study [7].

It said that approximately 15% of male infertility cases are related to the inflammation of the reproductive system, and approximately 60% of such cases are caused by uropathogenic Escherichia Coli (UPEC), which cited one article from Wiles, Kulesus et al [8]. Wiles, Kulesus et al said that these bacteria are the primary cause of community-acquired urinary tract infections (UTI) (70–95%) and a large portion of nosocomial UTIs (50%), accounting for substantial medical costs and morbidity worldwide. We are so sorry that we have summarized the content of this paper incorrectly, and we have changed this sentence into “Uropathogenic Escherichia coli (UPEC) is identified as the predominant etiological pathogen of genitourinary infections, and can be isolated from 50%-95% of patients with genitourinary infections” in the revised manuscript.

In light of this suggestion, we have checked thoroughly all references and the incorrectly cited references have been labeled and corrected in the revised manuscript.

At the start of the second page of the introduction, the authors indicate that during orchitis, testicular macrophages 'transform' into pro-inflammatory macrophages that highly express proinflammatory cytokines. This is inaccurate, and the authors need to review the appropriate literature to understand the origin of inflammatory cytokines in the testes. This should include the role of recruited monocytes as key proinflammatory mediators.

Response : We welcome the opportunity to clarify the value point. Indeed, the origin of inflammatory cytokines in the testes are mainly from circulating monocytes and macrophages differentiated from them, with a smaller portion originating from resident pro-inflammatory testicular macrophages. Thus, we have changed these descriptions into “inflammatory testicular macrophages mainly from circulating monocytes promote tissue damage and damaged spermatogenesis” in the revised manuscript.

Similarly, the APC ability of testicular macrophages is different to standard macrophages in dogma, so please review your statements and conduct a more thorough update of the literature to

understand these two topics. Please ensure that the inflammation pathway information is specific to testicular macrophages in line with your updated reading.

Response : Thank you for your suggestion. Indeed, the APC ability of testicular macrophages is not entirely the same as standard macrophages in dogma. Although it is tempting to speculate that peritubular testicular macrophages might have the ability of presentation of antigens to T cells, based on their high levels of MHCII[9], but our statements are also inappropriate. We have modified these statements “peritubular testicular macrophages have been reported to express high levels of MHC[9], suggesting that peritubular testicular macrophages might have the ability of presentation of antigens to T cells” in the revised manuscript.

RESULTS:

GENERAL NOTE FOR IMMUNOHISTOCHEMISTRY: all of the images in this manuscript show quite large gaps between tubules and where the interstitial space should be. This seems to be resultant from an error in fixation or tissue processing and the structure of the tissue no longer appears intact. If possible, this reviewer would recommend repeating these experiments to obtain higher quality staining and imaging. If it cannot be repeated, I recommend taking more images to see whether better imaging can be obtained, or removing this from the manuscript. If the authors feel the IHC is essential to the manuscript, then perhaps removing it to supplementary would be more appropriate. In any case, quantification of positive staining should be performed for all figures. The authors rely on staining to draw many conclusions, therefore it would be appropriate to have quantitative data rather than qualitative, as is currently provided.

Response : In response to this comment, we would like to clarify that we performed HE and immunofluorescent staining but not immunohistochemistry (IHC).

For HE staining, we would like to highlight that a large proportion of images in our manuscript were testicular staining from **orchitic** mice. Under inflammatory conditions, seminiferous tubules are destroyed and interstitial spaces between seminiferous tubules were enlarged, which is consistent with other studies[4, 5]. Furthermore, the images of HE staining from no-treated ASC and GSDMD knockout mice showed that gaps between seminiferous tubules were large in Fig S2B, which may be related to the smaller size of testicular organ in knockout mice. We speculate

that the diameter of seminiferous tubules in knockout mice is decreased, leading to an increase in interstitial spaces. Finally, in Fig S1B and Fig S2B, the images of HE staining from sham-operated or no-treated mice showed that the interstitial spaces between seminiferous tubules were normal, similar to previous studies[4, 6].

For immunofluorescent staining, we used PFA (paraformaldehyde) or PLP (paraformaldehyde lysine periodate) fixation solution, which resulted in worse morphologic details in testes, compared to MDF fixation solution[6]. However, due to the need for GSDMD/CASPASE1/NLRP3/F4/80 antibody staining, we had to use this fixation solution. This might have led to large gaps between the seminiferous tubules in immunofluorescence images. Moreover, the relative integral optical density (IOD) of NLRP3, CASPASE1, and GSDMD, and the number of GSDMD and F4/80 double positive cells were quantified and were statistically analyzed using unpaired t-test (see new **Fig. 1D** and **Fig. 3B** or **Figure I** above). Finally, we believe that immunofluorescence data, together with a comprehensive body of data from immunoblot, and public databases, are solid and in support of our conclusions.

In addition, although the above clarification, in light of the Reviewer's suggestion we also have removed the images of HE staining to supplementary Figure.

Finally, please specify how many fields of view your images are representative of in the figure legend or methods, once quantification as been completed.

Response : We apologize for not providing the detailed HE analysis. For HE analysis, we applied the categorization of Johnsen criteria method to assess seminiferous tubules and quantify seminiferous tubules as previously described [10, 11]. In brief, after cutting the tissue at the largest cross-sectional position of testis, we start sectioning from this location for slide preparation. In each mouse, one microscopy slide would be evaluated if 150-250 seminiferous tubules in a section were visible at 40X magnification. Then, the microscopy slides that met the criteria were assessed by evaluating all seminiferous tubules under optical microscope (magnification, 100X). Approximately, 10-15 fields of view in one section contain 150-250 seminiferous tubules, and the most advanced germ cell type detectable in each tubule cross-section were recorded as the representative germ cell stage, and then the percentage of tubule cross-sections showing the

respective germ cell stage were finally documented. For example, a single slide includes total 200 seminiferous tubules, and among them 10 tubules only containing Sertoli cells were recorded. The frequency of tubules with Sertoli cells only would be 5%. We have included the detailed information in the revised manuscript.

Figure 1:

The Figure is well laid out and the panels are clear, there are several issues however.

Response : We welcome the Reviewer's positive comments on Figure 1.

In Figure 1B, Z-sore should be Z-score.

Response : We apologize for this mistake, and we have corrected this word in Fig 1B and Fig S1D in the revised manuscript.

In Figure 1c, multiple bands are observed in the Western blot of Pro CASP11 and others, which is fine, but the blots are cropped, which means we cannot see other potential bands and this is not acceptable by the journal and today's standards. Either in the figure or as a supplementary, full sized, uncropped blots must be provided.

Response : Procaspase-11 has two isoforms 43 and 38 kDa, and we apologize for the incomplete showing of Procaspase-11 bands. The correction will be included and the specific target bands will be indicated by arrows in the revised manuscript. Additionally, we have provided full sized and uncropped blots (see **Figure III** below).

Figure III: The raw picture of NLRP3, ASC, Pro-Caspase-1 Cleaved-Caspase-1, Pro-Caspase-11 full length GSDMD, Cleaved GSDMD, and β -Actin in testes from Sham or UPEC-treated WT mice.

The immunostaining images in Figure 1D appear to be out of focus in panels for NLRP3 and GSDMD.

Response : Indeed, the immunostaining image for NLRP3 appear unclear because the image was not saved properly but not out of focus. We will replace the image in the revised manuscript. However, the immunostaining image for GSDMD appear fine.

The Sham-GSDMD staining in Figure 1D vs Figure 3B is quite different, with quite a lot more staining present in the latter. What is the reason for this? Have the authors ensured that, in both cases, representative images have been provided? If so, why are these so different?

Response : We apologized for not choosing the representative images in Figure 1D and Figure 3B. To address the concerns, we have now provided images of 3 mice per group from three independent experiments and the quantitative analysis in the revised manuscript (see **new Fig. 1D** and **Fig. 3B or Figure IV** below). Moreover, we have changed the representative images in Figure 1D and Figure 3B in the revised manuscript.

Figure IV: A. All immunofluorescent analysis GSDMD expression in the testes of WT mice injected with UPEC for 36 h from three independent experiments, (n=3 mice per group). **B.** All immunofluorescence staining for GSDMD (red) and F4/80 (green) in testicular tissue from WT mice injected with UPEC for 36 h from three independent experiments, (n=3 mice per group).

In the figure legend, please clarify whether n = 5 is the total number of mice used, or whether the three independent experiments conducted were each n = 5, giving a total of n = 15.

Response :We welcome the opportunity to clarify this point. We have clarified that N=5 is the total number of mice used in the revised manuscript.

Please define all of the abbreviations used in the figure, in the figure legend. E.g. CASP1.

Response : We will define all of the abbreviations used in the figure, in the figure legend. For example, testicular homogenate (TH), pertussis toxin (PTX), caspase1(CASP1), caspase11(CASP11).

Figure 2:

The figure legend indicates that Sham as well as UPEC infected mice were assessed, however, I cannot see any data from Sham mice in this figure. Please address this, most appropriately would be to include the Sham data in this figure for side-by-side comparisons of KOs.

Response : We apologize for this oversight, and we will remove the describing “Sham” from the figure legend.

For the testis/Body weight graph in A, please provide a unit of measure, e.g. mg.

In the figure legend, please define all abbreviations, e.g. SCO, SG, M/mL

Response : In light of this suggestion, we will include “ $\mu\text{g/g}$ ” in graph of testis-body weight ratio, and define all of the abbreviations used in the figure, in the figure legend. ES= elongated spermatids, RS = round spermatids, PSc= pachytene spermatocytes, SG=spermatogonia, SCO= Sertoli-cell only.

Authors state that the Johnsen Criteria method was used for assessing the seminiferous tubules in C, however, my understanding is that this method results in a score from 1 to 10, not a percentage, so please explain clearly in the methods/legend how this percentage has been obtained. Please also clarify how tubules were staged, in order to obtain the percentages for round and elongated spermatocytes, for example, as I currently cannot see if this one performed, to standardize the counting (i.e. the number of these types of cells is dependent on the stage, was this accounted for).

Response : We apologize for not describing the quantified method more clearly. In fact, we only utilized Johnsen criteria qualitative method, which categorized seminiferous tubules into different types including elongated spermatids (ES), round spermatids (RS), pachytene spermatocytes (PSc), spermatogonia (SG), and Sertoli-cell only (SCO), based on the presence or absence of the main cell types arranged in the order of maturity[12]. Next, we applied the categorization of Johnsen criteria method to assess seminiferous tubules and quantify seminiferous tubules as previously described [10, 11]. Briefly, 150-250 seminiferous tubules per mouse were evaluated, and the most advanced germ cell type detectable in each tubule cross-section were recorded as the representative germ cell stage, and then the percentage of tubule cross-sections showing the respective germ cell stage were finally documented. For example, a single slide includes total 200 seminiferous tubules, and among them 10 tubules only containing Sertoli cells were recorded. The frequency of tubules with Sertoli cells only would be 5%. We will provide the more detailed information in the revised manuscript.

In Figure 2/S1, there is a white line/spot observed on the testis, is this just light reflecting on damp

tissue at the time the photograph was taken, or something else that the authors have annotated on the image?

Response : A white line/spot observed on the testis is just light on damp tissue when taking the photograph.

For D, the image are really too small to be useful. Suggest moving these to supplementary if the authors feel they are necessary to the manuscript. Alternatively, enlarge the images and condense some of the graphical information. Condensing the graphical information would also help improve the readability of the figure.

Response : In light of this suggestion, we have enlarged the images of Figure 2D.

For E and F, please clarify your n values. Is this n = 2 mice repeated 3 times, to give n = 6 in total? By pooled data, do you mean to say that your triplicate independent experiments are shown on a single graph, or do you mean that the actual samples were pooled?

Response :We welcome the opportunity to clarify these points. We have made clear that N=6 is the total number of mice used in the revised manuscript. For pooled data, we showed triplicate independent experiments on a single graph.

In Figure S2A, the color of the testis appears different in *Gsdmd*^{-/-} and *Asc*^{-/-} mice. The reason for this color difference could be expanded upon if space permits, or if not relevant, please explain in response.

Response : We welcome the opportunity to clarify this point. The reason for difference color of the testes from *Gsdmd*^{-/-} and *Asc*^{-/-} mice may be attributed to pathological changes induced by knockout, which was an interesting phenotype. However, this fascinating finding requires further exploration to elucidate the underlying mechanism.

In Figure S2B, the H&E staining of *Gsdmd*^{-/-} and *Asc*^{-/-} mice shows an increase in interstitial space. The authors claim that the testis size is reduced in these mice, but it is unclear how they

relate the testis size to the increase in interstitial space.

Response : We speculate that the diameter of seminiferous tubules in *Gsdmd*^{-/-} and *Asc*^{-/-} mice is decreased, leading to an increase in interstitial spaces. However, the specific mechanisms by which ASC and GSDMD regulate the diameter of seminiferous tubules and the size of testes observation requires further exploration.

Figure 3:

B - comments as above about quality of sections and destruction of tissue structure. These images are not appropriate to draw conclusions from.

Response : For immunofluorescent staining, we used PFA (paraformaldehyde) or PLP (paraformaldehyde lysine periodate) fixation solution, which resulted in worse morphologic details in testes, compared to MDF fixation solution. However, due to the need for GSDMD/CASPASE1/NLRP3/F4/80 antibody staining, we had to use this fixation solution. This may have led to large gaps between the seminiferous tubules in immunofluorescence images. Moreover, we have quantified the numbers of cells which were GSDMD and F4/80 double positive (see new **Fig. 3B or Figure I** above). Finally, we believe that immunofluorescence data, together with a comprehensive body of data from immunoblot, and public databases, are solid and in support of our conclusions.

*The authors claim to have generated macrophage conditional knockout mice of GSDMD (*Gsdmd*^{fl/fl}*Cx3cr1-cre*), but they have not shown any data demonstrating GSDMD deletion in macrophages.*

Response : We welcome the opportunity to clarify the demonstration of GSDMD deletion in macrophages of *Gsdmd*^{fl/fl}*Cx3cr1-cre* mice. The GSDMD deletion of *Gsdmd*^{fl/fl}*Cx3cr1-cre* mice had been demonstrated using single-cell RNA-seq in our previous study[13].

The decrease in immune cells after UPEC infection is shown in *Gsdmd*^{fl/fl}*Cx3cr1-cre* mice, but the immune cell population in *Gsdmd*^{fl/fl}*Cx3cr1-cre* mice under physiological conditions is not

addressed.

Response : The reviewer raises a valuable point. To address this concern, we include new data to investigate the different immune cell types in *Cx3cr1-cre* and *Gsdmd^{fl/fl}Cx3cr1-cre* mice under physiological conditions by FCM. We observed there were no difference in the absolute numbers and proportion of immune cells (CD45⁺), macrophages (CD11b⁺F4/80⁺), CD4⁺ and CD8⁺ T cells between *Cx3cr1-cre* and *Gsdmd^{fl/fl}Cx3cr1-cre* mice (see new **Figure EV3C, D or Figure V** below). We have included this new data in the revised manuscript.

Figure V: A, B. Testes from *Gsdmd^{fl/fl}Cx3cr1-cre* and *Cx3cr1-cre* mice aged at 10-12 weeks were collected for flow cytometric analysis. The number of total immune cells (**A**) and the quantified absolute number (**B**) of the macrophages (CD11b⁺F4/80⁺), CD8⁺ T cells, CD4⁺ T cells, (n = 5 mice per group).

In Figure 3I, the y-axis represents immune cells in number (10⁴) in CD45⁺ cells. However, it is generally known that macrophages contribute around 60% of the immune cell population. This information is not reflected in the graph. Please justify why your results are widely different to existing literature.

Response : We welcome the opportunity to clarify this point. Indeed, our new data showed that macrophages contribute around 60% of the immune cell population in testes under physiological conditions (see new **Fig S3D or Figure VI** below). However, we showed immune cells of testes from UPEC-induced orchitic mice in Figure 3I, and it is generally known that the composition of testicular immune cells changes dramatically under inflammatory condition, because a large number of immune cells migrate into inflamed tissue.

Figure VI: Testes from *Gsdmd*^{fl/fl}*Cx3cr1-cre* and *Cx3cr1-cre* mice aged at 10-12 weeks were collected for flow cytometric analysis. Data were presented as representative flow cytometry plots and summary graphs of percentages of the macrophages.

In Figure S3B, In the gating strategy is not clearly written or shown which parent population were taken.

Response :We have included gating strategy of flow cytometry in the revised manuscript (see new **Figure EV3B or Figure VII below or Figure II A** above).

Figure VII: Flow Cytometry (FCM) gating strategy for analysis of different immune cell types in testes from *Gsdmd*^{fl/fl}*Cx3cr1-cre* and *Cx3cr1-cre* treated with UPEC.

In Figure S3B, the flow cytometry plot depicting macrophages reveals the presence of two distinct populations. It is possible that one of these populations corresponds to infiltrated monocytes rather than true macrophages. Notably, only a small proportion of the F4/80^{high}CD11b^{low} population is

identified as macrophages. This raises the question of how this particular plot is observed in the absence of UPEC infection.

Response : The reviewer raises a valuable point. To address this concern, we include new data to investigate the flow cytometry plot of macrophages in *Cx3cr1-cre* and *Gsdmd^{fl/fl}Cx3cr1-cre* mice in the absence of UPEC infection. We observed that the macrophages showed only one population in the absence of UPEC treatment (see **Figure VIII** below). In addition, the flow cytometry plot of macrophages in testes from UPEC-treated mice showed the two distinct groups, and one of which showed both F4/80^{high} and F4/80^{low} in CD11b^{high} cells. The expression of F4/80 by infiltrated monocytes is relatively lower than tissue macrophages[14]. Therefore, we speculated that a large proportion of CD11b^{high} population may include infiltrated monocytes and macrophages, which may represent functionally distinct macrophages from F4/80^{high} CD11b^{low} population. However, further labeling with Ly6C is needed to determine whether the two populations include infiltrated monocytes.

Figure VIII: Testes from *Gsdmd^{fl/fl}Cx3cr1-cre* and *Cx3cr1-cre* mice aged at 10-12 weeks were collected for flow cytometric analysis. Data were presented as representative flow cytometry plots of the macrophages.

As a general comment for ALL FLOW CYTOMETRY: the axis tick values need to be added back into ALL FIGURES containing flow cytometry. Some of the positive populations appear to be gated very close to what I assume is the zero tick on both the x and y axes, which makes the flow cytometry data questionable, although the separation of 'positive populations' does appear distinct. Showing the full gating strategies will help clear this issue up. Further, replicate values for the

flow cytometry data can be graphed, and statistical analysis performed, to add some quantitative data to the argument being made, rather than heavily relying on representative and qualitative data.

Response : We apologize for not showing flow cytometry data more clearly. The reason why positive populations appear to be gated very close to the zero tick is that the scale of x and y axes were Biex mode. In this mode, the position closed to zero tick does not necessarily correspond 10^1 , but it could be 10^3 (see **Figure II** above). We have included the axis values into flow cytometry data and show the full gating strategies in the revised manuscript (see **Figure II** above). Furthermore, every flow cytometry data in manuscript showed corresponding statistical analysis graph, some of which were placed in the supplementary figure, but the corresponding statistical results were in the main figure. We have rearranged the results of Figure3 and Figure S3, and put representative flow cytometry plot and corresponding statistical analysis graph into one Figure in the revised manuscript.

Figure 4:

In Figure 4F, the individual data points are not shown, which limits the ability to assess the variability and significance of the results. Y axis label is missing

Response : We apologize for this oversight. We will add “relative expression”, and change histogram into dot histogram in Figure 4F (new Figure 5F).

What is the difference in genes displayed in the Venn diagram vs the heatmap, is the Venn diagram the total number of genes detected whereas the heatmap is only the significantly dysregulated genes? If so, please make this clear in the figure legend and results. If not, and the UP and Down represent significantly altered genes, why was the selection displayed in the heatmap chosen?

Response : We apologize for not describing these data more clearly. The Venn diagram showed the total number of genes detected, including the upregulated differentially expressed genes (DEGs, fold change>1.2, p value<0.05), the downregulated DEGs, and no differentially expressed genes, and the heatmap showed the representative genes of the significantly dysregulated pathways after GSDMD deficiency, and we have clarified these details in the revised manuscript.

The pooling statement in the legend needs to be clarified as per previous comments. A brief overview of the analytical pipeline needs to be provided. E.g. DEGs were identified using ... Heatmap was generated using in the legend also. The program used for generating graphs and statistics needs to be added to every figure legend.

Response : Thank you for your suggestion. We have modified the text and add details of analytical pipeline. DEGs were identified using fold change>1.2, p value<0.05. Principal component analysis was performed on <https://www.omicstudio.cn>, and Venn diagram was performed on <https://www.omicshare.com/tools/>. Heatmap was generated using DEGs (fold change>1.2, p value<0.05) involved in most significantly enriched signaling pathways. The graphs for GO-BP and Reactome pathway analysis were performed on <http://david.ncifcrf.gov>, and the graphs were performed on GraphPad Prism 8.0. The analysis and graphs for GSEA were performed on software GSEA v3.0. The heatmap was performed on GraphPad Prism 8.0.

Figure 5: changes as per previous comments. Additionally, 21% interferon gamma expression is quite high, as IFNg is de novo assembled after stimulation. There does not appear to be clear separation between the positive and negative cells in this figure for most markers, please amend these plots as per previous comments and display a gating strategy.

Response : We have included gating strategy of flow cytometry in the revised manuscript (see new **Figure EV4B, 4C or Figure IX** below or **Figure II** above). Notably, the proportion of interferon gamma was consistent with previous studies[15, 16].

Figure IX: A, B. FCM gating strategy for analysis of BMDMs (A) and T cells (B) from co-culturing system.

Figure 6/7: comments as per previous

Response : Thank you for your suggestion, our answers were described above.

GENERAL COMMENTS ON DATA PRESENTATION AND STATISTICS:

The authors have used SEM, which should only be used in population statistics; here standard deviation (SD) would be appropriate.

Response : Thank you for your suggestion, we have changed SEM into SD in the revised manuscript.

Some graphs, or even some bars within graphs, have bidirectional error bars and some only have unidirectional error bars, and this needs to be made consistent, preferably with bidirectional bars.

Response: Thank you for your suggestion, we have changed all error bars into bidirectional error bars in the revised manuscript.

Many of the axis labels are not descriptive enough, lack units, or are not labelled at all (Figure 4, which also lacks individual data points in F).

Response : We apologize for this oversight. We have included “mg/g*10³” in graph of testis-body weight ratio, and add “relative expression” in Figure 4F, and add “Z score” in Figure 4G. We will also change histogram into dot histogram in Figure 4F.

Additionally, many of the graphs throughout the manuscript that have been given * or ** have a great deal of overlapping data between compared groups. This makes the statistical analysis seem less valid. For my own interest, I replicated Figure 6F, in Prism, as indicated. I ran normality tests,

and additional outlier analysis as the CXCL10 data has a lot of spread. I could not generate the same statistical significance that the authors have found, as expected with data that overlaps as much as seen here. A review of the data presentation, tests used, and significance outcome is required.

Response : The reviewer raises a valuable point. We apologize for not applying the data statistical method properly. We have applied all data to Shapiro–Wilk test to assess data distribution. For data with normal distribution, comparisons were performed using the unpaired t-test or multiple t-tests. Continuous variables data with non-normal distributions were analyzed by non-parametric tests such as the Mann-Whitney test. Indeed, there are some discrepancies between the p value obtained from Mann-Whitney test and unpaired t-test for some data with non-parametric tests, however, it does not affect the main conclusions in our manuscript. For example, CXCL10 data did not exhibit normal distribution, and we performed the Mann-Whitney test for statistical analysis, yielding a p-value of 0.0148 (*), indicating statistical significance. Furthermore, we have added the detailed statistical method in the figure legends, and have provided the original data for every graph.

DISCUSSION: Similar comments to the introduction about fact-checking some statements in the opening paragraph of the discussion.

Response : In light of this suggestion, we have modified some statements about mechanism and therapeutic target of discussion in the revised manuscript.

The limitations of the study need to be discussed. Injecting UPEC into the testes does not represent a natural method of infection, which would presumably be by ascending infection from the penile urethra. The testes and sperm health are susceptible to damage through systemic inflammation, much of recent COVID-19 literature displays this well.

Response : Thank you for your suggestion, we have included the limitations of the study about UPEC-induced orchitis mouse model. It is worth noting that although UPEC initially causes epididymis by ascending infection from the penile urethra, approximately in 60% of infected

patients experience orchitis[10]. Furthermore, fertility impairment following epididymo-orchitis can originate from direct exposure of testicular germ cells to pathogens, toxins or pathogens-induced immune response[10]. Indeed, the testes and sperm are susceptible to systemic inflammation. It has been reported that UPEC ascended to the testis at day 7 post inoculation into the lumen of the vas deferens[17]. Additionally, there are rare studies on the direct effects of UPEC on testis. Therefore, although some limitations of injecting UPEC into the testes, it is significant and meaningful to the investigate the direct effects of UPEC on testes.

MATERIALS AND METHODS:

The animal ethics committee that provided approval for this study needs to be named, and the approval number provided.

Response :All animal procedures were approved by the Ethical Review Committee for Laboratory Animal Welfare of Nanjing Medical University, and approval number was 1705038. We have included this information in the revised manuscript.

Original publications showing the generation and validation of various mouse strains needs to be included.

Response : We thank Reviewer for the valuable suggestion. *Nlrp3^{-/-}*, *Aim2^{-/-}*, *Asc^{-/-}*, and *Caspase1/11^{-/-}* mice were a gift from Dr.V.Dixit (Genentech, South San Francisco, CA), and were described previously[18]. The generation and validation of *Nlrp3^{-/-}*[19], *Aim2^{-/-}*[20], *Asc^{-/-}*[21], and *Caspase1/11^{-/-}*[18] mice were described previously. The generation and validation of *Gsdmd^{-/-}* mice were described previously[22]. The generation and validation of *Cx3cr1*-cre mice were described previously[23]. We have included all of these relevant description in our study.

"Experimental orchitis induced by UPEC in male mice was improved on previously described method(Jing, Cao et al., 2021a). Briefly, the mice were anesthetized and the testes and vas deferens were exposed. Approximately 1×10^5 colony-forming units (CFU) of bacteria in 5 μ l PBS were injected into the upper pole of the testis, and 5 μ l PBS was injected into the sham group mice. Then the vas deferens were ligated to prevent the spread of infection." - This part of the

methods requires more details, e.g. type of anaesthesia given, details of 'improvements' made, surgery details for vas ligation. Additionally, there seems to have been invasive surgery performed here, so details for pain relief and any other measures to ensure the health of the mice should be provided.

Response : The method of anesthesia used in our study was intraperitoneal injection of tribromoethanol (180mg/kg). For details of “improvement”, bacteria were injected into the upper pole of the testis instead of the lumen between the ligation site and the cauda epididymidis, which described by Klein et al., 2020. Thus, we have changed the word “improved” into “modified”. Furthermore, we apologized for the wrong cited references, and we have corrected in the revised manuscript. For vas ligation, the vas deferens were directly tied a knot using surgical sutures. After the wound was sutured, these mice were kept warm until recovery. Furthermore, the wound suture was disinfected with iodine twice daily. We have included these descriptions and products details in the revised manuscript.

Product details, e.g. MDF, are missing in multiple places, these need to be provided.

Response : We welcome the opportunity to clarify the details of Modified Davidson’s Fixative (MDF). The MDF contained 1.2% formaldehyde, 15% ethyl alcohol, 5% glacial acetic acid.

In summary, we appreciate all of the Reviewer’s very constructive comments and suggestions. We feel that the inclusion of new data addresses their suggestion in a comprehensive manner and adds further support to our original conclusions. We hope you will now consider the revised manuscript worthy of publication in *EMBO Molecular Medicine*.

Kind Regards

Shuo Yang

Corresponding Author

References

1. Liu, M., et al., *CXCL10/IP-10 in infectious diseases pathogenesis and potential therapeutic implications*. Cytokine & growth factor reviews, 2011. **22**(3): p. 121-130.
2. Lee, E.Y., Z.-H. Lee, and Y.W. Song, *CXCL10 and autoimmune diseases*. Autoimmunity reviews, 2009. **8**(5): p. 379-383.
3. Guazzone, V.A., et al., *Cytokines and chemokines in testicular inflammation: a brief review*. Microscopy research and technique, 2009. **72**(8): p. 620-628.
4. Wang, M.-W., et al., *Activation of PTH1R alleviates epididymitis and orchitis through Gq and β -arrestin-1 pathways*. Proceedings of the National Academy of Sciences, 2021. **118**(45): p. e2107363118.
5. Wang, M., et al., *Two populations of self-maintaining monocyte-independent macrophages exist in adult epididymis and testis*. Proceedings of the National Academy of Sciences, 2021. **118**(1): p. e2013686117.
6. Wang, H., et al., *Different fixative methods influence histological morphology and TUNEL staining in mouse testes*. Reproductive Toxicology, 2016. **60**: p. 53-61.
7. Su, Y., et al., *Prokineticin 2 via calcium-sensing receptor activated NLRP3 inflammasome pathway in the testicular macrophages of uropathogenic Escherichia coli-induced orchitis*. Frontiers in Immunology, 2020. **11**: p. 570872.
8. Wiles, T.J., R.R. Kulesus, and M.A. Mulvey, *Origins and virulence mechanisms of uropathogenic Escherichia coli*. Experimental and molecular pathology, 2008. **85**(1): p. 11-19.
9. Mossadegh-Keller, N., et al., *Developmental origin and maintenance of distinct testicular macrophage populations*. Journal of Experimental Medicine, 2017. **214**(10):

p. 2829-2841.

10. Klein, B., et al., *Differential tissue-specific damage caused by bacterial epididymo-orchitis in the mouse*. Molecular human reproduction, 2020. **26**(4): p. 215-227.
11. Rathi, R., et al., *Germ cell fate and seminiferous tubule development in bovine testis xenografts*. Reproduction, 2005. **130**(6): p. 923-929.
12. Johnsen, S.G., *Testicular biopsy score count—a method for registration of spermatogenesis in human testes: normal values and results in 335 hypogonadal males*. Hormone Research in Paediatrics, 1970. **1**(1): p. 2-25.
13. Jiang, Y., et al., *Gasdermin D restricts anti-tumor immunity during PD-L1 checkpoint blockade*. Cell Reports, 2022. **41**(4).
14. Gundra, U.M., et al., *Alternatively activated macrophages derived from monocytes and tissue macrophages are phenotypically and functionally distinct*. Blood, The Journal of the American Society of Hematology, 2014. **123**(20): p. e110-e122.
15. Hong, J.Y., et al., *Long-term programming of CD8 T cell immunity by perinatal exposure to glucocorticoids*. Cell, 2020. **180**(5): p. 847-861. e15.
16. Blagih, J., et al., *Cancer-specific loss of p53 leads to a modulation of myeloid and T cell responses*. Cell reports, 2020. **30**(2): p. 481-496. e6.
17. Michel, V., et al., *Uropathogenic Escherichia coli causes fibrotic remodelling of the epididymis*. The Journal of pathology, 2016. **240**(1): p. 15-24.
18. Kayagaki, N., et al., *Non-canonical inflammasome activation targets caspase-11*. Nature, 2011. **479**(7371): p. 117-121.

19. Martinon, F., et al., *Gout-associated uric acid crystals activate the NALP3 inflammasome*. Nature, 2006. **440**(7081): p. 237-241.
20. Jones, J.W., et al., *Absent in melanoma 2 is required for innate immune recognition of Francisella tularensis*. Proceedings of the National Academy of Sciences, 2010. **107**(21): p. 9771-9776.
21. Mariathasan, S., et al., *Differential activation of the inflammasome by caspase-1 adaptors ASC and Ipaf*. Nature, 2004. **430**(6996): p. 213-218.
22. Shi, J., et al., *Cleavage of GSDMD by inflammatory caspases determines pyroptotic cell death*. Nature, 2015. **526**(7575): p. 660-665.
23. Yona, S., et al., *Fate mapping reveals origins and dynamics of monocytes and tissue macrophages under homeostasis*. Immunity, 2013. **38**(1): p. 79-91.

9th Nov 2023

Dear Prof. Yang,

Thank you for the submission of your revised manuscript to EMBO Molecular Medicine. I am pleased to inform you that we will be able to accept your manuscript pending the following final amendments:

- 1) Please address minor points suggested by the referee #2.
- 2) Please pay particular attention to the grammar and syntax and I would recommend running the article by a native English speaker.
- 3) In the main manuscript file, please do the following:
 - Please address all comments suggested by our data editors listed below:
 - o Data availability section:
 1. Please note that the specific URL for PRJNA985126 dataset is not provided in the data availability statement.
 2. Please note that reviewer access link for PRJNA985126 dataset is provided in the data availability statement.
 - o Figure legends:
 1. Please note that the figure legend style does not comply with the journal guidelines i.e. all the figure legends are in a run-on style.
 2. Please note that a separate 'Data Information' section is required in the legends of all the figures.
 3. Please define the annotated p values ***/**/* in the legend of figure 1d; 2a-b, d-f as appropriate.
 4. Please indicate the statistical test used for data analysis in the legends of figures 5c-d
 5. Please note that the error bars are not defined in the legend of figure 1d.
 6. Please note that information related to n is missing in the legend of figures 6b-e.
 7. Please note that the scale bar needs to be defined for extended data figures 4d-e.
 - o Data citation:
 1. Please note that the data citation does not refer to deposited experimental data, but refers to journal article.
 2. Please note that the data callout in the text for GSE112393 data citation does not include "Data ref:" as a prefix.
 3. Please note that the URL for GSE112393 data citation is not provided.
 - Add up to 5 keywords.
 - Please add callout for Fig 5B, Fig 7G and Fig EV5.
 - Please rename "Conflict of interest" to "Disclosure Statement & Competing Interests". We updated our journal's competing interests policy in January 2022 and request authors to consider both actual and perceived competing interests. Please review the policy <https://www.embopress.org/competing-interests> and update your competing interests if necessary.
 - Author contributions: Please remove it from the manuscript and specify author contributions in our submission system. CRediT has replaced the traditional author contributions section because it offers a systematic machine-readable author contributions format that allows for more effective research assessment. You are encouraged to use the free text boxes beneath each contributing author's name to add specific details on the author's contribution. More information is available in our guide to authors: <https://www.embopress.org/page/journal/17574684/authorguide#authorshipguidelines>
 - Correct the reference citation in the reference list. Where there are more than 10 authors on a paper (Nicolas N et al (2017)), 10 will be listed, followed by "et al.". Please check "Author Guidelines" for more information. <https://www.embopress.org/page/journal/17574684/authorguide#referencesformat>
 - Please be aware that all datasets should be made freely available upon acceptance, without restriction. Please check "Author Guidelines" for more information. <https://www.embopress.org/page/journal/17574684/authorguide#availabilityofpublishedmaterial>
 - Correct data citation in the reference list. Each data reference should be labeled with the tag "[DATASET]" at the end to distinguish it from the other references and include authors, when possible, the year, the full name of the database where the data is available, the accession number or DOI and, importantly, a resolvable link that points directly to the dataset. Please check "Author Guidelines" for more information. <https://www.embopress.org/page/journal/17574684/authorguide#referencesformat>
 - Please correct the EV figure nomenclature in their legends to "Figure EV1" etc.
- 4) Remove the file with combined main and EV figures.
- 5) Funding: Please merge it with "Acknowledgement". Also make sure that information about all sources of funding are complete in both our submission system and in the manuscript. Currently, National Natural Science Foundation of China 82270539, SKLRM-2022BP3, NJMURC20220014, 2020YLXK017, BK20221352, SBK2022030080 are missing in our submission system.
- 6) The Paper Explained: I have gone through your text and included some changes (see below). Please review it, amend as you see fit:

Problem:

Orchitis, the inflammation of the testis resulting from infection or autoimmune disorders, contributes significantly to male infertility. Existing treatments involving antibiotics and broad-spectrum anti-inflammatory drugs often fail to fully restore fertility and are accompanied by undesirable side effects. Therefore, there's a pressing need to delve into the pathogenesis of orchitis and explore alternative therapeutic approaches for more effective treatment.

Results:

We found that Gasdermin D (GSDMD) is activated during uropathogenic Escherichia coli (UPEC)-induced orchitis. GSDMD deficiency in macrophages limits inflammation and spermatogenesis impairment during UPEC-induced acute orchitis and chronic autoimmune orchitis. On a mechanistic level, GSDMD ablation in macrophage impairs CD8+ T cell response by impeding antigen presentation capacities. The use of GSDMD inhibitors demonstrated protective effects in mice subjected to UPEC-induced orchitis.

Impact:

Our study demonstrates the role of GSDMD in driving orchitis. In addition, the study suggests that targeting GSDMD may be a potential therapeutic avenue to treat both UPEC-induced acute orchitis and chronic autoimmune orchitis.

7) Synopsis:

- Synopsis image: Please simplify the graphical abstract, make sure that essential parts are clearly labeled and readable and upload the image as a separate, high-resolution jpeg file 550 px-wide x (250-400)-px high.
- Synopsis text: I have gone through your text and included some changes (see below). Please review it, amend as you see fit and upload as a separate .doc file.

Inflammation in the testis triggered by infection or an autoimmune disorder may lead to male infertility. Uropathogenic Escherichia coli (UPEC)-induced orchitis and experimental autoimmune orchitis (EAO) mouse models were used to identify the critical role of the Gasdermin D (GSDMD)-mediated pyroptosis in the pathogenesis of orchitis.

Bullet points:

1. The cleavage of GSDMD in testis is induced by UPEC infection.
2. Gsdmdfl/flCx3cr1-cre mice are protected from UPEC-induced acute orchitis and chronic EAO.
3. GSDMD deficiency reduces inflammatory response and antigen presentation capacities of macrophages, thereby impairing CD8+ T cell response.
4. The FDA-approved drug DMF inhibiting GSDMD activation alleviates UPEC-induced acute orchitis.

8) Source data: Please upload source data for one figure as one zipped file per figure.

9) For more information: This space should be used to list relevant web links for further consultation by our readers. Could you identify some relevant ones and provide such information as well? Some examples are patient associations, relevant databases, OMIM/proteins/genes links, author's websites, etc...

10) As part of the EMBO Publications transparent editorial process initiative (see our Editorial at <http://embomolmed.embopress.org/content/2/9/329>), EMBO Molecular Medicine will publish online a Review Process File (RPF) to accompany accepted manuscripts. This file will be published in conjunction with your paper and will include the anonymous referee reports, your point-by-point response and all pertinent correspondence relating to the manuscript. Let us know whether you agree with the publication of the RPF and as here, if you want to remove or not any figures from it prior to publication. Please note that the Authors checklist will be published at the end of the RPF.

11) Please provide a point-by-point letter INCLUDING my comments as well as the reviewer's reports and your detailed responses (as Word file).

I look forward to reading a new revised version of your manuscript as soon as possible.

Yours sincerely,

Zeljko Durdevic

*** Instructions to submit your revised manuscript ***

*** PLEASE NOTE *** As part of the EMBO Publications transparent editorial process initiative (see our Editorial at <https://www.embopress.org/doi/pdf/10.1002/emmm.201000094>), EMBO Molecular Medicine will publish online a Review

Process File to accompany accepted manuscripts.

1) a .docx formatted version of the manuscript text (including Figure legends and tables)

2) Separate figure files*

3) supplemental information as Expanded View and/or Appendix. Please carefully check the authors guidelines for formatting Expanded view and Appendix figures and tables at <https://www.embopress.org/page/journal/17574684/authorguide#expandedview>

4) a letter INCLUDING the reviewer's reports and your detailed responses to their comments (as Word file).

5) The paper explained: EMBO Molecular Medicine articles are accompanied by a summary of the articles to emphasize the major findings in the paper and their medical implications for the non-specialist reader. Please provide a draft summary of your article highlighting

This may be edited to ensure that readers understand the significance and context of the research.

Please refer to any of our published articles for an example.

6) For more information: There is space at the end of each article to list relevant web links for further consultation by our readers. Could you identify some relevant ones and provide such information as well? Some examples are patient associations, relevant databases, OMIM/proteins/genes links, author's websites, etc...

7) Author contributions: the contribution of every author must be detailed in a separate section.

8) EMBO Molecular Medicine now requires a complete author checklist (<https://www.embopress.org/page/journal/17574684/authorguide>) to be submitted with all revised manuscripts. Please use the checklist as guideline for the sort of information we need WITHIN the manuscript. The checklist should only be filled with page numbers where the information can be found. This is particularly important for animal reporting, antibody dilutions (missing) and exact values and n that should be indicated instead of a range.

9) Every published paper now includes a 'Synopsis' to further enhance discoverability. Synopses are displayed on the journal webpage and are freely accessible to all readers. They include a short stand first (maximum of 300 characters, including space) as well as 2-5 one sentence bullet points that summarise the paper. Please write the bullet points to summarise the key NEW findings. They should be designed to be complementary to the abstract - i.e. not repeat the same text. We encourage inclusion of key acronyms and quantitative information (maximum of 30 words / bullet point). Please use the passive voice. Please attach these in a separate file or send them by email, we will incorporate them accordingly.

You are also welcome to suggest a striking image or visual abstract to illustrate your article. If you do please provide a jpeg file 550 px-wide x 300-800px high.

10) A Conflict of Interest statement should be provided in the main text

11) Please note that we now mandate that all corresponding authors list an ORCID digital identifier. This takes <90 seconds to complete. We encourage all authors to supply an ORCID identifier, which will be linked to their name for unambiguous name identification.

Currently, our records indicate that the ORCID for your account is 0000-0003-4164-4438.

Link Not Available

Photos 400-800 DPI

*Additional important information regarding figures and illustrations can be found at <https://bit.ly/EMBOPressFigurePreparationGuideline>. See also figure legend preparation guidelines: <https://www.embopress.org/page/journal/17574684/authorguide#figureformat>

***** Reviewer's comments *****

Referee #1 (Comments on Novelty/Model System for Author):

The authors have revised the text to address my concerns which is now clearer.

Referee #2 (Comments on Novelty/Model System for Author):

As per review 1. Additionally, the authors have greatly improved the clarity and believability of the data by providing quantitative data.

Referee #2 (Remarks for Author):

The authors have clearly put in a great deal of effort to improve this manuscript. The flow and accuracy is much improved. This reviewer appreciates the effort put into providing quantitative data. Almost all of my comments have been addressed.

Figure 1D still appears blurry, so I question the authors claim that they replaced this image with one that has been saved correctly.

I could not see Figure III, which has the full set of uncropped blots, I'm not sure whether I just don't have access to this and will refer this to the editor.

Neisseria gonorrhoeae in the discussion should be italicised.

Also in the discussion, please address some of the shortcomings of your study, for example, you discussed an excellent point about macrophage populations in your response to comments, and that would be best placed in the discussion. I appreciate that you have added a section on UPEC-orchitis, which I agree with.

The program used for statistical analysis and graphic is still missing from figure legends.

Please ensure that all of the product details are in your methods section, particularly for any new information added, e.g. tribromoethanol.

***** Reviewer's comments *****

Referee #1 (Comments on Novelty/Model System for Author):

The authors have revised the text to address my concerns which is now clearer.

Response : We are happy to see Referee's satisfaction with our addressing and the revised manuscript.

Referee #2 (Comments on Novelty/Model System for Author):

As per review 1. Additionally, the authors have greatly improved the clarity and believability of the data by providing quantitative data.

Response : We appreciate the Review's acknowledgment for our efforts to improve the manuscript, and pleased to address the remaining concerns.

Referee #2 (Remarks for Author):

The authors have clearly put in a great deal of effort to improve this manuscript. The flow and accuracy is much improved. This reviewer appreciates the effort put into providing quantitative data. Almost all of my comments have been addressed.

Response : We appreciate the Review's acknowledgment for our efforts to improve the manuscript,

and pleased to address the remaining concerns.

Figure 1D still appears blurry, so I question the authors claim that they replaced this image with one that has been saved correctly.

Response : We apologize for this oversight, and we have replaced the image for NLRP3 of Sham group which was blurry in the revised manuscript. Moreover, we have carefully checked all images in Figure 1D, and the immunostaining images appear fine.

I could not see Figure III, which has the full set of uncropped blots, I'm not sure whether I just don't have access to this and will refer this to the editor.

Response : We provided **Figure III** to both the Reviewers and the editor, and have provided **Figure III** for full sized and uncropped blots again (see below).

Figure III: The raw picture of NLRP3, ASC, pro-Caspase-1, cleaved-Caspase-1, pro-Caspase-11 full length GSDMD, cleaved GSDMD, and β -Actin in testes from Sham or UPEC-treated WT mice.

Neisseria gonorrhoeae in the discussion should be italicised.

Response : In light of this suggestion, we have changed *Neisseria gonorrhoeae* into italicized form in the revised manuscript.

Also in the discussion, please address some of the shortcomings of your study, for example, you discussed an excellent point about macrophage populations in your response to comments, and that would be best placed in the discussion. I appreciate that you have added a section on UPEC-orchitis, which I agree with.

Response : In light of this suggestion, we have added the limitations of study about macrophage population into the discussion in the revised manuscript.

The program used for statistical analysis and graphic is still missing from figure legends.

Response : We have carefully checked manuscript, and have added statistical analysis in all figure legends, e.g., adding the annotated p values in figure 1d.

Please ensure that all of the product details are in your methods section, particularly for any new information added, e.g. tribromoethanol.

Response :We welcome the opportunity to clarify the product details in methods section, including tribromoethanol (Sigma, T48402), protease inhibitor cocktail (Sigma, P8340), and TRIzol reagent (Thermo, 15596018). We have included the product details in the revised manuscript.

In summary, we appreciate all of the Reviewer's and Editor's very constructive comments and suggestions. We thank for considering our revised manuscript for publication in EMBO Molecular Medicine.

Kind Regards

Shuo Yang

Corresponding Author

1st Dec 2023

Dear Prof. Yang,

We are pleased to inform you that your manuscript is accepted for publication and is now being sent to our publisher to be included in the next available issue of EMBO Molecular Medicine.
